# Signatures of TOP1 transcription-associated mutagenesis in cancer and germline

Martin A. M. Reijns[1,22 ✉], David A. Parry[1,22], Thomas C. Williams[1,2,22], Ferran Nadeu[3,4], Rebecca L. Hindshaw[5], Diana O. Rios Szwed[1], Michael D. Nicholson[6], Paula Carroll[1], Shelagh Boyle[7], Romina Royo[8], Alex J. Cornish[9], Hang Xiang[10], Kate Ridout[11], The Genomics England Research Consortium*, Colorectal Cancer Domain UK 100,000 Genomes Project*, Anna Schuh[11], Konrad Aden[10], Claire Palles[5], Elias Campo[3,4,12,13], Tatjana Stankovic[5], Martin S. Taylor[2 ✉] & Andrew P. Jackson[1 ✉]

The mutational landscape is shaped by many processes. Genic regions are vulnerable to mutation but are preferentially protected by transcription-coupled repair[1]. In microorganisms, transcription has been demonstrated to be mutagenic[2,3]; however, the impact of transcription-associated mutagenesis remains to be established in higher eukaryotes[4]. Here we show that ID4—a cancer insertion–deletion (indel) mutation signature of unknown aetiology[5] characterized by short (2 to 5 base pair) deletions—is due to a transcription-associated mutagenesis process. We demonstrate that defective ribonucleotide excision repair in mammals is associated with the ID4 signature, with mutations occurring at a TNT sequence motif, implicating topoisomerase 1 (TOP1) activity at sites of genome-embedded ribonucleotides as a mechanistic basis. Such TOP1-mediated deletions occur somatically in cancer, and the ID-TOP1 signature is also found in physiological settings, contributing to genic de novo indel mutations in the germline. Thus, although topoisomerases protect against genome instability by relieving topological stress[6], their activity may also be an important source of mutations in the human genome.

Eukaryotic cells have many strategies to ensure the integrity of their genomes, with high-fidelity DNA replication[7] and DNA-repair processes countering exogenous and endogenous DNA lesions[8]. The process of transcription targets DNA repair machinery to expressed genes, preferentially reducing their mutation rate after DNA damage[1]. Despite this targeted repair, in microorganisms, the process of transcription itself is mutagenic—a phenomenon that is referred to as transcription-associated mutagenesis (TAM)[2,3]. In yeast, topoisomerase 1 (Top1) activity is a major source of TAM and results in a distinctive transcription-dependent signature of 2–5 bp deletions at tandem repeat sequences[9–11]. Genome-embedded ribonucleotides have been established as a cause of Top1-TAM deletions in yeast[12]. Such ribonucleotides are frequently incorporated by DNA polymerases during replication, and represent the most prevalent aberrant nucleotides in the eukaryotic genome[13,14]. These genome-embedded ribonucleotides are normally removed by ribonucleotide excision repair (RER), a process initiated by the heterotrimeric ribonuclease H2 enzyme[15]. However, when Top1 cleaves at embedded ribonucleotides instead of RNase H2, this can result in small deletions[16,17].

In the last decade, the widespread use of genome sequencing has enabled unbiased sampling of human mutations, substantially advancing understanding of mutagenesis in the germline[18] and in neoplasia[19]. Multiple mutational processes act during cancer evolution, and mathematical methods decomposing tumour mutational profiles have been developed to define signatures that may correspond to individual mutagenic mechanisms[19]. This has successfully defined cell-intrinsic, environmental and treatment-related origins for many base-substitution signatures in cancer[20–22]. However, the origin of a substantial number of signatures remains unknown, and some may be artefactual. Recently, cancer signature analysis has been extended to indels[5], small (1–49 bp) insertions and deletions. Such indels are an important class of mutations that contribute substantially to disease-causing germline variants (>20%) and human variation[23].

Here we investigate an indel signature of unknown cause—ID4. We show experimentally that ID4 deletions are increased in RNase-H2-deficient cell lines and cancers and delineate a human TOP1-mediated TAM signature (ID-TOP1) that is relevant to both somatic and germline mutagenesis.

[1]Disease Mechanisms, MRC Human Genetics Unit, Institute of Genetics and Cancer, The University of Edinburgh, Edinburgh, UK. [2]Biomedical Genomics, MRC Human Genetics Unit, Institute of Genetics and Cancer, The University of Edinburgh, Edinburgh, UK. [3]Institut d'Investigacions Biomèdiques August Pi i Sunyer (IDIBAPS), Barcelona, Spain. [4]Centro de Investigación Biomédica en Red de Cáncer (CIBERONC), Madrid, Spain. [5]Institute of Cancer and Genomic Sciences, University of Birmingham, Birmingham, UK. [6]Cancer Research UK Edinburgh Centre, Institute of Genetics and Cancer, The University of Edinburgh, Edinburgh, UK. [7]Genome Regulation, MRC Human Genetics Unit, Institute of Genetics and Cancer, The University of Edinburgh, Edinburgh, UK. [8]Barcelona Supercomputing Center (BSC), Barcelona, Spain. [9]The Institute of Cancer Research, London, UK. [10]Institute of Clinical Molecular Biology, Christian-Albrechts-University and University Hospital Schleswig-Holstein, Kiel, Germany. [11]Department of Oncology, University of Oxford, Oxford, UK. [12]Hospital Clínic of Barcelona, Barcelona, Spain. [13]Departament de Fonaments Clínics, Universitat de Barcelona, Barcelona, Spain. [22]These authors contributed equally: Martin A. M. Reijns, David A. Parry, Thomas C. Williams. *A list of authors and their affiliations appears at the end of the paper. ✉e-mail: martin.reijns@ed.ac.uk; martin.taylor@ed.ac.uk; andrew.jackson@ed.ac.uk

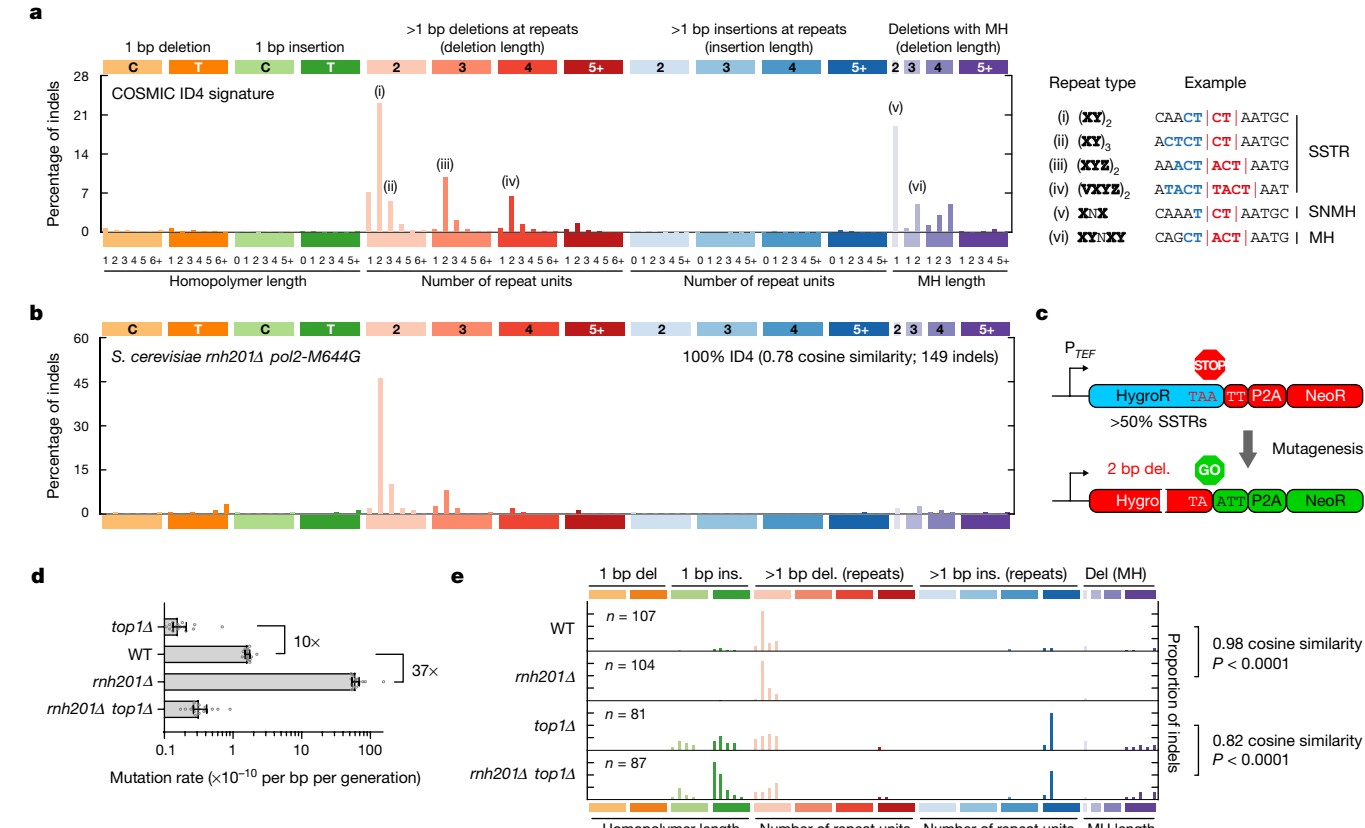

**Fig. 1 | Top1-dependent deletions in *S. cerevisiae* resemble ID4, a cancer mutational signature of unknown aetiology. a**, The ID4 signature comprises small deletions (typically 2, 3 or 4 bp in size) of one repeat unit at SSTR and MH sites. Repeated sequences (i–vi) are shown in bold and colour. Deletions are shown in red. **b**, Indel mutations similar to those detected in ID4 accumulate genome-wide in yeast with high levels of genome-embedded ribonucleotides. Reanalysis of WGS data for *rnh201Δ pol2-M644G* yeast[25]. **c**, Schematic of a frameshift mutation reporter containing many 2 bp SSTRs. Frameshift mutations in HygroR result in neomycin-resistant yeast colonies. P_TEF, TEF promoter; P2A, self-cleaving peptide. **d**, **e**, Fluctuation assays demonstrated that Top1-mediated 2 bp SSTR mutations occur in wild-type and RNase-H2-deficient (*rnh201Δ*) backgrounds. **d**, Mutation rates for *n* = 16 independent cultures per strain. Data are median ± 95% confidence intervals. **e**, WT and *rnh201Δ* have similar indel mutation spectra, and differ from *top1Δ* strains. Spectra of neomycin-resistant colonies. *n* indicates the number of independent indels detected. Cosine similarity *P* values were empirically determined (Extended Data Fig. 2e, f). Del, deletion; ins, insertion.

## ID4, a distinct cancer indel signature

The ID4 cancer signature, as categorized by COSMIC[24], comprises 2–5 bp deletions, often with the loss of a single repeat unit in short repeat sequences[5]. These most commonly occur in regions in which the deleted sequence is repeated one, two or three times in tandem (Fig. 1a). Hereafter we use the term short-short tandem repeats (SSTRs) to distinguish between such short tandem repeats (STRs) with less than 5 repeats (that is, less than 6 repeat units) and microsatellite STRs with many repeats. In addition to these SSTR deletions, ID4 is characterized by small deletions at sequences with microhomology (MH), in particular, 2 bp deletions with single-nucleotide MH (SNMH). Both features are distinct from cancer deletion signatures resulting from other well-recognized mechanisms such as replication slippage and non-homologous/MH-mediated end joining (NHEJ/MMEJ) (Extended Data Fig. 1a, b). In support of a distinct aetiology, SSTR and SNMH deletions are not apparent in cancer associated with homologous recombination or mismatch repair deficiency, which are expected to have higher levels of MMEJ and replication slippage mutagenesis, respectively (Extended Data Fig. 1c, d).

## ID4 resembles a yeast mutation signature

Noting similarities to a Top1-induced TAM (Top1-TAM) in *Saccharomyces cerevisiae*, we reanalysed published genome-wide mutation-accumulation experiments performed with *rnh201Δ pol2-M644G* yeast[25]. This strain is particularly susceptible to Top1-TAM as it accumulates genome-embedded ribonucleotides at high levels due to RNase H2/RER deficiency and enhanced ribonucleotide incorporation by a steric-gate mutation at the catalytic site of the replicative polymerase Pol ε[26]. Similarities to the ID4 signature were apparent with a comparable pattern of small deletions at SSTRs, although mutational events at sites of SNMH were not evident in the yeast data (Fig. 1b). As more than 1 million ribonucleotides are incorporated by DNA polymerases per replicating mouse cell[14], we reasoned that genome-embedded ribonucleotides might cause similar mutational events in mammalian cells. To experimentally assess whether TAM contributes to indel formation in human RER-deficient cells, we developed a reporter to enable sensitive and specific detection of mutational events arising from TOP1 activity in both yeast and mammals.

## Top1-dependent deletions in yeast

Mutation rates are routinely measured in *S. cerevisiae* using well-characterized but species-specific selectable markers (LYS2, URA3, CAN1). Thus, to establish a system that could be transferred between yeast and mammalian cells, we used an approach inspired by the Traffic Light reporter assay[27], incorporating both positive and negative selection cassettes in a single transcriptional unit (Fig. 1c). The hygromycin-resistance gene (HygroR) was used both as the

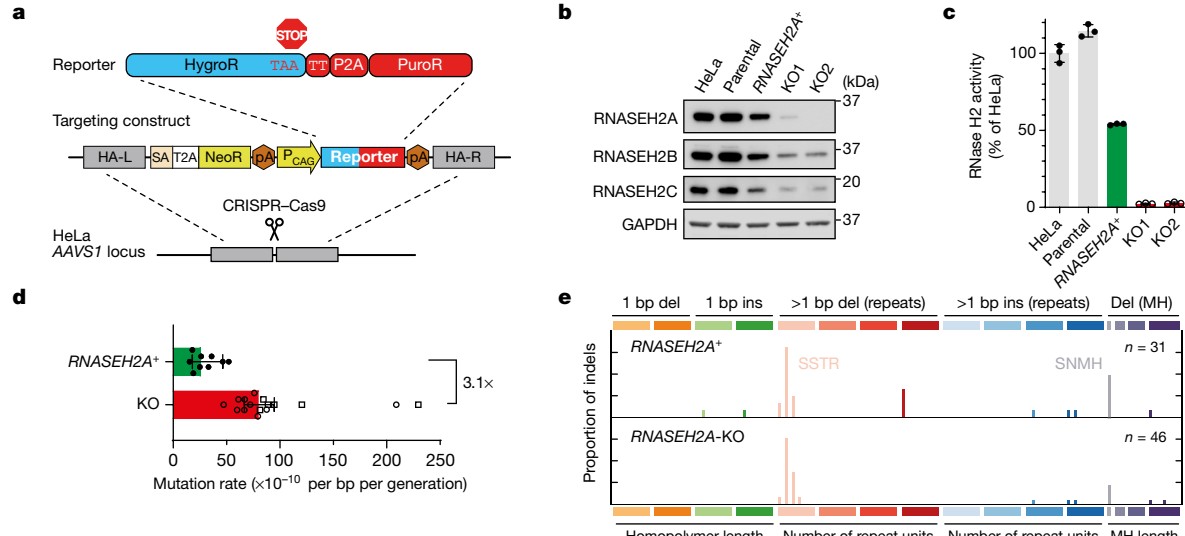

**Fig. 2 | SSTR deletions of 2 bp are increased in RNase-H2-null HeLa cells.**
**a**, Schematic of the reporter targeting the *AAVS1* safe harbour locus to generate reporter cells (Extended Data Fig. 3). HA, homology arm; L, left; R, right. **b**, **c**, Validation of *RNASEH2A*-KO reporter clones. **b**, Immunoblot analysis of cell lysates detecting the three RNase H2 subunits. GAPDH was used as the loading control. Gel source data are provided in Supplementary Fig. 1. **c**, Cellular RNase H2 enzyme activity. Data are mean ± s.d. *n* = 3 technical replicates. HeLa, no modification; parental, HeLa with reporter (grey); KO1 and KO2, CRISPR-mediated *RNASEH2A*-KO clones (red); *RNASEH2A*⁺, CRISPR-edited

reporter clone retaining RNase H2 activity (green). **d**, Fluctuation assays establish a significantly increased mutation rate in RNase-H2-null (KO) cells ($P = 2 \times 10^{-6}$). Statistical analysis was performed using a two-sided Mann–Whitney test. Data are median ± 95% confidence intervals. The data points show the rates for independent cultures. *n* = 9 (RNase H2 proficient, *RNASEH2A*⁺); *n* = 10 (KO1, open circles) and *n* = 6 (KO2, open squares). **e**, 2 bp SSTR and SNMH deletions are frequent in both *RNASEH2A*⁺ and *RNASEH2A*-KO cells. Indel mutation spectra. *n* shows the number of indels identified by sequencing colonies from independent cultures.

mutational target and negative selection marker. Indels causing a 2 bp frameshift within HygroR, including 2 bp deletions, result in translation of an otherwise out-of-frame P2A self-cleaving peptide and the neomycin-resistance (NeoR) gene, permitting positive selection of mutated colonies with neomycin (Extended Data Fig. 2a). To enrich the target for 2 bp tandem repeats, in silico redesign incorporated synonymous substitutions such that SSTRs accounted for >50% of the HygroR open reading frame.

For validation, the reporter was inserted into the *S. cerevisiae* genome and fluctuation assays were performed to assess the mutation rates in strains deficient for RER and/or Top1. A 37-fold increase in mutation rate was observed for the *rnh201Δ* (RNase H2 null) strain compared with the wild type (Fig. 1d), with a mutation rate of $6.1 \times 10^{-9}$ per bp per generation (95% confidence interval = $5.4 \times 10^{-9}$–$6.9 \times 10^{-9}$), whereas the increased mutation rate was abolished in the *rnh201Δ top1Δ* double-mutant strain, consistent with Top1-dependent mutagenesis at genome-embedded ribonucleotides[12,28]. Notably, there was a 10-fold decrease in the mutation rate for *top1Δ* compared with the wild-type strain, and a 35-fold decrease in 2 bp SSTR deletions (Extended Data Fig. 2b), consistent with previous reports[10,11]. Furthermore, the observed mutational spectrum was most similar for wild-type and *rnh201Δ* strains, but substantially different compared with the *top1Δ* and *rnh201Δ top1Δ* strains (Fig. 1e and Extended Data Fig. 2c–f). Taken together, we conclude that the same Top1-mediated mutations occur, albeit at different frequencies, in wild-type cells when RER is functional and in RNase-H2-deficient strains when elevated levels of ribonucleotides are present in the genome.

## TOP1-mediated mutations in human cells

Having validated the reporter in yeast, the same 2 bp repeat-enriched HygroR sequence was used to determine whether TOP1-mediated mutagenesis at embedded ribonucleotides is conserved in human cells (Fig. 2 and Extended Data Fig. 2g). NeoR was replaced by the puromycin-resistance (PuroR) gene, with reporter expression driven from the mammalian ubiquitous CAG promoter, permitting rapid

antibiotic selection in mammalian cells. This modified reporter was inserted at the *AAVS1* safe harbour locus in HeLa cells (Fig. 2a and Extended Data Fig. 3a–e). CRISPR–Cas9-mediated genome editing targeting the catalytic site of *RNASEH2A* was then used to generate two independent knockout (KO) reporter clones, alongside a control clone that had also been processed through the editing and clonal selection steps (Fig 2b, c and Extended Data Fig. 3). The control clone retained RNase H2 activity, whereas there was complete loss of cellular RNase H2 activity in the KO clones, accompanied by high levels of ribonucleotides in genomic DNA (Fig. 2b, c and Extended Data Fig. 3f, g).

In fluctuation assays, RNase-H2-null clones demonstrated a significant 3.1-fold increase in mutation rate (Fig. 2d) and 5.2-fold more 2 bp SSTR deletions (Extended Data Fig. 3h) compared with RNase-H2-proficient cells (*RNASEH2A*⁺), consistent with conservation of TOP1-directed mutagenesis in human cells. As in yeast (Fig. 1e), the overall mutational profile of reporter mutations was similar between RNase-H2-proficient and null HeLa cells (cosine similarity = 0.89, $P < 10^{-4}$), predominantly comprising 2 bp SSTR deletions (Fig. 2e).

The mutation rate for RNase-H2-null HeLa cells ($8.0 \times 10^{-9}$ per bp per generation; 95% confidence interval = $6.7$–$9.5 \times 10^{-9}$) was similar to that observed for *rnh201Δ* yeast (Fig. 1d), whereas the rate was substantially higher for *RNASEH2A*⁺ control cells compared with wild-type yeast. However, the increased mutation rate in RNase-H2-null HeLa cells probably underestimates the true impact of RER deficiency in human cells as, despite the fact that the control *RNASEH2A*⁺ HeLa reporter cells retained protein expression (Fig. 2b), the clone had also acquired mutations at the CRISPR editing site that reduced enzymatic activity (Fig 2c), causing a moderate increase in genomic ribonucleotide content (Extended Data Fig. 3f, g).

To confirm these findings, we used a complementary approach to establish the relevance of such mutational events genome-wide, performing mutation-accumulation experiments using human hTERT RPE-1 (*TP53*⁻/⁻) diploid cell lines. Ancestral populations for RNase-H2-wild-type and RNase-H2-null cells (*RNASEH2A*-KO or *RNASEH2B*-KO; Extended Data Fig. 4a–d) were established after initial

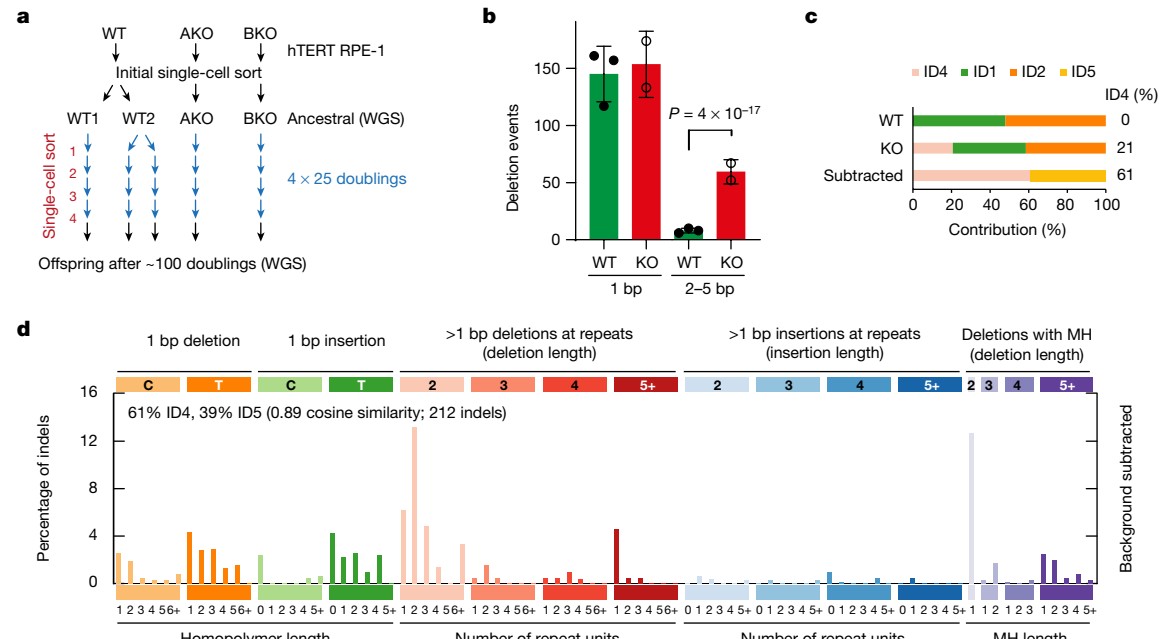

**Fig. 3 | ID4 SSTR and MH mutations are increased genome-wide in RNase-H2-deficient RPE-1 cells. a**, Schematic of the mutation-accumulation experiment. Long-term culture of hTERT RPE-1 *TP53*[−/−] RNase-H2-wildtype (WT) and RNase-H2-null cell lines (*RNASEH2A*-KO (AKO), *RNASEH2B*-KO (BKO)) bottlenecked every 25 doublings by single-cell sorting. **b**, Mutations acquired during long-term culture were significantly enriched for 2–5 bp deletions in RNase-H2-null cells, but the other mutation categories were not (Extended Data Fig. 4e). Data are mean ± s.d. Statistical analysis was performed using two-sided Fisher's exact tests with Bonferroni correction, comparing wild type (counts pooled from *n* = 3 independent clones) versus KO (*n* = 2 independent clones) for 2–5 bp deletions versus all of the other indel types. **c**, **d**, ID4 occurs in RNase-H2-null cells (**c**) and is the major signature after subtracting background mutations that are observed in wild-type cells (**d**).

single-cell sorting, and the clones were then grown for approximately 100 generations. Single-cell sorting was performed every 25 generations, creating bottlenecks to capture accumulating mutations (Fig. 3a). Combined variant calling on whole-genome sequencing (WGS) from paired ancestral and end-point cultures identified a total of 1,698 acquired high-confidence indel mutations, captured by at least 3 out of 4 variant callers. Consistent with TOP1-mediated mutagenesis, among all indel categories, only 2–5 bp deletions were found to be substantially (7.4-fold) and significantly enriched in RNase-H2-null RPE-1 cells compared with the wild type (Fig. 3b and Extended Data Fig. 4e, f), with an estimated rate of $1.1 \times 10^{-10}$ 2–5 bp deletions per generation per bp for KO and $1.4 \times 10^{-11}$ for the wild type. Of these deletions in RNase-H2-null cells, 82% were 2 bp deletions, of which 48% were at SSTRs (Extended Data Fig. 4g). Furthermore, signature decomposition using SigProfilerExtractor[5] reported a 21% ID4 contribution in RNase-H2-null cells that increased to 61% after background mutation patterns were subtracted to identify RER-deficiency-specific mutation signatures (Fig. 3c, d and Extended Data Fig. 5). The ID4 signature was substantially enriched in transcribed genomic regions (Extended Data Fig. 5e). ID5, a clock-like signature[5], was also enriched in KO cells, probably due to slower growth and the longer culture time needed to achieve the same number of doublings for RNase-H2-null cells[14].

## MH deletions specific to mammals

Small deletions at sequences with MH are an additional feature of ID4 (Fig. 1a) that is not observed in *rnh201Δ pol2-M644G* yeast (Fig. 1b). However, consistent with a ribonucleotide-induced mutational origin in mammalian cells, MH deletions are observed frequently in RNase-H2-deficient RPE-1 cells, in which SNMH sites account for 31% of 2 bp deletions, indicating that, in humans, deletions at SNMH sites share the same aetiology as deletions occurring at SSTRs. Taken together, our reporter and mutation-accumulation experiments demonstrate that

genome-embedded ribonucleotides cause a similar mutational signature in yeast and mammalian cells. Thus, topoisomerase-1-mediated mutagenesis probably also occurs in humans and is associated with 2–5 bp deletions at SSTR and SNMH sequences.

## ID4 mutations in a mouse cancer model

To determine whether TOP1-induced mutations resulting in the ID4 signature can be detected in vivo, we next studied an RER-deficient mouse cancer model in which *Villin-cre* conditional deletion of *Rnaseh2b* and *Tp53* results in intestinal malignancy[29]. WGS analysis of paired tumour–normal tissue samples from 6 mice identified a total of 989 high-confidence tumour-specific somatic indels. An analysis of the resulting mutational signature established that ID4 substantially contributed in all tumours (Fig. 4a, b and Extended Data Fig. 6a), accounting for 32% of acquired indels. Consistent with a transcription-associated process, the ID4 signature was again most evident in transcribed genomic regions (Fig. 4b). The commonly occurring cancer signatures[5] ID1, ID2 and ID5 were also observed, consistent with expectations of multiple mutational processes active in neoplasia.

The observed ID4 mutation spectrum corresponded closely to that observed in the RPE-1 mutation-accumulation experiment: 28% of indels were at 2–5 bp deletions, of which the majority were again 2 bp deletions (82%) predominantly at SSTRs (51%) and sites of SNMH (34%) (Extended Data Fig. 6b, c). This is consistent with the occurrence of TOP1-induced somatic mutations at genome-embedded ribonucleotides in vivo, conserved across different tissue and cellular contexts, and shows that this process can be detected in a cancer setting.

## A sequence motif for ID4 mutations

Although COSMIC defines the ID4 signature on the basis of indel size and repeat/MH context (Fig. 1a), the number of indels in the mouse

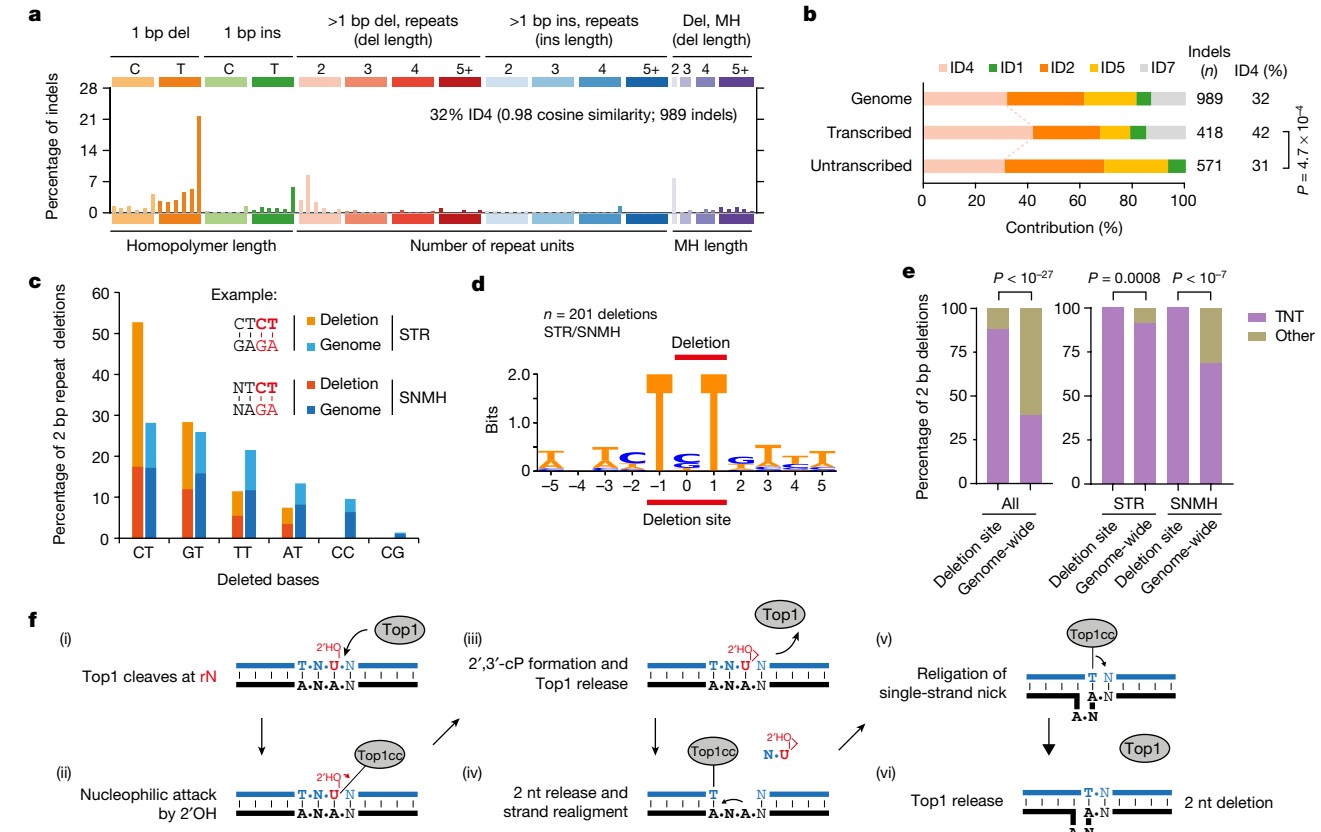

**Fig. 4 | RER-deficient tumours have an ID4 signature associated with transcription and a TNT sequence motif. a**, ID4 contributes substantially to the mutational spectrum of *Rnaseh2b*-KO mouse intestinal tumours (WGS, paired tumour–normal samples from *n* = 6 mice). **b**, ID4 contribution is greater in transcribed regions of the genome. Statistical analysis was performed using a two-sided Fisher's exact test, comparing ID4 versus other indels. *n* = 969 indels from 6 biologically independent tumours. **c**, 2 bp STR/SNMH deletions have biased sequence composition. Genome, frequency of dinucleotides in STR/SNMH sequences in the mappable genome. Deletions are right aligned and indicated by bold red font. **d**, **e**, A TNT sequence motif is present at all 2 bp

STR and SNMH deletions. **d**, Sequence logo: two-bit representation of the sequence context of 2 bp deletions at STR and SNMH sequences. **e**, Deletion sites are significantly enriched for the TNT sequence motif compared with genome-wide occurrence, for all genome sequences, as well as STR and SNMH sites. Statistical analysis was performed using two-sided Fisher's exact tests, comparing observed versus expected. *n* = 228 (all, *P* = 1.7 × 10⁻²⁸), *n* = 124 (STR, *P* = 0.0008), *n* = 77 (SNMH, *P* = 1.4 × 10⁻⁸) deletions in 6 biologically independent tumours. **f**, Model for TOP1-mediated mutations at TNT sequences containing embedded ribonucleotides, in which strand realignment results in a two-nucleotide deletion (see main text). nt, nucleotide.

RER-deficient tumour model enabled us to further investigate the characteristics of mammalian topoisomerase-1-induced mutations. We focussed our analysis on 2 bp deletions, as such events represented 81% of >1 bp deletions in the context of tandem repeats and 85% of deletions in sequences with MH.

First, we classified all 2 bp deletions at STR/SNMH sequences into six non-redundant dinucleotide classes, grouping together complementary sequences (Fig. 4c). We noted that the deleted sequences substantially deviated from genome-wide frequencies, with a complete absence of CC/GG and CG/GC deletions, as well as an overrepresentation of the CT category (containing CT, TC, GA and AG deletions). All of the observed deletions therefore included at least one thymidine (T), which functionally could be accounted for by the very strong preference of mammalian topoisomerase 1 to cleave at a phosphodiester bond with a T immediately upstream[30].

Next, to investigate the wider sequence context, we aligned sequences containing all 228 two-bp deletions (Extended Data Fig. 6d), which indicated that deletions preferentially occur when T nucleotides are spaced at a two-base interval. Indeed, this TNT motif was present in 100% of SNMH (*n* = 77) and STR sites (*n* = 124), providing a common unifying sequence context for both deletion types (Fig. 4d), a finding that was replicated in both our RPE-1 (Extended Data Fig. 6e) and yeast datasets (Extended Data Fig. 7). We found that TNT is substantially

over-represented at deletion sites compared with the genome-wide null expectation. Furthermore, although the TNT motif is common at tandem repeat sequences, 2 bp deletions at this motif are still significantly enriched when considering the occurrence of 2 bp STR and SNMH sequences in mouse and human genomes (Fig. 4e and Extended Data Fig. 6f), and STR sequences in the yeast genome (Extended Data Fig. 7).

## A model for TOP1-mediated deletions

To account for thymidines spaced at a two-base interval and the occurrence of mammalian SNMH deletions, we developed a revised model based on the established strand realignment model for yeast Top1-mediated mutagenesis[12,16,17]. In this 'TNT model', TOP1 cleaves preferentially 3′ of an embedded ribouridine, with nucleophilic attack by the 2′-OH of the ribose ring resulting in TOP1 release and formation of a non-ligatable nick with a terminal 2′,3′-cyclic phosphate (Fig. 4f (i–iii)). This then provides a substrate for TOP1 cleavage 2 bp or more upstream[17], preferentially at a thymidine[30]. When this second cleavage event happens at a base that is identical to that of the first cleaved nucleotide—an event that is more likely at STR and MH sequences—strand realignment can then occur, resulting in a nick that is permissive to religation and TOP1 cleavage complex (TOP1cc) reversal (Fig. 4f (iv–vi)). An alternative mechanism of sequential Top1 cleavage, in which

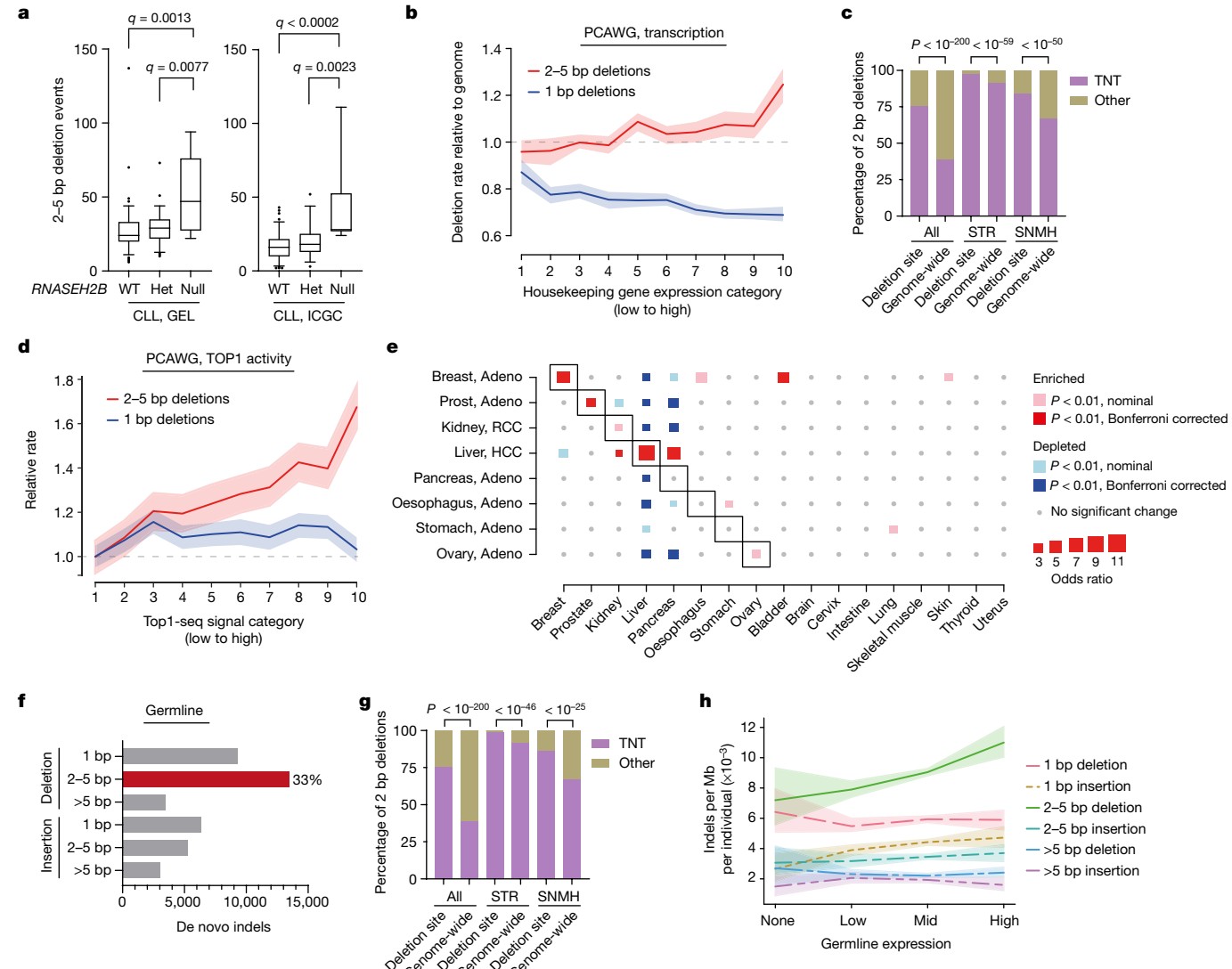

**Fig. 5 | TOP1-mediated deletions in human cancer and germline. a**, Deletions of 2–5 bp are significantly increased in CLL with biallelic *RNASEH2B* deletions (null). For the box plots, the box limits show from 25% to 75%, the centre line shows the median, the whiskers show from 5% to 95% and the data points show values outside the range. For GEL and ICGC, respectively, $n = 116$ and $n = 85$ (wild type); $n = 72$ and $n = 59$ (heterozygous (het)); and $n = 10$ and $n = 6$ (null) tumours. Multiple-testing-corrected $q$ values were determined using two-sided Mann–Whitney $U$-tests. **b–d**, ID-TOP1 deletions are frequent somatic mutations in cancer. **b**, Indels per expression stratum of ubiquitously expressed genes (defined in Extended Data Fig. 8e). The dotted line shows the genome-wide rate. **c**, Deletions of 2 bp preferentially occur at TNT motifs. Statistical analysis was performed using two-sided Fisher's exact tests, comparing observed versus expected. $n = 11,853$ (all; $P < 10^{-200}$), $n = 6,699$ (STR; $P = 1.9 \times 10^{-60}$), $n = 2,872$ (SNMH; $P = 1.5 \times 10^{-51}$) deletions. **d**, Deletions of 2–5 bp increase with TOP1 cleavage activity in ID4-positive PCAWG tumours. The solid lines show the relative deletion rate. The shading shows the 95% confidence intervals from 100 (**b**) or 1,000 (**d**) bootstrap replicates. For **b–d**, $n = 11,853$ biologically

independent tumours[50]. **e**, Deletions of 2–5 bp are enriched at tissue-specific highly transcribed genes in associated cancers. Heat map of significant odds ratio scores (2–5 bp deletions in top 10% tissue-restricted genes versus 2–5 bp deletions in other genes, relative to expected frequency from all other tissues) for normal-tissue–tumour pairs. Statistical analysis was performed using two-sided Fisher's exact tests. Adeno, adenocarcinoma; HCC, hepatocellular carcinoma; RCC, renal cell carcinoma. **f–h**, ID-TOP1 deletions are frequent human de novo mutations that are enriched in highly transcribed germ cell genes. **f**, Deletions of 2–5 bp are the most common indels in the human germline. Gene4Denovo WGS data[39] ($n = 40,936$ indels). **g**, TNT sequence motif is significantly enriched in de novo 2 bp deletions. Statistical analysis was performed using two-sided Fisher's exact tests, comparing observed versus expected. $n = 5,569$ 2 bp deletions ($P < 10^{-200}$), at STR ($n = 3,294$; $P = 5.2 \times 10^{-47}$) and SNMH sequences ($n = 1,093$; $P = 2.9 \times 10^{-26}$). **h**, The 2–5 bp deletion frequency is correlated with gene transcription level in germ cells. Solid lines, Gene4Denovo indel mutations per individual per Mb. The shading shows the 95% confidence intervals from 100 bootstrap replicates.

double-strand breaks occur due to nicking of opposite strands[31] could not be reconciled with our TNT model, but may account for deletions occurring at non-STR/SNHM sites. Within the TNT motif, deletions were most common at CT and GT dinucleotides in both mammals and yeast (Fig. 4c and Extended Data Figs. 6 and 7b, e), which may be explained, at least in part, by preferential incorporation of ribouridine at CT and GT dinucleotides[32] (Extended Data Fig. 7f).

Implicating TOP1-TAM as the cause of the ID4 signature enables us to include additional features in the definition of this COSMIC signature, namely preference for a TNT sequence motif at 2 bp deletion sites and enrichment in transcribed genes. Hereafter, we refer to this extended definition as ID-TOP1. To establish the relevance of the ID-TOP1 signature for human disease and genetic variation, we next examined publicly available datasets.

## ID-TOP1 in human cancer

*RNASEH2B* is frequently deleted in human cancer, in particular, in chronic lymphocytic leukaemia (CLL), given its proximity to a tumour suppressor locus, the *DLEU2-miR-15a/miR-16-1* microRNA cluster[33]. Such RNase-H2-deficient human cancers should therefore be enriched for the ID4/ID-TOP1 signature. We analysed WGS data for 348 patients with CLL from two independent cohorts[34,35], stratified on the basis of *RNASEH2B*-deletion status. Somatic variant calling identified a significant increase in 2–5 bp deletions in RNase-H2-null tumours (Fig. 5a), whereas other indels were equally represented across wild-type, heterozygous and null categories (Extended Data Fig. 8a). Of the 2–5 bp deletions in tumours with biallelic *RNASEH2B* loss, more than half (57%) were 2 bp deletions, which were predominantly at STR and SNMH sequences and substantially enriched for the TNT motif (Extended Data Fig. 8b, c), consistent with the ID-TOP1 mutational signature. Furthermore, mutational signature decomposition for RNase-H2-null CLL cases confirmed the presence of the ID4 signature, most apparent in genic regions (Extended Data Fig. 8d). We therefore conclude that the ID-TOP1 signature is present in human cancer and enriched in tumours that are RNase H2 deficient.

Topoisomerase 1 also causes mutations in RER-proficient cells[10,11] (Fig. 1d–f) and is therefore likely to cause mutations in other cancers, with deletions expected to occur most frequently in highly transcribed genes[4]. Accordingly, analysis of WGS data across cancer types (International Cancer Genome Consortium (ICGC)/Pan-Cancer Analysis of Whole Genomes (PCAWG)) demonstrated that the 2–5 bp deletion rate correlates with expression levels of ubiquitously expressed genes (Pearson's $r = 0.86$, $P = 0.0014$), with deletions markedly elevated in the most highly expressed genes (Fig. 5b), consistent with previous reports of such deletions in certain cancer genes[36,37]. Examination of 2 bp deletions (42% of 2–5 bp deletions) across cancer types also demonstrated them to be predominantly in STR and SNMH contexts (Extended Data Fig. 8f) and enriched for the TNT sequence motif (Fig. 5c). Furthermore, using a dataset of TOP1 cleavage events captured by TOP1-seq[38], we found that 2–5 bp deletions increase in frequency with TOP1 enzymatic activity, with such deletions more prevalent in regions of high TOP1 activity (Fig. 5d). Similarly, TOP1-ID deletion rates also corresponded to TOP1 activity and transcription level, in contrast to all other deletions (Extended Data Fig. 8g, h). Taken together, this establishes a substantial role for TOP1-mediated mutagenesis in the generation of somatic deletions.

To further examine the role of transcription in deletion mutagenesis of cancer genomes, we identified genes that are highly expressed, but only in certain tissues. For prostate adenocarcinoma, highly expressed prostate-restricted genes were significantly enriched for 2–5 bp deletion mutations compared with other genes in this cancer type, as well as the same genes in other cancers (two-tailed Fisher's exact test, odds ratio = 3.5, $P = 2.5 \times 10^{-8}$ after Bonferroni correction; Extended Data Fig. 8i). Importantly, this analysis considers the same sets of genes between cancer types and therefore rules out sequence composition biases as a confounding effect for elevated ID-TOP1 mutagenesis in highly expressed genes. Extending this approach in an all-versus-all comparison between 8 cancer types and 17 tissues demonstrated specificity between high expression in a tissue of origin and enrichment for 2–5 bp deletions (Fig. 5e). These results extend the relevance of TOP1-mediated mutagenesis to other cancers, confirm the ID-TOP1 mutational signature to be transcription-associated and support the occurrence of TAM in humans.

## TOP1-mediated deletions in the germline

TOP1 is ubiquitously expressed, so we reasoned that it could cause germline as well as somatic mutations. To investigate this possibility, we examined mutations from parent–child trio WGS studies in the Gene4Denovo database[39]. De novo mutations identified in such datasets represent germline events, as they occur in germ cells or during early embryonic cell divisions. Strikingly, 2–5 bp deletions were the largest category identified, accounting for 33% of the 40,936 de novo indels (Fig. 5f), and the majority of these were compatible with the ID-TOP1 signature. Analysis of 2 bp deletions (41% of 2–5 bp deletions) demonstrated that most occur at SSTR or MH sites (Extended Data Fig. 9a, b), with enrichment of the TNT sequence motif both genome wide and in the context of STR/SNMH sites (Fig. 5g and Extended Data Fig. 9c). Similarly, for 3 and 4 bp deletions, respectively, TNNT and TNNNT motifs were significantly over-represented compared with genome-wide expectation (Extended Data Fig. 9d), supporting sequential TOP1 cleavage and strand realignment as the underlying cause. Consistent with TOP1-TAM aetiology, 2–5 bp deletion and ID-TOP1 deletion frequency correlated with transcript expression in male germ cells (Fig. 5h and Extended Data Fig. 9e). We therefore conclude that the ID-TOP1 mutational signature also occurs in the human germline, implicating TOP1-induced strand realignment mutagenesis as an important mutational process in mammalian cells.

## Discussion

Here we establish a biological basis for the ID4 cancer signature[5], experimentally demonstrating that it occurs in RNase-H2-deficient cells both in vitro and in vivo. This implicates TOP1-mediated cleavage at genome-embedded ribonucleotides as its cause. TOP1 is cell-essential in mammals and it is therefore not possible to similarly confirm a genetic dependency on TOP1 in human cells, as has been done in yeast[12]. However, conservation of this mechanism across eukaryotes is supported by us finding a topoisomerase-1-dependent TNT deletion motif that is present in both yeast and humans, and demonstrating that deletion frequency is dependent on human TOP1 activity levels. Previously published research also provides evidence for TOP1 mutagenesis at ribonucleotide sites in humans. The reversible transesterification reaction of type 1 topoisomerases is conserved from yeast to humans[6], and human TOP1 has site-specific activity for ribonucleotides[40], causing DNA breaks in mammalian RNase-H2-deficient cells[33]. Furthermore, the generation of 2 bp deletions through sequential TOP1 cleavage at embedded ribonucleotides has been biochemically reconstituted with both human and yeast enzymes[17,31].

We define additional features of this ID-TOP1 mutational signature, with deletions strongly enriched at TNT motifs in both yeast and mammals, a sequence context that is specific to topoisomerase 1 (Extended Data Fig. 7g, h) and deletions that are most frequent in highly transcribed regions. As a consequence, we show that a TAM process that was first identified in yeast[10,12,41] is relevant to higher eukaryotes, establishing TOP1-induced mutagenesis as an important process for human variation and disease. Additional signatures associated with topoisomerases or indeed RNase H2 may be identified in the future, particularly given that ID17 has recently been linked to TOP2A[K743N] cancers[42].

The substantial contribution of ID-TOP1 deletions to germline mutagenesis has particular importance given that such deletions will be disproportionately disruptive, particularly in transcribed regions. Notably, such deletions occur in the context of normal RER function, consistent with the mutagenic potential of topoisomerase 1 in physiological wild-type settings[10,11] (Fig. 1d). Given that genome-embedded ribonucleotides are the most common endogenous lesion in replicating mammalian cells[14], they are the most likely sites of TOP1-TAM mutagenesis, where TOP1 could cleave before their removal by RNase H2-dependent RER. Processing of TOP1cc may be an alternative, less frequent source of 2–5 bp deletions[41], but we did not detect ID4 in cancers treated with topoisomerase 1 inhibitor (Extended Data Fig. 8j). The canonical function of TOP1 is to relieve DNA topological stress, arising during both transcription and replication[6] (Extended Data Fig. 10). Thus, TOP1-mediated deletions are not restricted to transcribed regions

of the genome, with deletions also evident in non-genic regions with high TOP1 activity (Extended Data Fig. 8k). However, overall, enhanced TOP1 activity associated with transcription accounts for more frequent mutagenesis within genes.

Given the essential nature of topoisomerase activity across tissues and cell states, TOP1-mediated mutagenesis probably occurs in many contexts. The frequent TOP1-mediated human germline mutations (Fig. 5i–k) and the identification of ID4 at early embryonic stages[43] suggest developmental vulnerability to TOP1-TAM. Moreover, 2–5 bp somatic deletions at SSTRs are also observed at high frequency in non-dividing neurons[36], and ID4 has been identified in multiple tumour types[5]. As such, this mutational process is likely to be important not only in cancers with RER deficiency, but also those with high TOP1 activity and tumours with defects in relevant repair mechanisms, such as enzymes that process TOP1cc[6] or non-ligatable TOP1-induced nicks[44–46]. Furthermore, alternative RER pathways may exist[47] that could reduce TOP1 mutagenesis. The ID-TOP1 signature may provide a useful biomarker with potential future diagnostic and therapeutic use[48], for example, as an indicator of TOP1-induced genome instability targetable by PARP or ATR inhibitors[33,49].

In conclusion, alongside its essential role in relieving DNA torsional stress, TOP1 also drives mutagenesis in somatic and germline contexts, relevant to neoplasia, inherited disease and human variation.

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

**The Genomics England Research Consortium**

**John C. Ambrose**[14]**, Prabhu Arumugam**[14]**, Roel Bevers**[14]**, Marta Bleda**[14]**, Freya Boardman-Pretty**[14,15]**, Christopher R. Boustred**[14]**, Helen Brittain**[14]**, Mark J. Caulfield**[14,15]**,

Georgia C. Chan[14], Greg Elgar[14,15], Tom Fowler[14], Adam Giess[14], Angela Hamblin[14], Shirley Henderson[14,15], Tim J. P. Hubbard[14], Rob Jackson[14], Louise J. Jones[14,15], Dalia Kasperaviciute[14,15], Melis Kayikci[14], Athanasios Kousathanas[14], Lea Lahnstein[14], Sarah E. A. Leigh[14], Ivonne U. S. Leong[14], Javier F. Lopez[14], Fiona Maleady-Crowe[14], Meriel McEntagart[14], Federico Minneci[14], Loukas Moutsianas[14,15], Michael Mueller[14,15], Nirupa Murugaesu[14], Anna C. Need[14,15], Peter O'Donovan[14], Chris A. Odhams[14], Christine Patch[14,15], Mariana Buongermino Pereira[14], Daniel Perez-Gil[14], John Pullinger[14], Tahrima Rahim[14], Augusto Rendon[14], Tim Rogers[14], Kevin Savage[14], Kushmita Sawant[14], Richard H. Scott[14], Afshan Siddiq[14], Alexander Sieghart[14], Samuel C. Smith[14], Alona Sosinsky[14,15], Alexander Stuckey[14], Mélanie Tanguy[14], Ana Lisa Taylor Tavares[14], Ellen R. A. Thomas[14,15], Simon R. Thompson[14], Arianna Tucci[14,15], Matthew J. Welland[14], Eleanor Williams[14], Katarzyna Witkowska[14,15] & Suzanne M. Wood[14,15]

[14]Genomics England, London, UK. [15]William Harvey Research Institute, Queen Mary University of London, London, UK.

**Colorectal Cancer Domain UK 100,000 Genomes Project**

Daniel Chubb[9], Alex Cornish[9], Ben Kinnersley[9], Richard Houlston[9], David Wedge[16], Andreas Gruber[16], Anna Frangou[17], William Cross[18], Trevor Graham[19], Andrea Sottoriva[9], Gulio Caravagna[9], Nuria Lopez-Bigas[20], Claudia Arnedo-Pac[20], David Church[17], Richard Culliford[9], Steve Thorn[6], Phil Quirke[21], Henry Wood[21], Ian Tomlinson[6] & Boris Noyvert[5]

[16]Manchester Interdisciplinary Biocentre, University of Manchester, Manchester, UK. [17]Wellcome Centre for Human Genetics, Oxford, UK. [18]Cancer Institute, University College London, London, UK. [19]Barts Cancer Institute, Barts and The London School of Medicine and Dentistry, Queen Mary University of London, London, UK. [20]Institute for Research in Biomedicine (IRB Barcelona), The Barcelona Institute of Science and Technology (BIST), Barcelona, Spain. [21]Pathology and Data Analytics, Leeds Institute of Medical Research, St James's University Hospital, University of Leeds, Leeds, UK.

## Methods

### Plasmids

A description of all plasmids used in this work is provided in Supplementary Table 1. The *S. cerevisiae* reporter was generated by DNA synthesis (GeneArt Gene Synthesis, Thermo Fisher Scientific; gBlocks Gene Fragments, IDT) and conventional cloning (restriction, ligation and Quikchange site-directed mutagenesis). The final construct (pTCW12) was used for *S. cerevisiae* reporter strain construction and fluctuation assays. A Gateway compatible reporter construct for mammalian cells (pTCW14) was generated in a similar manner using a combination of DNA synthesis and conventional cloning strategies. Gateway cloning was then used to move the reporter cassette into pAAVS-Nst-CAG-Dest (a gift from K. Woltjen; Addgene plasmid, 80489)[51] to generate pTCW15 for targetting it to the human *AAVS1* locus.

### In silico redesign of the hygromycin-resistance gene

To increase the frequency of 2 bp tandem repeats, synonymous substitutions were introduced in the 1 kb *hph* coding sequence, the *Klebsiella pneumoniae* hygromycin-resistance gene (HygroR)[52]. Using Python, a 5-codon (15-base) sliding window was moved one codon at a time, to identify all possible synonymous permutations. Permutations were ranked on the basis of tandem dinucleotide repeat sequence length, with the highest ranking sequences used to replace whole codons, prioritising dinucleotide repeats over mononucleotide repeats. Edited codons were then censored from subsequent permutation. Subsequently, to eliminate stop codons that would arise after a 2 bp deletion or equivalent frameshift mutations, further synonymous changes were made, where possible preserving tandem repeat sequences.

### Yeast strains and growth conditions

All *S. cerevisiae* strains used in this research (Supplementary Table 2) are isogenic with BY4741 (ref. [53]) and were grown at 30 °C. *TOP1* and *RNH201* open reading frames were deleted using one-step allele replacement using PCR products generated from plasmid templates with selection cassettes (Supplementary Table 2) and primers containing 60-nucleotide homology directly upstream and downstream of the open reading frame. Gene deletions were confirmed by PCR. The 2 bp deletion reporter was inserted at the *AGP1* locus using a PCR product amplified from pTCW12 using primers AGP1-MX6-F and AGP1-MX6-R (Supplementary Table 3). The correct reporter insertion was confirmed by PCR and Sanger sequencing. Growth under selection was on YPD medium (10 g l$^{-1}$ yeast extract, 20 g l$^{-1}$ bactopeptone, 20 g l$^{-1}$ dextrose, 20 g l$^{-1}$ agar) supplemented with hygromycin B (300 mg l$^{-1}$), nourseo-thricin (100 mg l$^{-1}$) and/or G418 (1 g l$^{-1}$), or on synthetic defined medium (6.7 g l$^{-1}$ yeast nitrogen base without amino acids, complete supplement single dropout mixture (Formedium), 20 g l$^{-1}$ dextrose, 20 g l$^{-1}$ agar).

### Fluctuation assays in yeast

Fluctuation assays were performed as previously described[54]. Yeast was grown overnight in YPD medium with hygromycin B (300 mg l$^{-1}$), plated on YPD and grown at 30 °C to obtain individual colonies derived from a single cell without HygroR mutations. For each strain, 16 independent colonies were then used to inoculate 5 ml YPD, and grown for 3 days at 30 °C with shaking at 250 rpm. Cells were pelleted by centrifugation and resuspended in 1 ml of H$_2$O. Undiluted suspensions for each culture were plated (100 µl per plate) on 2 YPD plates supplemented with 1 g l$^{-1}$ G418, with the exception of *rnh201Δ* for which a 10$^{-2}$ dilution was used. Furthermore, each suspension was serially diluted to 10$^{-6}$ of which 100 µl per plate was spread on 2 YPD plates to estimate the total number of viable cells per culture. Plates were incubated at 30 °C for 2–3 days, and colonies counted. Mutation rates were determined in Microsoft Excel 2016 for each individual culture, and an overall rate for each strain calculated using the Lea Coulson method of the median[55]. The number of mutants for each culture was ranked, and those ranked 4th and 13th were used to calculate the rates that define the lower and upper limits of the 95% CI[56]. A single G418-resistant colony for each independent culture was used to determine the spectrum of frame-shift mutations. A 1.3 kb region including HygroR was amplified in two overlapping amplicons (primers S297F and S1113R; S752 and S1658R) using FastStart PCR Master Mix (Roche) and direct colony PCR (5 min at 95 °C; then 35 cycles of 30 s at 95 °C, 30 s at 58 °C and 45 s at 72 °C; and then 45 s at 72 °C). Each amplicon was Sanger sequenced using the primers described in Supplementary Table 3, and analysed using Sequencher v.5.4.6 (Gene Codes Corporation) and/or Mutation Surveyor v.3.30 (SoftGenetics). Mutation rates (per bp) were calculated for 1,032 bp of sequence in which productive frameshift mutations can occur.

### Cell lines

A summary of the human cell lines used in this research is provided in Supplementary Table 4. All cells were grown at 37 °C and 5% CO$_2$, authenticated using STR DNA profiling in the laboratories of origin and shown to be mycoplasma negative through routine testing. HeLa cells (a gift from G. Stewart, University of Birmingham, UK; originally purchased from ATCC) were grown in Dulbecco's Modified Eagle Medium (DMEM; Gibco/Thermo Fisher Scientific) supplemented with 10% fetal bovine serum (FBS), 50 U ml$^{-1}$ penicillin and 50 µg ml$^{-1}$ streptomycin. hTERT RPE-1 cells (a gift from D. Durocher, University of Toronto, Canada; originally purchased from ATCC) were grown in DMEM/F12 medium mixture (Gibco/Thermo Fisher Scientific) supplemented with 10% FBS, 50 U ml$^{-1}$ penicillin and 50 µg ml$^{-1}$ streptomycin. The 2 bp deletion reporter was integrated at the *AAVS1* safe harbour locus in HeLa cells using a published CRISPR–Cas9 targeting protocol[51]. HeLa cells were transfected with pXAT2 and pTCW15 in Opti-MEM reduced-serum medium using Invitrogen Lipofectamine 2000 (Thermo Fisher Scientific). After 48 h cells were replated in medium containing 500 µg ml$^{-1}$ G418 and, after another 48 h and a second round of replating in selective medium, single cells were sorted into 96-well plates using a BD FACSJazz instrument (BD Biosciences). The resulting G418-resistant clones were screened by PCR for reporter integration at the correct locus, retention of integration-free *AAVS1* and Sanger sequencing of resulting PCR products. Single-locus integration was confirmed by fluorescence in situ hybridization (FISH) as previously described[57], using pTCW16 to generate a fluorescently labelled probe. The full reporter sequence of selected clones was checked, with amplification of a 1.9 kb fragment using Prime Star Max PCR Master Mix (Takara Bio) with the primers HygroR_up and PuroR_rev (40 cycles of 10 s at 98 °C, 15 s at 70 °C and 2 min at 72 °C), followed by Sanger sequencing with additional primers (Supplementary Table 3). To generate *RNASEH2A*-KO reporter cells, the selected parental HeLa reporter clone was transfected with pMAR526 and pMAR527 (Supplementary Table 1), using Lipofectamine 2000. Then, 48 h after transfection, single EGFP-expressing cells were sorted into 96-well plates and grown until colonies formed. Initial screening was based on PCR amplification (primers RNASEH2A-ex1F and RNASEH2A-ex1R) of the CRISPR–Cas9-targeted region of *RNASEH2A* with mutations present in selected clones determined by Sanger sequencing. The cellular RNase H2 status was then confirmed by immunoblotting, RNase H2 enzymatic activity assay and alkaline gel electrophoresis to determine ribonucleotide content of genomic DNA (detailed methods are provided below).

### Fluctuation assays in human

Hygromycin resistant HeLa reporter cells (400 µg ml$^{-1}$ hygromycin B) were recovered from frozen stocks in the absence of selection. The next day, 10 wells of a 96-well plate were seeded with 2,000 cells per well for each line. The experiment was performed with the operator blinded to the identity of the cell lines. Cells were cultured under non-selective conditions and replated subsequently in 24-well, 6-well plates and ultimately T75 flasks, in which they were grown to confluence. Cells were then dissociated using Gibco TrypLE (Thermo Fisher Scientific)

and the cells were counted using a Moxi Z automated cell counter. After serial dilution, 1,000 cells were plated into two 10 cm plates for each culture and grown for 14 days to determine plating efficiency. All other cells were plated into two 10 cm plates, and 0.5 µg ml$^{-1}$ puromycin was added after 4 h, with the medium subsequently changed every 2–3 days for 14 days to remove dead cells and maintain a puromycin concentration of 0.5 µg ml$^{-1}$.

To establish mutation spectra, colonies were removed by scraping and then cultured in a 96-well plate. When confluent, cells were lysed with 75 µl DirectPCR Lysis Reagent (Viagen Biotech) and 0.4 mg ml$^{-1}$ PCR-grade Proteinase K (Roche), heating overnight at 55 °C followed by 45 min at 85 °C. Only one sample per independent culture was used for PCR amplification and Sanger sequencing to determine the nature of mutations in the HygroR coding sequence. A 1.24 kb region including HygroR was amplified with Prime Star Max PCR Master Mix (Takara Bio), HygroR_up and H1327R primers (40 cycles of 10 s at 98 °C, 15 s at 70 °C, 2 min at 72 °C). Sanger sequencing was then performed with additional primers (Supplementary Table 3) and mutations identified using Mutation Surveyor v.3.30 (SoftGenetics). All mutants showed double traces of equal height from the point of indel mutations, consistent with the presence of two copies of the reporter in all reporter lines. As FISH indicated the presence of the reporter at a single *AAVS1* locus, we inferred that two copies of the reporter were inserted in tandem at this locus. As a 2 bp deletion or equivalent frameshift mutation in either HygroR copy would bring the associated PuroR coding sequence into the translated reading frame, we corrected mutation rate calculations (per bp) for the presence of two copies.

To determine colony numbers, plates were washed with PBS, fixed with 2% formaldehyde in PBS for 10 min, rinsed with water and colonies were stained with 0.1% crystal violet solution for 10 min. Plates were then washed with water and left to dry before counting colonies. After counting, the experiment was performed in an unblinded manner. Mutation rates were determined for each individual culture in Microsoft Excel 2016, and an overall rate for wild-type and KO strains was calculated using the Lea Coulson method of the median. The number of mutants for each culture was ranked, and appropriate ranks[56] were used to calculate the rates that define the lower and upper limits of the 95% CI.

## Immunoblotting

Whole-cell extracts (WCE) to determine protein levels of RNase H2 subunits by immunoblotting and for RNase H2 activity assays were prepared as previously described[58]. Equal amounts of protein from WCE were separated by SDS–PAGE on 4–12% NuPAGE gels and transferred to PVDF. Membranes were probed in 5% milk (w/v; Marvel Original Dried Skimmed), TBS + 0.2% Tween-20 (v/v) with the following antibodies: sheep anti-RNase H2 (raised against human recombinant RNase H2, 1:1,000)[14]; mouse anti-RNASEH2A G-10 (Santa Cruz Biotechnologies, sc-515475, A1416, 1:1,000); rabbit anti-GAPDH (Abcam, ab9485, 1:2,000, GR3380498-1). For detection, we used rabbit anti-sheep immunoglobulins/HRP (Dako, P04163, 00047199, 1:2,000); goat anti-mouse immunoglobulins/HRP (Dako, P0447, 20039214, 1:10,000); anti-rabbit IgG, HRP-linked antibodies (Cell Signaling Technologies, 7074S, 29, 1:10,000); Amersham ECL Prime Western Blotting Detection Reagent (GE Healthcare Life Sciences) and an ImageQuant LAS4000 device, or IRDye secondary antibodies and an Odyssey CLx Imaging System (LI-COR Biosciences). The uncropped immunoblots are presented in Supplementary Fig. 1.

## RNase H2 activity assays

To assess cellular RNase H2 activity, a FRET-based fluorescence substrate release assay was performed as previously described[14]. In brief, RNase-H2-specific activity was determined by measuring the cleavage of double-stranded DNA substrate containing a single embedded ribonucleotide. Activity against a DNA-only substrate of the same sequence was used to correct for background activity. Substrates were formed by annealing a 3′-fluorescein-labelled oligonucleotide (GATCTGAGCC TGGGaGCT or GATCTGAGCCTGGGAGCT; uppercase DNA, lowercase RNA) to a complementary 5′-DABCYL-labelled DNA oligonucleotide (Eurogentec). The reactions were performed in 100 µl reaction buffer (60 mM KCl, 50 mM Tris–HCl pH 8.0, 10 mM MgCl$_2$, 0.01% BSA, 0.01% Triton X-100) with 250 nM substrate in black 96-well flat-bottomed plates (Costar) at 24 °C. WCE was prepared as described above, protein concentrations determined using the Bio-Rad Bradford Protein Assay and the final protein concentration per reaction was 50 ng µl$^{-1}$. Fluorescence was read (100 ms) every 5 min for up to 90 min using the VICTOR2 1420 multilabel counter (Perkin Elmer), with a 480 nm excitation filter and a 535 nm emission filter. Initial substrate conversion after background subtraction was used to calculate RNase H2 enzyme activity.

## Alkaline gel electrophoresis

To determine the presence of excess genome-embedded ribonucleotides in nuclear DNA, alkaline gel electrophoresis of RNase-H2-treated genomic DNA was performed as previously described[58]. In brief, total nucleic acids were isolated from pellets from around 1 million cells by incubation in ice-cold buffer (20 mM Tris-HCl pH 7.5, 75 mM NaCl, 50 mM EDTA) with 200 µg ml$^{-1}$ proteinase K (Roche) for 10 min on ice, followed by addition of *N*-lauroylsarcosine sodium salt (Sigma-Aldrich) to a final concentration of 1%. Nucleic acids were extracted using phenol–chloroform, then isopropanol-precipitated and dissolved in nuclease-free water. For alkaline gel electrophoresis, 500 ng of total nucleic acids was incubated with 1 pmol of purified recombinant human RNase H2 (isolated as previously described[59]) and 0.25 µg of DNase-free RNase (Roche) for 30 min at 37 °C in 100 µl reaction buffer (60 mM KCl, 50 mM Tris–HCl pH 8.0, 10 mM MgCl$_2$, 0.01% Triton X-100). Nucleic acids were ethanol-precipitated, dissolved in nuclease-free water and 250 ng was separated on 0.7% agarose gels in 50 mM NaOH, 1 mM EDTA. After overnight electrophoresis, the gel was neutralised in 0.7 M Tris–HCl pH 8.0, 1.5 M NaCl and stained with SYBR Gold (Invitrogen). Imaging was performed on a FLA-5100 imaging system (Fujifilm), and densitometry plots were generated using AIDA Image Analyzer v.3.44.035 (Raytest).

## Mutation-accumulation experiment

*TP53*-KO hTERT RPE-1 cells without and with loss-of-function mutations in *RNASEH2A* or *RNASEH2B*, introduced by CRISPR–Cas9 genome editing, a gift from D. Durocher, have previously been described[33]. RNase-H2-proficient (wild type), *RNASEH2A*-KO and *RNASEH2B*-KO cells were single-cell sorted into 96-well plates using the BD FACSJazz instrument (BD Biosciences). Multiple individual clones for each were expanded to confluent T75 flasks for cryopreservation and genomic DNA isolation of these ancestral populations. Moreover, lines were again single-cell sorted into 96-well plates to start the mutation-accumulation experiment. Cultures were expanded by subsequent growth in 24-well, 6-well plates and T75 flasks until confluent (approximately 25 population doublings), and this process of single-cell sorting and expansion was repeated four more times, providing bottlenecks to capture mutations that occurred since the previous sort. From the first to the last single-cell sort, a total of approximately 100 population doublings occurred and the final culture was expanded for cryopreservation and genomic DNA isolation of these end-point populations.

Genomic DNA was isolated using phenol extraction as previously described[58], for alkaline gel electrophoresis and WGS. Library preparations and sequencing were performed by Edinburgh Genomics. Libraries were prepared using Illumina SeqLab specific TruSeq PCRFree High Throughput library preparation kits according to the manufacturer's instructions, with DNA samples sheared to a mean insert size of 450 bp. Libraries were sequenced using paired-end reads on the Illumina HiSeqX instrument using v2.5 chemistry to achieve minimum mean genome-wide sequencing depth of 30× per sample.

## Mouse WGS analysis

*Villin-cre+Trp53fl/flRnaseh2bfl/fl* mice with epithelial-specific deletion of *Trp53* and *Rnaseh2b* on a C57Bl/6J background have been described previously[29]. Animal experiments were conducted with appropriate permission, in accordance with guidelines for animal care of the Christian-Albrechts-University, in agreement with national and international laws and policies. No randomization or blinding was performed. Paired tumour–normal DNA was isolated from small intestinal tumours (*Trp53−/−Rnaseh2b−/−*) and liver tissue (*Trp53+/+Rnaseh2b+/+*) from female mice (aged 52 weeks), using the Qiagen DNeasy Blood & Tissue Kit. Library preparations and sequencing were performed by Edinburgh Genomics using Illumina DNA PCR-Free Library Prep according to the manufacturer's instructions. Paired-end sequencing was performed by Edinburgh Genomics on a NovaSeq 6000 using v1.5 chemistry. Mean genome-wide sequencing depth of at least 30× for liver samples and 60× for tumour samples was obtained.

## *S. cerevisiae* WGS analysis

WGS SRA files for *rnh201Δ pol2-M644G S. cerevisiae*[25] from the NCBI Sequence Read Archive (SRA) were converted to FASTQ files using SRA Toolkit v.2.5.4-1 (SRA Toolkit Development Team; http://ncbi.github.io/sra-tools/). FASTQ reads were aligned to the GSE56939_L03_ref_v2 reference genome[60] (Supplementary Table 5) and sorted BAM files were created using BWA-MEM (v.0.7.12)[61], and deduplicated with SAMBLASTER (v.0.1.22)[62]. To select high-quality indel variants, GATK (v.3.6-0) Haplotype Caller (without base quality score recalibration)[63] variant calling was performed with 'Hard Filters' (--filterExpression "QD < 2.0 || FS > 200.0 || ReadPosRankSum < −20.0"). Filtering for strain-specific variants was performed as previously described[60], with minor modifications. The filters were as follows: (1) eliminate variants shared with an ancestral clone; (2) required ≥20 reads for variant allele in descendent; (3) exclusion of repetitive sequences as defined in ref. [60]; and (4) reference/variant depth ratio 0.4–0.6, <0.4 if homozygous variant allele.

## RPE-1 WGS analysis

FASTQ files were converted to unaligned BAM format and Illumina adaptors were marked using GATK (v.4.1.9.0) FastqToSam and MarkIlluminaAdapters tools[64]. Reads were aligned to the human genome (hg38, including alt, decoy and HLA sequences) using BWA-MEM (v.0.7.16)[61] and read metadata were merged using GATK's MergeBamAlignment tool. PCR and optical duplicate marking and base quality score recalibration were performed using GATK. Variants from NCBI dbSNP build 151 were used as known sites for base quality score recalibration. Post-processed alignments were genotyped using Mutect2, Strelka2, Platypus and SvABA using somatic calling models for each pair of ancestral and end-point cultures, as described below.

## Mouse WGS analysis

FASTQ processing and alignment were performed as for RPE-1 WGS analysis, using the GRCm38 mouse genome reference and known variant sites from the Mouse Genomes Project[65] (REL-1807-SNPs_Indels) for base quality score recalibration. Somatic variant calling of post-processed alignments was performed using Mutect2, Strelka2, Platypus and SvABA for each tumour-liver pair, as described below. Somalier v.0.2.12 (https://github.com/brentp/somalier) was used to confirm each paired tumour and liver sample originated from the same animal.

## Human ethics approval

Data generated from Genomics England 100,000 genomes and ICGC-CLL studies were analysed. In these respective studies, informed consent for participation was obtained. Ethical approval for Genomics England 100,000 genomes project: East of England and South Cambridge Research Ethics Committee; CLL-ICGC: International Cancer Genome Consortium (ICGC) guidelines from the ICGC Ethics and Policy committee were followed and the study was approved by the Research Ethics Committee of the Hospital Clínic of Barcelona.

## CLL WGS analysis

**Genomics England.** CLL tumour–normal pairs (*n* = 198) were processed as part of the 100,000 Genomes Project (pilot and main programme v8). Samples were sequenced using the Illumina HiSeq X System with 150 bp paired-end reads at a minimum of 75× coverage for tumours and 30× coverage for germline samples. Reads were mapped to GRCh38 using ISAAC aligner (v.03.16.02.19)[66]. Single-nucleotide variants (SNVs) and indels were called using Strelka v.2.4.7 using somatic calling mode. Structural and copy number variants were called using Manta (v.0.28.0) and Canvas (v.1.3.1)[67], respectively. Samples with a tumour purity estimate from Canvas of less than 50% were excluded from analysis. *RNASEH2B* copy number was determined using a combination of Canvas, Manta, read depth counts with samtools (v.1.9) and confirmed by manual inspection using IGV (v.2.5.0)[68].

**ICGC.** WGS from the ICGC-CLL cohort[35] (*n* = 150) was reanalysed. Raw reads were mapped to the human reference genome (GRCh37) using BWA-MEM (v.0.7.15)[61]. BAM files were generated, sorted and indexed, and optical or PCR duplicates were flagged using biobambam2 (https://gitlab.com/german.tischler/biobambam2, v.2.0.65). Copy-number alterations were called from WGS data using Battenberg (cgpBattenberg, v.3.2.2)[69], ASCAT (ascatNgs, v.4.1.0)[70], and Genome-wide Human SNP Array 6.0 (Thermo Fisher Scientific) data[35] reanalysed using Nexus 9.0 Biodiscovery software (Biodiscovery). *RNASEH2B* copy number was established by combining the three analyses and manual review with IGV.

## Colorectal cancer WGS analysis

Irinotecan-treated (*n* = 39) and irinotecan-untreated (*n* = 78) colorectal cancers from the 100,000 Genomes Project Colorectal Cancer Domain were 1:2 matched using a multivariate greedy matching algorithm without replacement, implemented in the Matching R-package[71]. Matching was conducted considering sex, age at sampling, whether a primary tumour or metastasis had been sequenced, microsatellite instability status, and whether the individual had previously received radiotherapy, oxaliplatin, capecitabine or fluorouracil treatment.

## Somatic variant calling

Somatic variant calling was performed in parallel using four distinct methods: Mutect2 (as part of GATK v.4.1.9.0)[72,73], Strelka2 (v.2.1.9.10)[74], SvABA (v.1.1.3)[75] and Platypus (v.0.8.1)[76]. High-confidence indel calls were defined as the intersected output of these four tools, where variants passed all filters for ≥3 of 4 callers. The intersection was performed using the bcftools (v.1.10.2)[77] isec function after normalizing variant calls and left-aligning ambiguous alignment gaps using the bcftools norm function. For Platypus (v.0.8.1)[76], joint calling all samples in each cohort was performed before filtering for somatic variants; the other variant callers were run in paired tumour–normal mode. For the RPE-1 mutation-accumulation experiment the end-point and ancestral cultures were defined as 'tumour' and 'normal' samples, respectively. Variant filtering strategies were optimized to both available information on segregating genetic variation for humans and mice, and the functionality of each calling method as detailed below.

**Mutect2.** Unfiltered genotypes for all normal samples were combined to filter germline variants. Somatic calls were obtained using GATK's FilterMutectCalls command. Human polymorphism data and allele frequencies from gnomAD[78] were provided to Mutect2 for the filtering of germline variants.

**SvABA.** Germline indel and structural variants were filtered using the --dbsnp-vcf and --germline-sv-database options. Mouse indels

were obtained from Mouse Genomes Project version 5 SNP (ftp://ftp-mouse.sanger.ac.uk/REL-1505-SNPs_Indels/mgp.v5.merged.indels.dbSNP142.normed.vcf.gz); structural variants from SV release version 5 (ftp://ftp-mouse.sanger.ac.uk/REL-1606-SV/mgpv5.SV_insertions.bed.gz and ftp://ftp-mouse.sanger.ac.uk/REL-1606-SV/mgpv5.SV_deletions.bed.gz). Human indels were extracted from NCBI dbSNP build 151 and common structural variants from dbVAR (https://hgdownload.soe.ucsc.edu/gbdb/hg38/bbi/dbVar/).

**Strelka2.** Candidate small indels for each pair were first generated by Manta (v.1.6.0)[79] in somatic calling mode. Strelka2 was then executed in somatic calling mode for each pair with Manta's candidate small indels output provided to the --indelCandidates option.

**Platypus.** Germline variants were filtered on the basis of any normal sample with ≥2 variant allele reads. Somatic variant calls for each sample pair were retained if tumour/end-point sample > 2 variant reads; site depth > 9; and normal sample read depth ≥ 20, <2 variant reads. Moreover, a >10× ratio of tumour to normal for variant/total depth was required.

For Genomics England CLL tumour–normal pairs, pre-existing Strelka2 calls from the 100,000 Genomes Project pipeline were used, while variant calling with Mutect2, Platypus and SvABA was performed as above. Colorectal cancer tumour–normal pairs from Genomics England were processed as for Genomics England CLL but without Mutect2 analysis. For ICGC CLL, somatic indels were called using Mutect2 (GATK v.4.0.2.0)[72,73], Strelka2 (v.2.8.2)[74], SvABA (v.1.1.0)[75] and Platypus (v.0.8.1)[76]. Candidate small indels generated by Manta (v.1.2)[79] were used as input for Strelka2. Mutect2, Strelka2 and SvABA were run in paired tumour–normal mode. somaticMutationDetector.py (https://github.com/andyrimmer/Platypus/tree/master/extensions/Cancer) was used to identify somatic indels called by Platypus with a minimum posterior of 1. SNVs called by Platypus were considered to be somatic if they had at least 2 alternative reads in the tumour, fewer than 2 alternative reads in the normal, a minimum tumour VAF of 10× the control VAF, and a minimum depth of 10.

### Germline mutation analysis
De novo WGS variants were downloaded from the Gene4Denovo database (Supplementary Table 5). Reference assembly conversion errors were removed by discarding variants for which the reference allele did not match the genome reference at the given position or for which the variant position was greater than the length of the reference chromosome. Furthermore, individuals with total de novo variants below the 10th ($n = 33$) or above the 90th ($n = 140$) percentile were excluded. For germline gene expression we used predefined expression groups[80] based on Ensembl release 90 annotation (ftp://ftp.ensembl.org/pub/release-90/gtf/homo_sapiens/Homo_sapiens.GRCh38.90.gtf.gz). Initially stratified as nine expression groups from 1 (unexpressed) to 9 (high), we collapsed them into a smaller set of unexpressed (1), low (2, 3, 4), mid (5, 6, 7) and high (8, 9). The annotations were converted to GRCh37 coordinates using liftover (kent source v.417). Genomic segments overlapping multiple distinct expression groups, due to overlapping genes, were assigned to the higher of those expression groups. For each expression group, we summed the count ($c$) of de novo indels contained within the genomic span of those genes. This was converted to rate estimates by dividing by the union genomic span ($g$ nucleotides) of genes in that expression group, and adjusting for the number of mutated genomes considered ($n$); rate = $c/(gn)$. To obtain 95% CIs, gene selection was bootstrapped (sampled to an identical number with replacement) 100 times and the 0.025 and 0.975 quantiles of the bootstrapped rate calculation taken as the 95% CI.

### ICGC pan-cancer expression analysis
The ICGC PCAWG somatic mutations[50] (https://dcc.icgc.org/api/v1/download?fn=/PCAWG/consensus_snv_indel/final_consensus_passonly.snv_mnv_indel.icgc.public.maf.gz) and ICGC PCAWG 'baseline' gene expression data[50] were obtained (ArrayExpress, https://www.ebi.ac.uk/arrayexpress/experiments/E-MTAB-5200/). Genomic annotation of gene extents on the GRCh37 reference genome match the Ensembl version 75 annotation (http://ftp.ensembl.org/pub/release-75/gtf/homo_sapiens/Homo_sapiens.GRCh37.75.gtf.gz) of the ICGC gene expression calls. Mean, median and maximal gene expression (transcripts per million (TPM)) were calculated for each gene across the 76 ICGC baseline gene expression tissues/samples. Only genes annotated on the main autosomal chromosomes, 1 to 22, and the X chromosome were considered. Overlapping genes were removed, retaining only the most abundantly (highest median, then mean in the case of ties) expressed genes from overlapping pairs. This filtering was applied hierarchically, starting with the most abundant. Following ref. [81], genes with housekeeping-like expression were defined as those with maximal expression of less than ten times median expression. Housekeeping-like genes were decile-binned into expression groups on the basis of median expression. Mutations were stratified by type (1 bp deletion, 2–5 bp deletion) or by the 'TN*T' motif defined below and counted by intersection with the annotated genomic extents of genes in each expression group.

For the analysis of tissue-biased gene expression, the 76 ICGC baseline samples were grouped by annotated tissue (such as breast, prostate, kidneys, liver) and matched where possible to the tissue of origin for ICGC cancer types. For each tissue, the median expression (in TPM) of each gene was calculated for (1) within-tissue samples and (2) for all other samples. The 90th quantile of gene expression (q90, top 10%) within a tissue was set as a threshold for high level expression. Genes with high expression in a tissue (1) but a median expression of less than q90*0.1 in the other tissues (2) were considered to be highly expressed but tissue restricted (HETR). For the set of HETR genes from a tissue, we counted the number of 2–5 bp deletions within the annotated genomic extent of the HETR genes in a cancer type of interest. We similarly counted 2–5 bp deletions in all other genes for that cancer type, and counted both the HETR and non-HETR 2–5 bp deletions from all other cancer types within the ICGC cohort. For each cancer–tissue pair, this provided four sets of counts, analysed using two-tailed Fisher's exact tests using the R function fisher.test. A positive odds ratio indicated enrichment of 2–5 bp deletions in the HETR genes, compared with a background of the remainder of the ICGC cohort in which HETR genes are not highly expressed. For each cancer type considered, this test was repeated for each tissue type ($n = 17$). Analyses were carried out for eight of the ICGC cohort cancer types that met the combined criteria of having a well-matched and known tissue of origin among the ICGC baseline samples, and requiring the cancer type cohort to have at least $n = 2,500$ 2–5 bp deletions in aggregate. This represents $n = 17 \times 8 = 136$ statistical tests, adjusted by Bonferroni correction. Odds ratios ($r$) for mutation depletion were transformed to their reciprocal ($1/r$) for display purposes.

### ICGC pan-cancer TOP1-seq analysis
Data corresponding to two replicates of TOP1-seq, a modified ChIP–seq technique to immunoprecipitate only catalytically engaged TOP1 (ref. [38]), were downloaded from the NCBI Gene Expression Omnibus database (accession code GSE57628, samples GSM1385717 and GSM1385718). Autosomal chromosomes 1 to 22 and the X chromosome were divided into 1 kb bins and, for each bin, the amount of mappable sequence was determined using Umap's regions mappable using 36-mers[82] to approximate the read length of the TOP1-seq data. For each 1 kb window, the TOP1-seq signal within mappable regions was summed for each replicate and the mean signal was calculated. This mean was divided by the amount of mappable sequence to calculate the TOP1-seq signal per bp and each 1 kb window was then assigned to decile bins using this value.

Somatic deletion calls from ID4-positive PCAWG samples (as defined in https://dcc.icgc.org/api/v1/download?fn=/PCAWG/mutational_signatures/Signatures_in_Samples/SP_Signatures_in_Samples/

PCAWG_SigProfiler_ID_signatures_in_samples.csv) were counted within the same 36-mer mappable regions for each 1 kb window and either stratified by type (1 bp deletion, 2–5 bp deletion) or by the TN*T motif defined below. Relative rates of deletions in each category were calculated relative to the first TOP1-seq signal decile.

## Mutational signatures

De novo extraction and decomposition of mutational signatures was performed in Python v.3.8.5 using SigProfilerExtractor (v.1.1.0)[5], along with SigprofilerMatrixGenerator (v.1.1.14/1.1.15)[83] and SigprofilerPlotting (v.1.1.27). The recommended default settings (including 500 NMF replicates) were applied (https://github.com/AlexandrovLab/SigProfilerExtractor). Subtraction of mutations in RPE-1 wild-type cells from those detected in RNase H2 null cells was performed as follows. The average number of indels per line for each of the 83 categories was determined for the three wild-type lines. Counts per category for AKO and BKO lines were subtracted using these averages, with negative values set to 0. SigProfilerExtractor was then performed on the resulting WT-subtracted AKO and BKO ID-83 matrices for both de novo signature detection and decomposition analysis.

## Indel sequence context analysis

WGS indels were categorized on the basis of repeat sequence context. Genome-wide occurrence of short repeats and regions of MH were identified and filtered to include only the mappable genome, defined by Umap's regions mappable using 100-mers[82]. For both WGS-identified indel variants and genome-wide occurrence, scoring of 2 bp deletions compliant with the TNT motif at MH/SSTR sites required the deleted bases to match the sequence NT with a T immediately 5' of the deleted dinucleotides. More generally, for varying sized deletions these were considered to fit a TN*T motif if the deletion lay within an SSTR or region of MH containing the motif $TN_{(d-1)}T$ where $d$ is the length of the deletion. Genome-wide occurrences were estimated from 100,000 randomly generated deletions of given lengths within the mappable genome. For SSTRs and MH regions, all regions containing the respective motifs $(TN_{(r-1)})_n$ or $TN_{(r-1)}T$ were identified (where $r$ is the length of the repeat unit and $n > 1$), and the fraction of SSTR/MH sequence containing TNT motifs was determined against total SSTR/MH sequence in the mappable genome.

To derive a null expectation for de novo deletions matching the TNT, TNNT and TNNNT motif for 2, 3 and 4 bp deletions, respectively, deletions at repeats from the Gene4Denovo database were first classified by deletion length, repeat type (STR or MH) and repeat length. Bootstrap samples of corresponding repeats from the genome were generated with 1,000 replicates. That is, for each deletion category an equal number of repeats of matching repeat type, repeat unit length and total repeat length were randomly drawn from the genome for each bootstrap sample.

## Sequence logos

Genomic sequences containing 2 bp deletions were reversed and complemented when the deleted dinucleotide contained an adenosine (A), except when the dinucleotide was AT or TA. For SNMH and STR deletions, the position of the deleted dinucleotide cannot be unequivocally assigned and, therefore, the deleted sequence was right-aligned in the repeat/MH region, either to the most 3' T, where present, or otherwise to the limit of the repeat/MH region. Sequences were converted to bit score matrices and logos were drawn using Logomaker (v.0.8)[84].

## Embedded ribonucleotide sequence context analysis

EmRiboSeq data from *rnh201Δ* yeast prepared during mid-log phase growth[85] were obtained (Supplementary Table 5) and aligned to the sacCer3 reference genome as previously described to identify the genomic coordinates of genome-embedded ribonucleotides[86]. The Bedtools (v.2.30.0)[87] utilities groupby, slop and getfasta were used to extract and count the sequence context of genome-embedded ribonucleotides

with downstream analysis and plotting implemented in R (v.4.0.5). Genome sequence composition-adjusted relative rates were calculated as previously described[32] such that, under the null expectation of no sequence bias in ribonucleotide incorporation, all sequence contexts have an expected relative rate of $1/n$ where $n$ is the number of contexts considered.

## Statistical methods

Statistical testing was performed using GraphPad Prism v.9.1.1, Python v.3.8.5 or R v.3.3.1. Two-sided non-parametric Mann–Whitney $U$-tests were performed for quantitative measurements; multiple testing correction, FDR set at 0.05; and, for categorical data, Fisher's exact tests were performed in Python using stats.fisher_exact from scipy v.1.6.3. Calculation of cosine similarities was performed as follows. Mutations for each strain were converted into a vector, with ordered values representing different mutation categories as a proportion of total mutations. These were then compared in a pairwise manner. Given two vectors **A** and **B**, the cosine similarity ($\cos(\theta)$) was calculated as:

$$\cos(\theta) = \frac{\sum_{i=1}^{n} \mathbf{A}_i \mathbf{B}_i}{\sqrt{\sum_{i=1}^{n}(\mathbf{A}_i)^2} \sqrt{\sum_{i=1}^{n}(\mathbf{B}_i)^2}}$$

Hierachical clustering used the hclust function of R (v.4.1.0) with complete linkage clustering of pairwise cosine distances (1 − cosine similarity) between ID-83 mutation spectra, with 41 categories of productive reporter frameshift mutations. For bootstrap support, $n = 1,000$ bootstrap datasets were generated by sampling with replacement the mutations observed with a strain, for each strain, and then calculating the cosine distance and hierarchical clustering for each bootstrap dataset. Reported bootstrap scores are the percentage of bootstrap replicates hierarchical clustering of which supports the clustering to the right of the indicated position.

To test the significance of cosine similarities, we used a null model based on the Dirichlet-multinomial distribution. In brief, when comparing two mutation count vectors, with total mutations $m_1$ and $m_2$, over $n$ mutation classes, we constructed a distribution of cosine values by comparing 10,000 simulated pairs of random vectors generated as follows. For each simulated pair, we sampled from a Dirichlet-multinomial distribution with the concentration parameters as a vector of ones of dimension $n$, and number of trials as $m_1$ for the first vector in the pair, and $m_2$ for the second vector. The null distribution was obtained by computing the cosine similarity of the 10,000 pairs of mutation count vectors.

## Reporting summary

Further information on research design is available in the Nature Research Reporting Summary linked to this paper.

## Data availability

RPE-1 mutation-accumulation experiment and mouse tumour WGS data are available from the European Nucleotide Archive under accession number PRJEB48753. All other data were previously published and the sources are cited in Supplementary Table 5.

## Code availability

Code documented in the Methods is available online (https://git.ecdf.ed.ac.uk/ID-TOP1).

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

**Acknowledgements** We thank S. Jinks-Robertson for suggesting the traffic light reporter approach; H. Klein for guidance on fluctuation assays; R. van Boxtel for sharing sequencing data for MLH1-KO organoids; A. Bretherick, O. B. Reina and G. Kudla for advice on HygroR re-coding; staff at the IGC core services (L. Murphy, C. Nicol, C. Warnock, E. Freyer, S. Brown and J. Joseph), C. Logan, A. Fluteau, A. Robertson and the staff at Edinburgh Genomics for technical assistance; staff at Liverpool CLL Biobank (funded by Blood Cancer UK) for samples used to generate GEL WGS data; A. Ewing, C.-A. Martin, N. Hastie and W. Bickmore for discussions. Funding for this work: UK Medical Research Council Human Genetics Unit core grants (MC_UU_00007/5 to A.P.J., MC_UU_00007/11 to M.S.T.); Edinburgh Clinical Academic Track PhD programme (Wellcome Trust 204802/Z/16/Z) to T.C.W.; 2021 AACR-Amgen Fellowship in Clinical/Translational Cancer Research (grant number 21-40-11-NADE) to F.N.; a CRUK Brain Tumour Centre of Excellence Award (C157/A27589) to M.D.N.; EKFS research grant (2019_A09), Wilhelm Sander-Stiftung (2019.046.1) to K.A., CRUK programme grant (C20807/A2864) to T.S.; La Caixa Foundation (CLLEvolution-LCF/PR/HR17/52150017, Health Research 2017 Program HR17-00221) to E.C.; E.C. is an Academia Researcher of the Institució Catalana de Recerca i Estudis Avançats of the Generalitat de Catalunya. Edinburgh Genomics is partly supported by NERC (R8/H10/56), MRC (MR/K001744/1) and BBSRC (BB/J004243/1). This research was made possible through access to the data and findings generated by the 100,000 Genomes Project. The 100,000 Genomes Project is managed by Genomics England Limited (a wholly owned company of the Department of Health and Social Care). The 100,000 Genomes Project is funded by the National Institute for Health Research and NHS England. The Wellcome Trust, Cancer Research UK and the Medical Research Council have also funded research infrastructure. The 100,000 Genomes Project uses data provided by patients and collected by the National Health Service as part of their care and support.

**Author contributions** M.A.M.R., T.C.W., M.S.T. and A.P.J. conceived the project and designed the experiments. T.C.W. and M.A.M.R., with help from P.C., performed fluctuation assays and sequencing experiments. M.A.M.R., with help from P.C., performed the RPE-1 mutation-accumulation experiment. S.B. performed FISH experiments. M.A.M.R., T.C.W. and D.O.R.S. performed all of the other molecular biology experiments. H.X. and K.A. provided mouse tumour and control tissue samples. D.A.P., T.C.W., M.D.N. and M.S.T. designed and implemented computational analyses. D.A.P., T.C.W. and M.S.T. analysed yeast, mouse, RPE-1 and Gene4Denovo WGS data. D.A.P. and M.S.T. performed pan-cancer analyses. The Genomics England Research Consortium, K.R. and A.S. provided CLL WGS data. A.J.C. provided CRC data. D.A.P., F.N., R.L.H., R.R. and C.P. analysed CLL data. D.A.P. analysed CRC data. M.A.M.R., C.P., T.S., E.C., M.S.T. and A.P.J. supervised the work. T.C.W., F.N., E.C., T.S., M.S.T. and A.P.J. funded the work. M.A.M.R. and A.P.J. wrote the manuscript. All of the authors had the opportunity to edit the manuscript. All of the authors approved the final manuscript.

**Competing interests** The authors declare no competing interests.

**Additional information**
**Correspondence and requests for materials** should be addressed to Martin A. M. Reijns, Martin S. Taylor or Andrew P. Jackson.

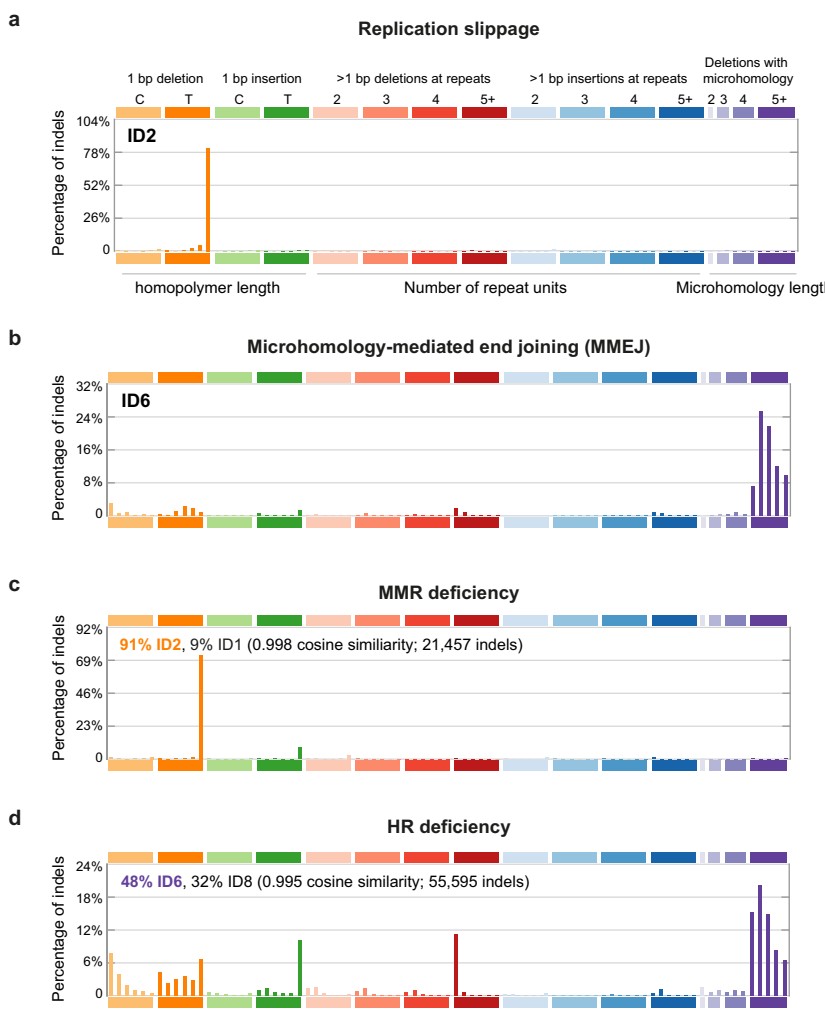

**Extended Data Fig. 1 | ID4 is distinct from small deletion signatures of known aetiology. a**, **b**, The mechanistic basis for many COSMIC indel signatures is unknown, with only 9 out of 18 having a proposed aetiology. ID2 (**a**) is attributed to DNA polymerase slippage[88,89] and ID6 (**b**) to microhomology mediated end-joining (MMEJ) activity, associated with HR deficiency[5,90]. **c**, **d**, Mechanism for these signatures supported by: impaired MMR promoting replication slippage mutagenesis in MLH1[−/−] colonic organoids resulting in ID2 (and ID1) signatures (**c**); ID6 contributing substantially (along with ID8) to the indel signature in ovarian cancer, in which HR deficiency is common (**d**). Analysis of data from[91] in **c**; data for 73 ovarian adenocarcinomas with ID6 contribution from ICGC[5,50] in **d**.

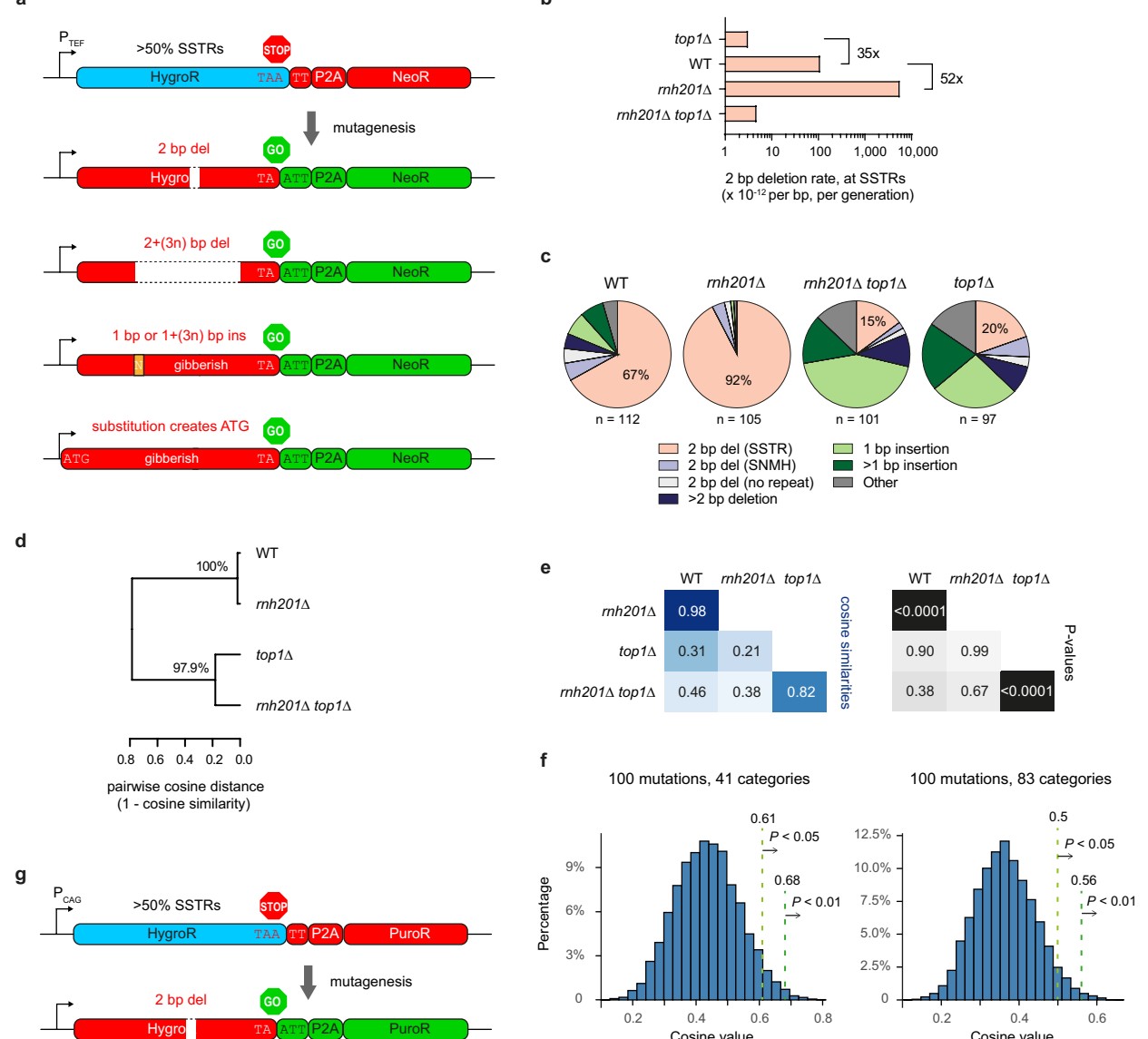

**Extended Data Fig. 2 | Yeast and human frameshift mutation reporters detect indels at tandem repeats. a**, Yeast reporter. Synonymous substitutions were made in the hygromycin resistance gene (HygroR), such that it contained many short 2 bp tandem repeats (SSTRs). Expression from the TEF promoter ($P_{TEF}$) ensures a constitutive high level of transcription. Mutations within HygroR that result in a frameshift simultaneously put the HygroR coding sequence out of frame and the downstream neomycin resistance (NeoR) sequence in frame, allowing antibiotic selection of cells with such mutations. **b**, Top1-dependent 2 bp SSTR deletions occur in both WT and *rnh201Δ* (RNase H2 null) yeast, with the highest mutation rate for *rnh201Δ* (related to Fig. 1d). **c**–**e**, WT and *rnh201Δ* have similar spectra, and differ from *top1Δ* strains. Mutation spectra of neomycin resistant colonies. n, number of independent colonies sequenced. Other: complex indels, missense mutations or mutation not characterised (**c**). Tree for pairwise clustering with percent bootstrap support to the right of the indicated position, based on cosine scores calculated for mutation spectra (Fig. 1e) of the 41 mutation categories that give

productive reporter frameshift mutations (**d**). Matrix of pairwise cosine similarities and P-values between reporter mutation spectra in different yeast strains. Darker blue indicates greater similarity; darker grey greater significance. Test statistic is the cosine similarity value for 41 mutation categories and the null hypothesis is that that the cosine value will be distributed according to the Dirichlet-multinomial model, as described in Methods. The test is one-sided and no adjustments were made for multiple comparisons (**e**). **f**, Null distribution for cosine pairwise vector comparisons for 41 and 83 mutation categories. Plots, cosine values for 10,000 randomly generated pairs of vectors of mutation spectra. Each vector contained 100 randomly assigned mutations (see Methods for further details). Cosine value thresholds indicated for $P < 0.05$ and $P < 0.01$. **g**, The human reporter is expressed from the ubiquitous CAG promoter ($P_{CAG}$), and NeoR is replaced with the puromycin resistance gene (PuroR) to allow more rapid antibiotic selection in mammalian cell culture.

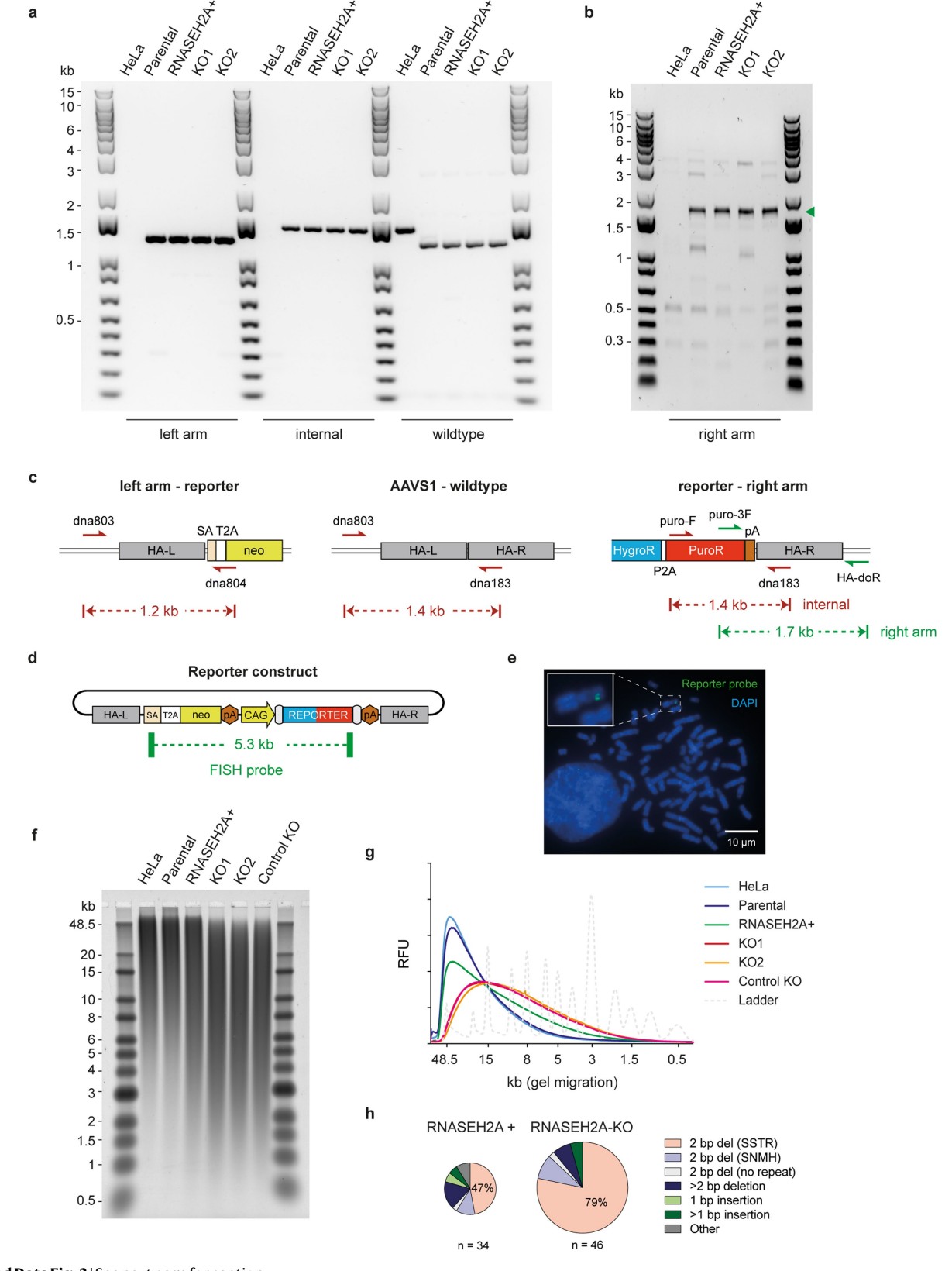

**Extended Data Fig. 3** | See next page for caption.

**Extended Data Fig. 3 | Validation and characterisation of *RNASEH2A+* and KO HeLa reporter cells. a–c,** Reporter integration at the *AAVS1* locus and retention of a reporter-free locus with a 200 bp deletion at the target site was confirmed by PCR and Sanger sequencing. Green arrow head, specific PCR product. Representative of at least 2 independent experiments. **d, e,** FISH shows integration of the reporter (**d**) at a single *AAVS1* locus (**e**). Representative image of approximately one hundred mitotic chromosome spreads in 3 independent experiments. SA, splice acceptor; T2A, self-cleaving peptide; pA, polyadenylation site; also see Fig. 2a. **f, g,** Alkaline gel electrophoresis of RNase H2 treated genomic DNA (**f**) shows a small increase in fragmentation for the *RNASEH2A+* control clone and a more substantial increase in two independent *RNASEH2A*-KO clones (representative of 4 independent experiments), indicating the presence of more genome-embedded ribonucleotides compared to HeLa and parental reporter cells (**g**). "Control KO" cells were reported previously[33,58]. RFU, relative fluorescence units. **h,** 2 bp SSTR deletions are frequent in both *RNASEH2A+* and KO cells. Mutation spectra, quantitation of indel type. Relative area of pie charts scaled to mutation rate. n, number of colonies sequenced from independent cultures. Other: complex indels or missense mutations.

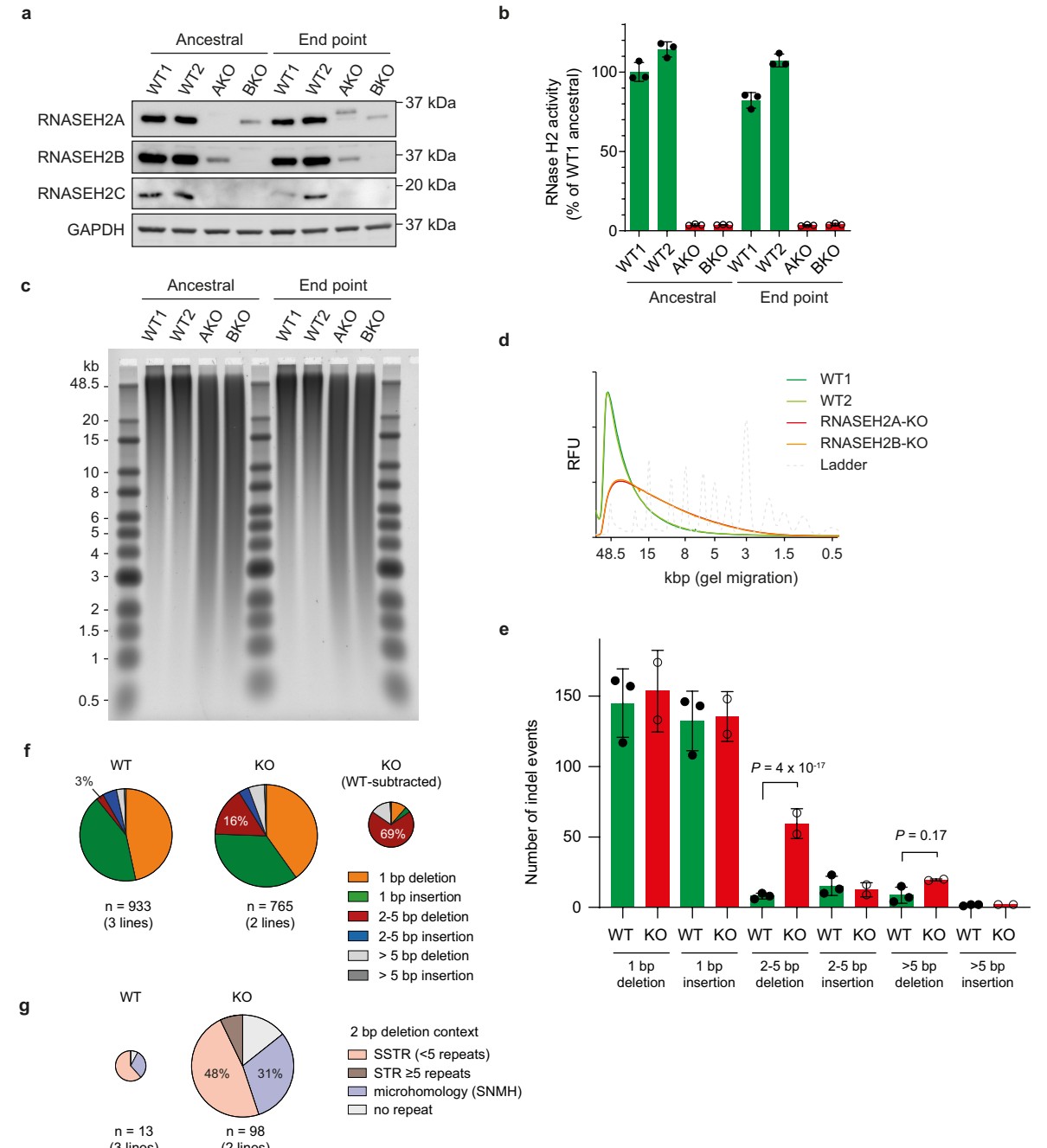

**Extended Data Fig. 4 | RPE1 RNase H2 null cells accumulate embedded ribonucleotides and 2-5 bp deletions across the genome. a, b,** *RNASEH2A* and *RNASEH2B* KO cells (AKO, BKO, respectively) have substantially reduced cellular levels of RNase H2 subunits (**a**) and are deficient for RNase H2 enzyme activity (**b**) at the outset (ancestral) and at the end of the mutation accumulation experiment (end point). Individual data points, *n* = 3 technical replicates; mean ± s.d. For gel source data, see Supplementary Fig. 1. **c, d,** Alkaline gel electrophoresis of RNase H2 treated genomic DNA (**c**) shows a substantial increase in fragmentation for *RNASEH2A* and *RNASEH2B* KO clones (representative of 3 independent experiments), indicating the presence of more genome-embedded ribonucleotides compared to two WT control clones (**d**). Densitometry plots of **c**. RFU, relative fluorescence units. As RNase H2

deficiency activates the p53 pathway[14,92], experiments were performed in a *TP53* knockout background. **e,** Only 2–5 bp deletions are significantly increased in RNase H2 null cells. Data points for acquired indel mutations in individual cell lines after 100 population doublings. Individual data points, indel counts per cell line; mean ± s.d.; P-values for two-sided Fisher's exact test between WT (pooled counts from *n* = 3 independent clones) and KO (*n* = 2 independent clones) for one indel type vs all other indel types, after Bonferroni correction. **f,** Proportions of acquired indels in WT and KO RPE cells. After correction for indels occurring in WT, 69% of indels in RNase H2 null cells are 2–5 bp deletions. n, total indel counts. **g,** Quantification of 2 bp deletions by context. n, total number of 2 bp deletions. For **f, g,** chart areas scaled to mutation counts per line.

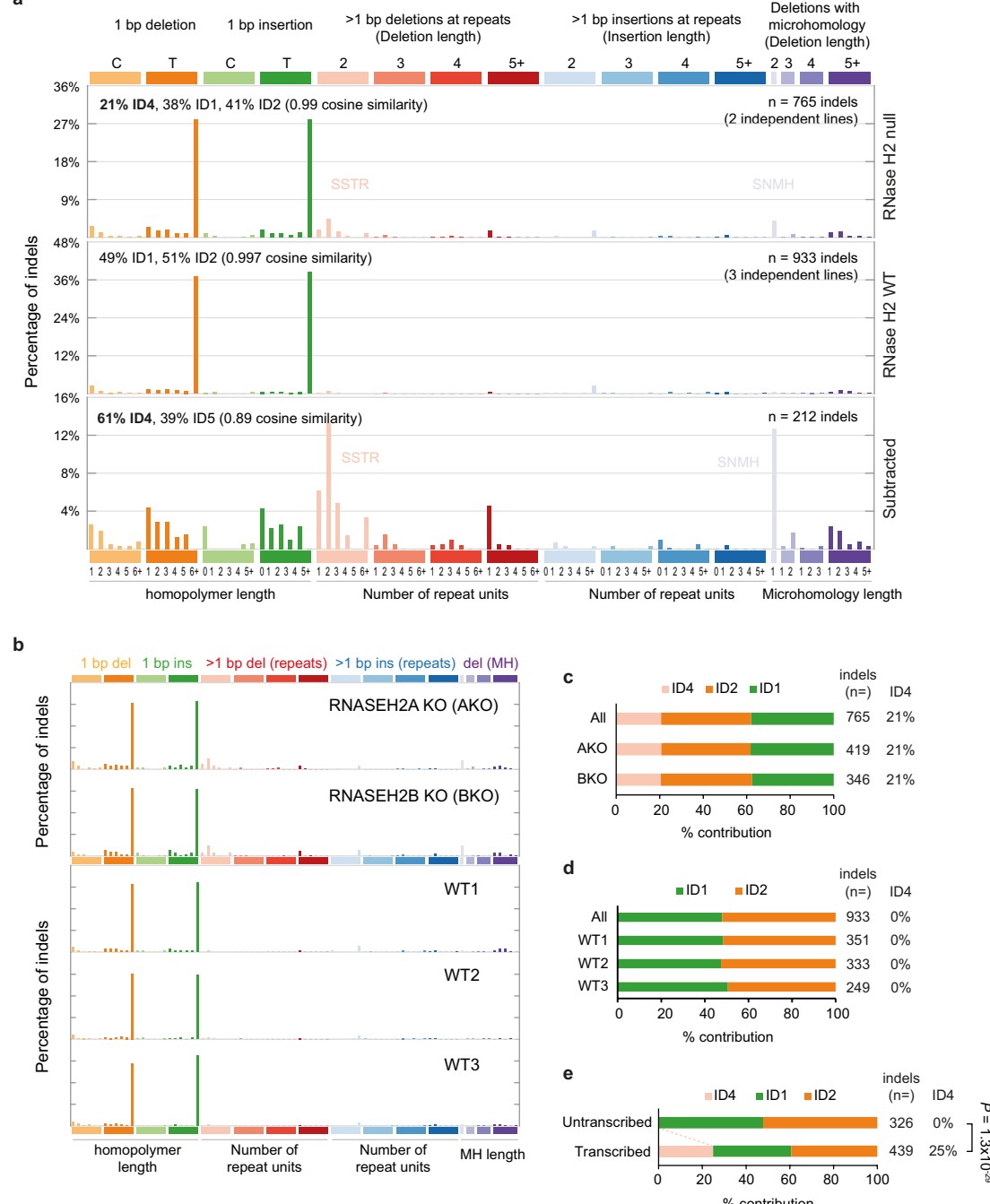

**Extended Data Fig. 5 | ID4 occurs in RNase H2 null RPE1 cells, particularly in transcribed regions. a–d,** Mutational spectra detected by WGS after 100 population doublings in RPE1 cells demonstrates that SSTR and SNMH/MH deletions are enriched in RNase H2 null cells. Spectra for combined RNase H2 null and wildtype cell lines (**a**), and individual cell lines (**b**). Mutational signature analysis confirms ID4 contribution in RNase H2 null (**c**), but not WT cells (**d**). **e,** In RNase H2 null cells, ID4 contributes significantly more to indel mutations in transcribed genomic regions ($P = 1.3 \times 10^{-29}$). Two-sided Fisher's exact test, ID4 indels vs other indels.

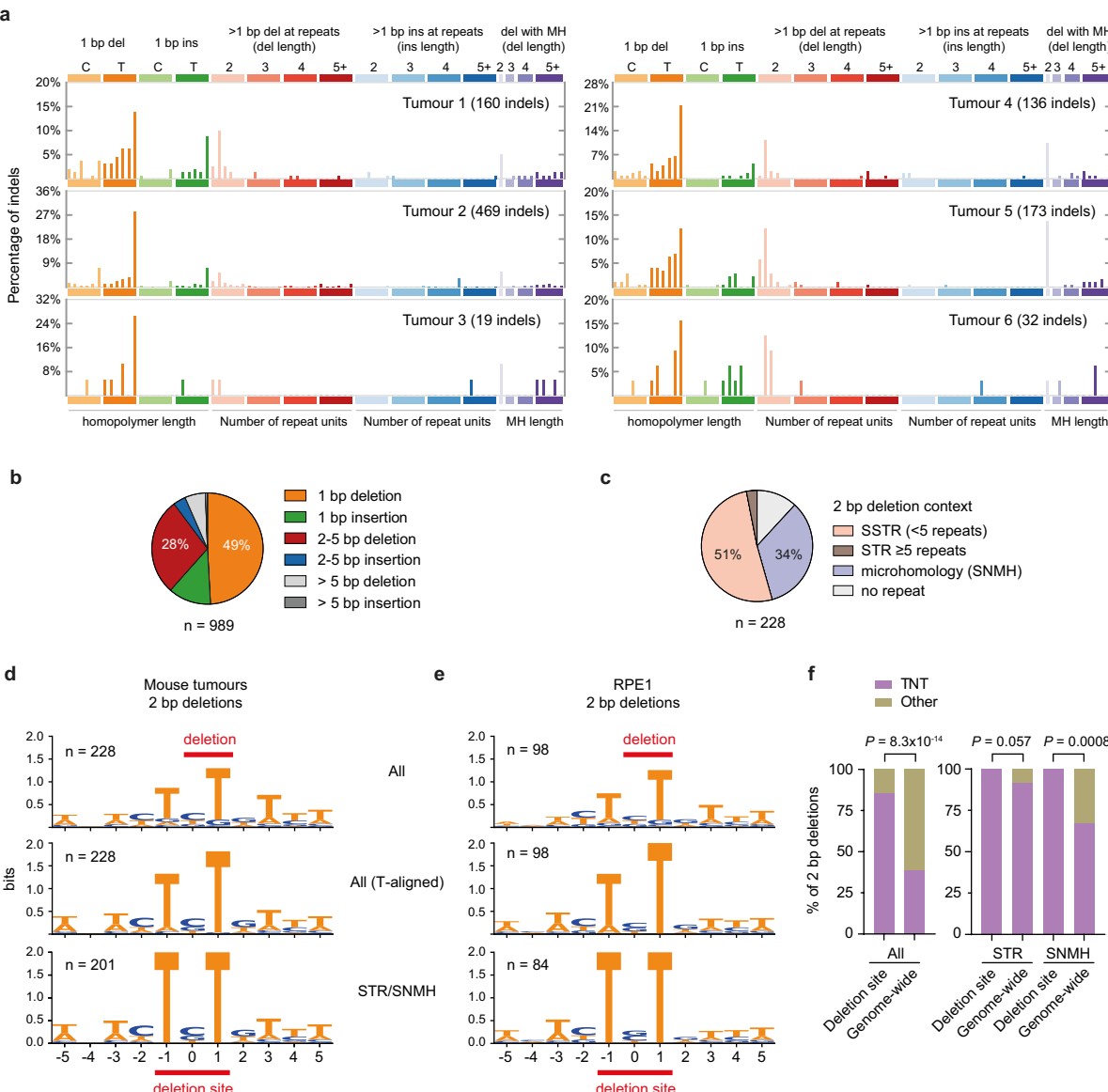

**Extended Data Fig. 6 | ID4 mutations in RNase H2 null mouse tumours and RPE1 cells occur at a TNT motif, defining ID-TOP1. a**, Mutation spectra for individual *Rnaseh2b*-KO mouse intestinal tumours (WGS, paired tumour–normal samples from 6 mice). **b**, Indel classes, detected in mouse *Rnaseh2b*-KO tumours. n, total indel count for 6 tumours. **c**, Most 2 bp deletions in these tumours occur at SSTRs and sites of single nucleotide microhomology (SNMH). n, number of 2 bp deletions. **d**, **e**, A TNT sequence motif is present at all 2 bp STR and SNMH deletions in RNase H2 null mouse tumours (**d**) and RPE1 cells (**e**). Related to Fig. 4d and Fig. 3, respectively. Sequence logo: 2-bit representation

of the sequence context of 2 bp deletions. Top, all deletions, with those sequences containing a deleted adenosine (except AT/TA) reverse complemented, and deletions right-aligned. Middle, re-aligned on right-hand T. Bottom, aligned on T (STR and SNMH context only). n, number of deletions. **f**, Deletion sites in RNase H2 null RPE1 cells are significantly enriched for the TNT sequence motif compared to genome-wide occurrence, for all genome sequence, as well as SNMH sites. P-values, two-sided Fisher's exact, observed vs expected. $n = 98$ (all; $P = 8.3 \times 10^{-14}$), 54 (STR; $P = 0.057$), 30 (SNMH; $P = 0.0008$) deletions.

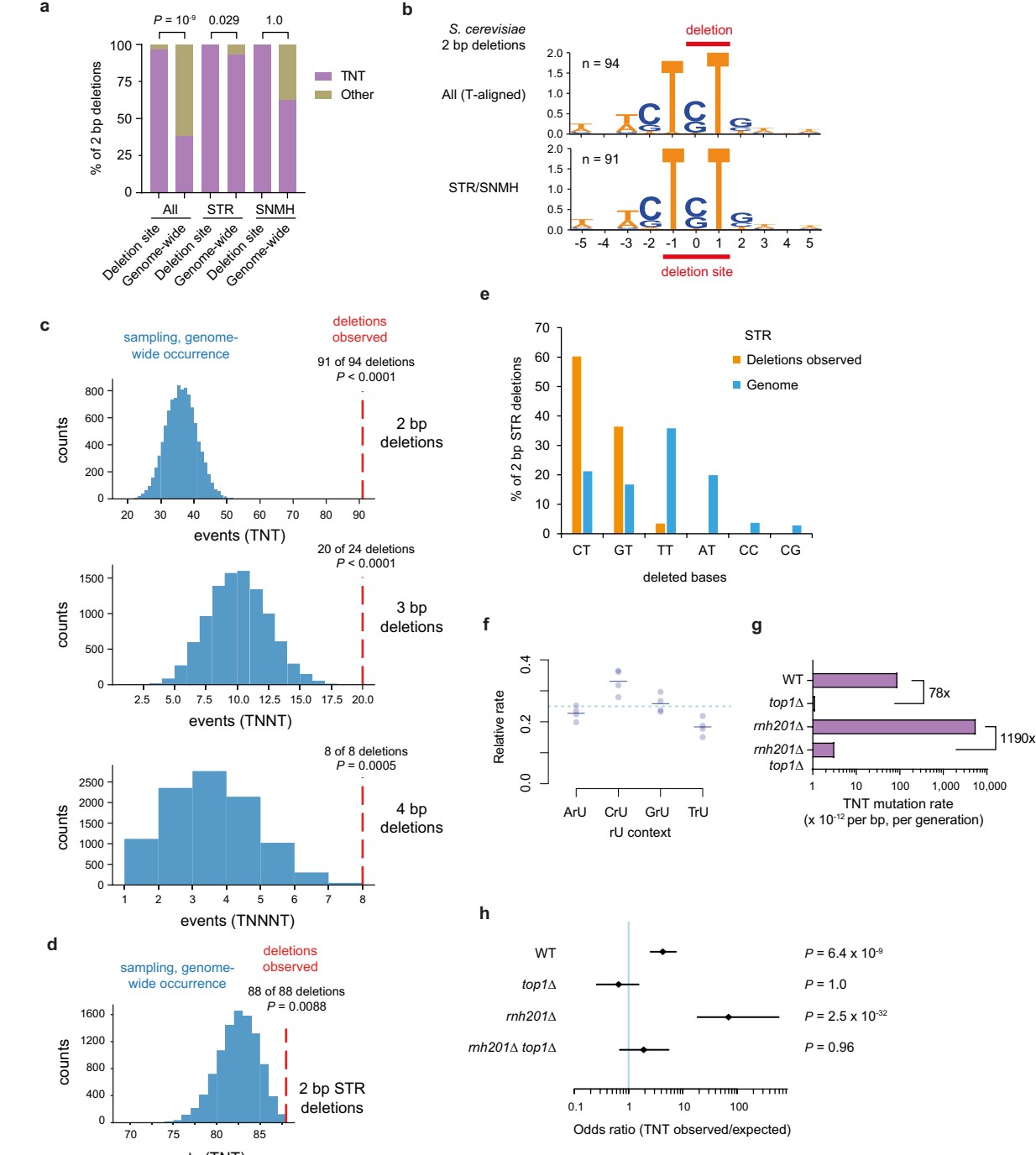

**Extended Data Fig. 7** | See next page for caption.

**Extended Data Fig. 7 | ID4 deletions in RNase H2 null *S. cerevisiae* occur at a TNT motif in a Top1-dependent manner. a**, 2 bp deletion sites in *rnh201Δ pol2-M644G* yeast are significantly enriched for the TNT sequence motif compared to genome-wide occurrence, for all genome sequence, as well as STR sites. P-values, two-sided Fisher's exact, observed vs expected. $n = 94$ (all; $P = 1.0 \times 10^{-9}$), 91 (STR; $P = 0.029$), 3 (SNMH; $P = 1$) deletions. **b**, A TNT sequence motif is present at all 2 bp STR and SNMH deletions in *rnh201Δ pol2-M644G* yeast. Sequence logo: 2-bit representation of the sequence context of 2 bp deletions. Top, all deletions, with those sequences containing a deleted adenosine (except AT/TA) reverse complemented, and deletions aligned on right-hand T. Bottom, aligned on T (STR and SNMH context only). n, number of deletions. **c**, **d**, TN*T motifs extend beyond 2 bp deletions, with enrichment above expectation for 2 bp deletions at TNT, 3 bp deletions at TNNT and 4 bp deletions at TNNNT motifs in *rnh201Δ pol2-M644G* yeast WGS. Null expectations were generated by randomly simulating deletions of 2, 3 and 4 bp (**c**) or 2 bp STR sequences (**d**) genome-wide and scoring those simulated events for TN*T compliance. Each simulated dataset matched the count of observed mutations for the corresponding deletion class and $n = 1,000$ replicate simulated datasets were produced. The frequency distribution of TN*T compliance in simulations is plotted as histograms, and comparison to the observed frequency of TN*T compliance (dotted red lines) used to derive a two-tailed empirical P-value. **e**, 2 bp STR deletions have biased sequence composition. Deletions observed in *rnh201Δ pol2-M644G* yeast WGS. Genome, frequency of dinucleotides in STR sequences in mappable genome. **f**, Ribouridine (rU) is more common in a CrU/GrU than in an ArU/TrU dinucleotide context. Genome-embedded ribonucleotide frequency determined by emRiboSeq[86]. Dotted line indicates relative rate in absence of bias (=0.25). Horizontal lines, mean; individual data points, values for $n = 4$ independent experiments[85]. **g**, **h**, 2 bp TNT deletions in wildtype and RNase H2 null cells are dependent on Topoisomerase 1. Mutation rates for 2 bp deletions at TNT-compliant SSTRs (**g**). Deletions at TNT motifs are significantly increased above expectation in WT and *rnh201Δ*, but not in *top1Δ* and *rnh201Δ top1Δ* yeast. Horizontal bars, 95% confidence intervals for odds ratio estimates (diamonds). P-values, two-sided Fisher's exact after Bonferroni correction; $n = 86, 28, 103, 19$ 2-bp deletions, with each deletion from an independent culture, for WT, *top1Δ*, *rnh201Δ*, *rnh201Δ top1Δ*, respectively. Null expectation, random occurrence of mutations in reporter target sequence (**h**).

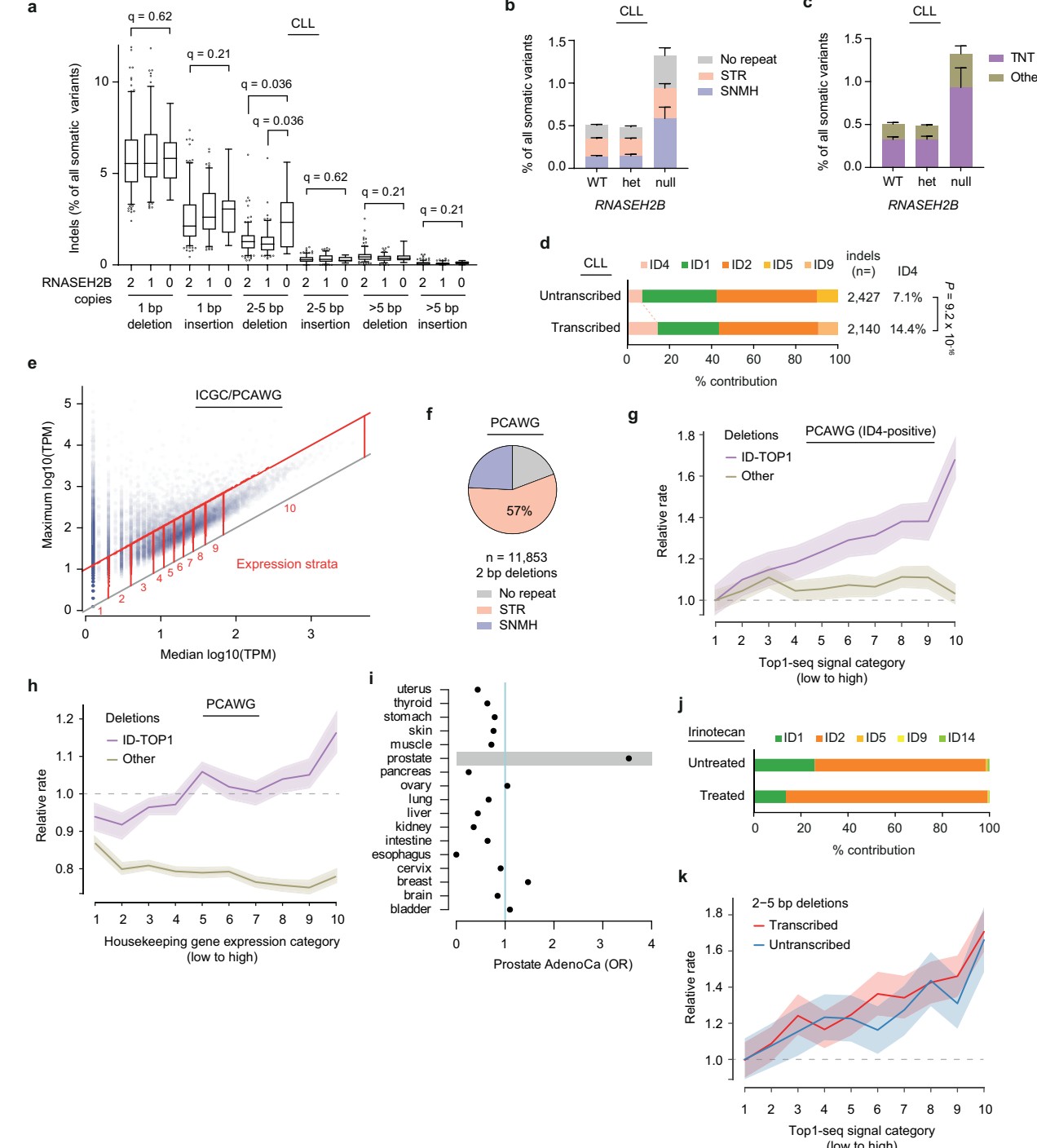

**Extended Data Fig. 8 | See next page for caption.**

**Extended Data Fig. 8 | TOP1-mediated mutagenesis causes increased 2–5 bp deletions in cancer. a**, Of all indels, only 2–5 bp deletions are significantly increased in CLL with biallelic *RNASEH2B* loss. Box, 25–75%; line, median; whiskers 5–95% with data points for values outside this range. WT (2 copies), $n = 201$; monallelic loss (1 copy), $n = 131$; biallelic loss (0 copies), $n = 16$ independent tumours. Indels as percentage of all variants per sample (GEL and ICGC data combined). q-values, 2-sided Mann-Whitney test with 5% FDR. **b, c**, In RNase H2 null CLL, 2 bp deletions predominantly occur at STR and SNMH sequences (**b**), and at the TNT sequence motif (**c**), consistent with TOP-mediated mutagenesis. Mean ± s.e.m., percentage of all variants per sample. GEL and ICGC data combined. $n = 1,711; 1,244; 443$ 2-bp indels identified in 201, 131, 16 biologically independent tumours, respectively. **d**, ID4 contribution in RNase H2 null CLL is greater in transcribed regions. Two-sided Fisher's exact test, ID4 indels vs other indels ($P = 9.2 \times 10^{-16}$). **e**, Pan-cancer transcript expression data divided into ten expression strata for ubiquitously expressed genes (used in panel **h** and Fig. 5b analysis). Data points, median/maximum expression across cancer types for individual genes. Genes with similar median and maximum TPMs were considered to be ubiquitously expressed and divided into expression groups from low (1) to high (10) expression. **f**, Two bp deletions in cancer preferentially occur at STRs. **g**, ID-TOP1 deletions increase in frequency with TOP1 cleavage activity (measured by TOP1-Seq;[38]). Dotted line, relative rate in lowest TOP1-seq category set to 1. Solid lines, relative deletion rate. ID-TOP1, 2–5 bp MH and SSTR deletions containing the TN*T sequence motif. **h**, ID-TOP1, but not deletions in other sequence contexts, correlate with transcription. **i**, 2–5 bp deletions from prostate adenocarcinoma are most enriched amongst the top 10% of highly expressed prostate 'tissue-restricted' genes. Odds ratio (OR): number of 2–5 bp deletions in top 10% tissue restricted genes vs 2–5 bp deletions in other genes, relative to expected frequency from all other tissues. **j**, ID4 is not detected in the indel signature of irinotecan-treated colorectal cancers. Untreated ($n = 78$), treated ($n = 39$). **k**, 2–5 bp deletion frequency in cancer corresponds to TOP1 cleavage activity, in both genic and non-genic regions. Data analysed from PCAWG[50], all tumours in **e**, **h**; ID4 positive tumours in **g**, **k**; Genomics England in **j**. In **g**, **h** and **k**, solid line, relative deletion rate; shading indicates 95% confidence intervals from 1,000 (**g**, **k**) or 100 (**h**) bootstrap replicates.

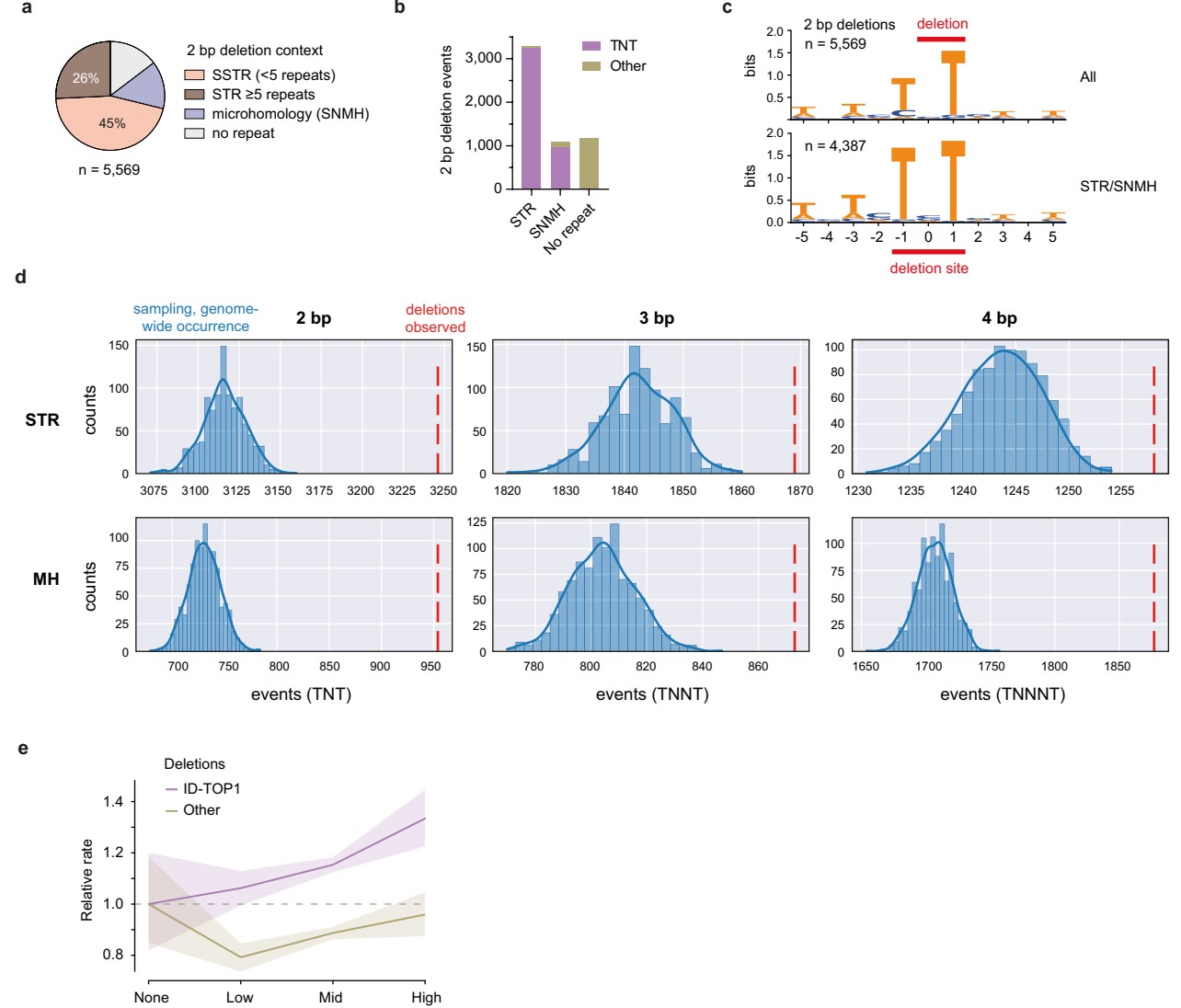

**Extended Data Fig. 9 | Human germline de novo indels are enriched for ID-TOP1 deletions. a**, Most de novo 2 bp deletions occur at SSTR, STR and SNMH sequences. **b**, **c**, A TNT sequence motif is present at the majority of 2 bp STR and SNMH deletions (**b**). Sequence logos: 2-bit representation of the sequence context of 2 bp deletions. Top, all deletions, with those containing A (except AT/TA) reverse complemented, and deletions right-aligned on T (where present). Bottom, STR/SNMH deletions only (**c**). **d**, TN*T motifs extend beyond 2 bp deletions, with enrichment above expectation for 2 bp deletions at TNT, 3 bp deletions at TNNT and 4 bp deletions at TNNNT motifs ($P < 0.001$; two-tailed empirical P-value determined for each category). Bootstrap sampling ($n = 1,000$) of 2, 3 and 4 bp STR/MH sequences genome-wide to derive

expected frequencies of those matching TN*T motifs. Sampling was performed to match the numbers of deletions at repeats observed in the Gene4Denovo database for each category defined by repeat type, repeat unit length and total repeat length. Histograms, distribution of the number of repeats matching TN*T motifs over these samplings. Solid blue lines, kernel density estimates for these distributions. Dotted red lines, number of deletions observed in Gene4Denovo matching TN*T motifs for each category. **e**, ID-TOP1 correlates with germline expression level. ID-TOP1, defined as 2–5 bp MH and SSTR deletions containing the TN*T sequence motif. Shading, 95% confidence intervals from 100 bootstrap replicates.

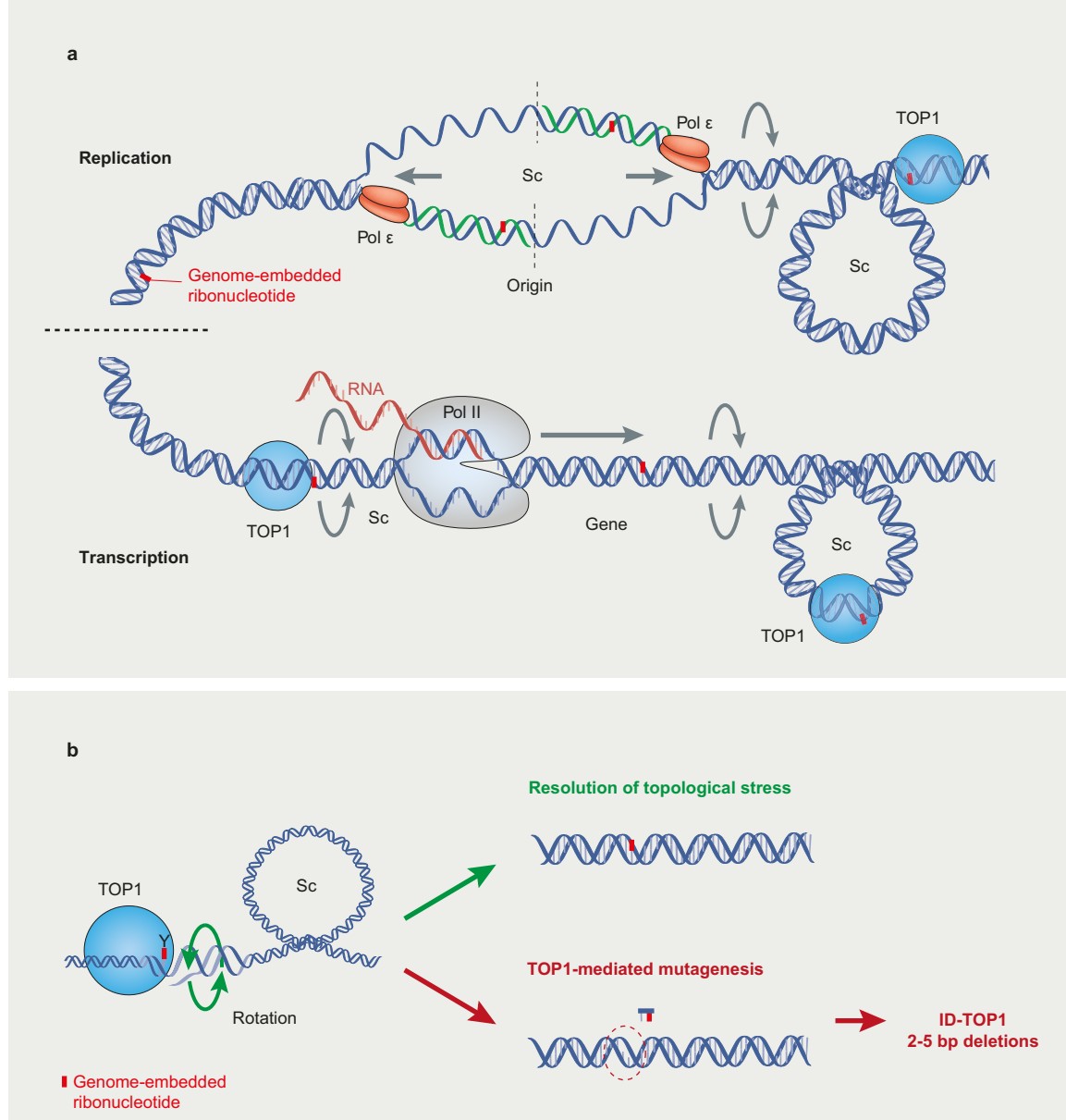

**Extended Data Fig. 10 | Topoisomerase 1 causes small deletions while protecting against topological stress. a**, The canonical role of Topoisomerase 1 (TOP1) is to relieve torsional stress (sc, supercoiling) during replication and transcription. **b**, TOP1 acts by forming ssDNA nicks to release supercoils and then religates the relaxed DNA. However, TOP1 cleavage at genome-embedded ribonucleotides (frequently incorporated by replicative polymerases such as Pol ε), can lead to short deletions that will be most frequent at sites of torsional stress in the genome, such as occurs at highly transcribed genes. Adapted with permission from ref. [6], Springer Nature.

| --- | --- |

# Reporting Summary

## Statistics

For all statistical analyses, confirm that the following items are present in the figure legend, table legend, main text, or Methods section.

| n/a | Confirmed | |
| --- | --- | --- |
| ☐ | ☒ | The exact sample size ($n$) for each experimental group/condition, given as a discrete number and unit of measurement |
| ☐ | ☒ | A statement on whether measurements were taken from distinct samples or whether the same sample was measured repeatedly |
| ☐ | ☒ | The statistical test(s) used AND whether they are one- or two-sided<br>*Only common tests should be described solely by name; describe more complex techniques in the Methods section.* |
| ☒ | ☐ | A description of all covariates tested |
| ☐ | ☒ | A description of any assumptions or corrections, such as tests of normality and adjustment for multiple comparisons |
| ☐ | ☒ | A full description of the statistical parameters including central tendency (e.g. means) or other basic estimates (e.g. regression coefficient) AND variation (e.g. standard deviation) or associated estimates of uncertainty (e.g. confidence intervals) |
| ☐ | ☒ | For null hypothesis testing, the test statistic (e.g. $F$, $t$, $r$) with confidence intervals, effect sizes, degrees of freedom and $P$ value noted<br>*Give P values as exact values whenever suitable.* |
| ☒ | ☐ | For Bayesian analysis, information on the choice of priors and Markov chain Monte Carlo settings |
| ☒ | ☐ | For hierarchical and complex designs, identification of the appropriate level for tests and full reporting of outcomes |
| ☐ | ☒ | Estimates of effect sizes (e.g. Cohen's $d$, Pearson's $r$), indicating how they were calculated |

*Our web collection on statistics for biologists contains articles on many of the points above.*

## Software and code

Policy information about availability of computer code

| Data collection | No software was used for data collection |
| --- | --- |
| Data analysis | Sequencher v5.4.6; Mutation Surveyor v3.30; AIDA Image Analyzer v3.44.035; GraphPad Prism v9.1.1; Microsoft Excel 2016; Python v3.8.5; R v3.3.1/v4.0.5/v4.1.0; SRA Toolkit v2.5.4-1; BWA-MEM v0.7.12/v0.7.15/v0.7.16; SAMBLASTER v0.1.22; GATK v3.6-0/v4.0.2.0/v4.1.9.0; Somalier v0.2.12; ISAAC aligner v03.16.02.19; cgpBattenberg v3.2.2;ascatNgs v4.1.0; Nexus 9.0 Biodiscovery; Strelka v2.1.9.10/v2.4.7/v2.8.2; SvABA v1.1.0/v1.1.3; Platypus v0.8.1; Manta v0.28.0/v1.2/v1.6.0; Canvas v1.3.1; biobambam2 v2.0.65; IGV v2.5.0; bcftools v1.10.2; Bedtools v2.30.0; samtools v1.9; liftover (kent source version 417); Logomaker v0.8; SigProfilerExtractor v1.1.0; SigprofilerMatrixGenerator v1.1.14/1.1.15; SigprofilerPlotting v1.1.27; scipy v1.6.3. Custom code available at https://git.ecdf.ed.ac.uk/ID-TOP1 |

For manuscripts utilizing custom algorithms or software that are central to the research but not yet described in published literature, software must be made available to editors and reviewers. We strongly encourage code deposition in a community repository (e.g. GitHub). See the Nature Portfolio guidelines for submitting code & software for further information.

## Data

Policy information about availability of data

All manuscripts must include a data availability statement. This statement should provide the following information, where applicable:

- Accession codes, unique identifiers, or web links for publicly available datasets
- A description of any restrictions on data availability
- For clinical datasets or third party data, please ensure that the statement adheres to our policy

RPE1 mutation accumulation experiment and mouse tumour WGS data are available from European Nucleotide Archive under accession number PRJEB48753 (https://www.ebi.ac.uk/ena/browser/view/PRJEB48753).

CLL and CRC data from the 100,000 Genomes Project are held in a secure Research Environment to protect participant privacy and can be accessed by joining an appropriate GECIP Domain using the application form at https://www.genomicsengland.co.uk/join-a-gecip-domain/. Detailed information on accessing 100,000 Genomes Project data including expected application timeframes and data use restrictions can be found at https://research-help.genomicsengland.co.uk/display/OC/GeCIP+and+your+access+to+data.

ICGC-CLL data is available at the European Genome-Phenome Archive (EGA, http://www.ebi.ac.uk/ega/), hosted at the EBI, under accession number EGAS00001001306.

ICGC PCAWG somatic mutations and mutational signature data were obtained from https://dcc.icgc.org/releases/PCAWG.

ICGC PCAWG "baseline" gene expression data were obtained from ArrayExpress (https://www.ebi.ac.uk/arrayexpress/experiments/E-MTAB-5200/)

Genome sequencing data from S. cerevisiae rnh201Δ pol2-M644G strains were obtained from NCBI SRA database, study no. SRP062900 (https://www.ncbi.nlm.nih.gov/sra?linkname=bioproject_sra_all&from_uid=293995).

Human de novo mutations were downloaded from the Gene4Denovo database (http://genemed.tech/gene4denovo/download).

Human germ cell transcriptome data is available at the NCBI GEO database (accession code GSE125372) and as a supplementary table from https://www.cell.com (https://www.cell.com/cms/10.1016/j.cell.2019.12.015/attachment/08d8d7db-2f52-499b-999e-4be5d6316e71/mmc5.xlsx).

Top1-seq data were obtained from NCBI GEO database, accession code GSE57628; samples GSM1385717 and GSM1385718 (https://www.ncbi.nlm.nih.gov/geo/query/acc.cgi?acc=GSE57628).

emRiboSeq data from rnh201Δ yeast were obtained from NCBI SRA database (https://www.ncbi.nlm.nih.gov/sra), accession codes SRX824147, SRX824139, SRX824136 and SRX824134.

Human GRCh37 reference genome sequence was obtained from ftp://ftp-trace.ncbi.nih.gov/1000genomes/ftp/technical/reference/phase2_reference_assembly_sequence/hs37d5.fa.gz.

Human hg38 reference genome was obtained from ftp://hgdownload.cse.ucsc.edu/goldenPath/ hg38/bigZips/hg38.fa.gz.

Mouse GRCm38 reference genome was obtained from ftp://ftp-mouse.sanger.ac.uk/ref/GRCm38_68.fa.gz.

The delta|(-2)|-7B-YUNI300 S. cerevisiae reference genome was obtained from NCBI GEO accession GSE56939  (https://ftp.ncbi.nlm.nih.gov/geo/series/GSE56nnn/GSE56939/suppl/GSE56939_L03_ref_v2.fa.gz).

Gene annotations were obtained from Ensembl (https://www.ensembl.org, ftp://ftp.ensembl.org/pub/release-90/gtf/homo_sapiens/Homo_sapiens.GRCh38.90.gtf.gz and http://ftp.ensembl.org/pub/release-75/gtf/homo_sapiens/Homo_sapiens.GRCh37.75.gtf.gz) and from GENCODE (https://ftp.ebi.ac.uk/pub/databases/gencode/Gencode_human/release_38/gencode.v38.annotation.gff3.gz and https://ftp.ebi.ac.uk/pub/databases/gencode/Gencode_mouse/release_M25/gencode.vM25.annotation.gff3.gz).

Mouse short indel and structural variant data were obtained from the Mouse Genomes Project (https://ftp.ncbi.nih.gov/snp/organisms/human_9606_b151_GRCh37p13/VCF/All_20180423.vcf.gz).

Human short polymorphism data were obtained from dbSNP151 (https://ftp.ncbi.nih.gov/snp/organisms/human_9606_b151_GRCh37p13/VCF/All_20180423.vcf.gz).

Human Structural Variant Data were obtained from dbVar (https://hgdownload.soe.ucsc.edu/gbdb/hg38/bbi/dbVar).

Genome mappability data was downloaded from https://bismap.hoffmanlab.org.

# Field-specific reporting

Please select the one below that is the best fit for your research. If you are not sure, read the appropriate sections before making your selection.

☒ Life sciences ☐ Behavioural & social sciences ☐ Ecological, evolutionary & environmental sciences

For a reference copy of the document with all sections, see nature.com/documents/nr-reporting-summary-flat.pdf

# Life sciences study design

All studies must disclose on these points even when the disclosure is negative.

| | |
|---|---|
| Sample size | No statistical methods were used to pre-determine sample size. Sample sizes for fluctuation assays were performed in line with standard practice in the field (doi.org/10.1385/1-59259-761-0:003). Samples sizes for the mutation accumulation experiment were determined based on the practicality of large-scale tissue culture and time frame of the experiment (6-9 months). Sample availability of stored mouse tumours determined sample size for WGS. For CLL-GEL , CLL-ICGC, PCAWG and Gene4Denovo, sample size was determined by available data. |
| Data exclusions | CLL-GeL: samples were excluded from analysis on pre-established criteria of <50% tumor cellularity. Subsequently, 6 further cases were excluded as indel calling with Mutect2, Platypus and/or SvABA failed. Likewise, for mouse intestinal tumours, n=3 were excluded due to low cellularity, on the basis of median SNV MAF<10%. |
| Replication | The number of times each experiment was repeated with similar results is stated in figure legends or the Methods section. Fluctuation assays were performed with a minimum of 9 independent cultures. The mutation accumulation experiment was performed once with 2 independent KO clones and 3 independent WT clones. All attempts at replication were successful for the experiments described in the manuscript. |
| Randomization | Samples were allocated to groups on the basis of genotype of interest (e.g. RNase H2 status). No randomization was performed. |
| Blinding | The investigator performing the fluctuation assays with the HeLa reporter cells was blinded to the identity of the different clones. RNASEH2B genotypic assignment for CLL samples was done prior to and independent of the person who performed the indel variant count analyses. For other experiments the investigators were not blinded during data collection and analysis, as this is not standard practice in the field (enzyme assays, alkaline gels, immunoblotting, yeast fluctuation assays). Automated colony counting was performed for fluctuation assays to avoid observer bias, and indel counts were called programmatically; therefore analyst blinding was not necessary. |

# Reporting for specific materials, systems and methods

We require information from authors about some types of materials, experimental systems and methods used in many studies. Here, indicate whether each material, system or method listed is relevant to your study. If you are not sure if a list item applies to your research, read the appropriate section before selecting a response.

## Materials & experimental systems

| n/a | Involved in the study |
|-----|----------------------|
| ☐ | ☒ Antibodies |
| ☐ | ☒ Eukaryotic cell lines |
| ☒ | ☐ Palaeontology and archaeology |
| ☐ | ☒ Animals and other organisms |
| ☐ | ☒ Human research participants |
| ☒ | ☐ Clinical data |
| ☒ | ☐ Dual use research of concern |

## Methods

| n/a | Involved in the study |
|-----|----------------------|
| ☒ | ☐ ChIP-seq |
| ☒ | ☐ Flow cytometry |
| ☒ | ☐ MRI-based neuroimaging |

# Antibodies

| | |
|--|--|
| Antibodies used | Sheep anti-pan-RNase H2 (raised against human recombinant RNase H2, 1:1,000; not commercially available, human RNase H2 was purified in the A.P.Jackson laboratory and antibody raised as part of a custom program by Eurogentec, and affinity purified using recombinant RNase H2); mouse anti-RNASEH2A G-10 (Santa Cruz Biotechnologies sc-515475, lot #A1416, 1:1,000); rabbit anti-GAPDH (Abcam ab9485, 1:2,000, lot #GR3380498-1); horseradish peroxidase (HRP)-linked Rabbit Anti-Sheep Immunoglobulins, (Dako, P04163, lot #00047199, 1:2,000); Goat Anti-Mouse Immunoglobulins/HRP-linked Antibody (Dako, P0447, lot #20039214, 1:10,000); Anti-rabbit IgG, HRP-linked Antibody (Cell Signaling Technologies, 7074S, lot #29, 1:10,000) |
| Validation | RNase H2 and RNASEH2A antibodies were previously validated using knockout cell lines (doi.org/10.1038/s41586-018-0291-z); GAPDH antibody has been previously demonstrated by the manufacturer to yield a single band of the expected size and is stated on the manufacturer's website to be cited by 2,175 publications. |

# Eukaryotic cell lines

Policy information about cell lines

| | |
|--|--|
| Cell line source(s) | HeLa cells (originally from ATCC) were a gift from G. Stewart (University of Birmingham); hTERT-RPE1 cells (originally from ATCC) were a gift from D. Durocher (The Lunenfeld–Tanenbaum Research Institute, Toronto). |
| Authentication | Cell lines were authenticated using STR DNA profiling in the labs of origin. |
| Mycoplasma contamination | All cell lines tested negative for mycoplasma contamination. |
| Commonly misidentified lines (See ICLAC register) | No commonly misidentified cell lines were used in this study. |

# Animals and other organisms

Policy information about studies involving animals; ARRIVE guidelines recommended for reporting animal research

| | |
|--|--|
| Laboratory animals | Tissues for WGS were collected from 52-week old female Villin-Cre+ Trp53-fl/fl Rnaseh2b-fl/fl mice on a C57Bl/6J background. Mice have been described previously (doi.org/10.1053/j.gastro.2018.09.047). All mice were maintained in a specific pathogen-free facility, and the quarterly health report did not indicate the presence of pathogenic species. Mice were provided with food and water ad libitum and maintained in a 12h light–dark cycle under standard conditions (ambient temperature 20-22°C, 40-60% humidity) at Kiel University. |
| Wild animals | The study did not involve wild animals. |
| Field-collected samples | The study did not involve samples collected from the field. |
| Ethics oversight | Animal experiments were conducted with appropriate permission, in accordance with guidelines for animal care of the Christian-Albrechts-University (Kiel, Germany), in agreement with national and international laws and policies. |

Note that full information on the approval of the study protocol must also be provided in the manuscript.

# Human research participants

Policy information about studies involving human research participants

| | |
|--|--|
| Population characteristics | IGCC-CLL WGS cohort: age at diagnosis 18-87 years old; 91 male, 59 female. Samples were collected before administration of any treatment.<br><br>CLL Genomics England: Demographics available for 172/198 cases: age at diagnosis 38-87y (median 65); 123 male, 49 female. WGS for 174 patients prior to treatment; 24 patients at relapse or refractory to treatment. |

CRC Genomics England: Age at diagnosis 33-81y (median 64); 71 male, 46 female. 105 patients received radiotherapy, capecitabine, irirotecan, fluorouracil or oxaliplatin treatment prior to sampling, and 12 patients were treatment-naive.

Recruitment

ICGC-CLL: CLL patients diagnosed at the Hospital Clínic of Barcelona or at a collaborative hospital and sent to the Hospital Clínic of Barcelona for further evaluation. Only patients that fulfilled the diagnosis of CLL or MBL (Monoclonal B-cell lymphocytosis) were included.

CLL Genomics England: Patients were treatment-naïve and in need of treatment according to iwCLL criteria (doi.org/10.1182/blood-2017-09-806398) as part of their enrollment in ARTIC, AdMIRe, RIAltO, FLAIR studies; apart from small subsets enrolled in CLEAR (early stage disease) and CLL210 (relapse refractory).

CRC Genomics England: Patient recruitment was organised by 13 Genomic Medicine Centres (GMCs) and their affiliated hospitals across the UK.

Ethics oversight

ICGC-CLL: All patients gave informed consent for participation in the study following the International Cancer Genome Consortium (ICGC) guidelines and the ICGC Ethics and Policy committee. The study was approved by the Research Ethics Committee of the Hospital Clínic of Barcelona.

CLL Genomics England: All patients gave written informed consent and the study was approved under the 100,000 Genomes Project Ethics Committee, East of England and South Cambridge Research Ethics Committee and the CLL Pilot Ethics approval (MREC 09/H1306/54).

CRC Genomics England: All patients gave written informed consent and the study was approved under the 100,000 Genomes Project Ethics Committee.

Note that full information on the approval of the study protocol must also be provided in the manuscript.

