## [Peer Review File · Nature]

Manuscript Title: Signatures of TOP1 transcription-associated mutagenesis in cancer and germline

Reviewer Comments & Author Rebuttals:

Reviewer Reports on the Initial Version:

Referee #1 (Remarks to the Author):

The work by Reijns et al describes a compelling set of data that links a cancer associated mutation signature ID4 to the action of Topoisomerase I at embedded ribonucleotides. Analysis of mutation signatures in yeast models of Top1 induced mutagenesis made the connection to ID4. These mutations are enhanced by loss of RNaseH2, which removes embedded ribonucleotides. Mutation accumulation studies in vitro and in mouse models confirm the association. The same mutation signature is enhanced in RNaseH2 deficient cancers at the same TnT motifs. The authors present a logical and supported mechanistic model to explain how Top1 can drive these specific mutations at TnT motifs. Finally, the authors show that the human germline can also accumulate deletion mutations at TnT motifs, suggesting Top1 as a source of de novo mutations. Overall, I thought this was a really strong manuscript, supporting interesting and broadly relevant conclusions with solid datasets. It is well written, clear and exciting. Below are several questions for clarifications or suggestions for improvements.

1. I didn't really understand the statement on Line 144 that this model underestimates the true impact of RER deficiency? Are the authors saying mutated but expressed RNaseH2 retains activity? Or that another pathway exists not found in yeast? Or that there are cells within the population that are expressing RNaseH2 normally?
2. In the in vivo experiments only 28% of indels were at 2-5bp deletions. How does this correspond to the in vitro RPE mutation accumulation study in terms of the proportion of indels? I am wondering to what extent the in vivo context, in particular the p53 deficiency, contributes to alternative mutational processes compared to the p53 proficient RPE1 cells.
3. The TnT model is very elegant and logical. Did the authors extract the same data from yeast TAM studies to determine if this model fits across species?
4. One prediction from this study is that Top1 inhibitors would alter the mutational landscape in accordance with the authors model. Are there datasets from Topotecan treated patient genomes that can be analyzed to determine effects on the ID4 signature? Could this be used to assess the importance of trapped cleavage

complexes for this particular signature?

5. If contribution of ID4 to observed signature is 61%, what is causing the remaining 39%? Or is RER-deficiency causing more than a single mutational signature? I see ID5 in Fig 3 which likely accumulates in normal cells so is this just the result of many generations? More explanation needed I think. Is it possible that R-loop accumulation due to RNaseH2 loss accounts for other mutations? Could an R-loop mutation signature be extracted, now that the RER signature is clear?

Technical queries

- I still see a faint band for RNaseH2A in the KO1 lane. It would be nice if they included some sequencing validation of the knockouts just to be safe. I appreciate given the phenotypic experiments, and instability of 2B and 2C, that this is likely a formality.
- There must be a better way to represent the schematic of the experiment (Fig 3a). The dotted lines and asterisks make it a little cumbersome.
- 5 of the mice have ~25-40% ID4 composition but number 6 shows 66%, any explanation for this variation?

Referee #2 (Remarks to the Author):

The study by Reijns et al analyzes an indel mutation signature of unknown cause, ID4, found in the COSMIC, Catalogue of Somatic Mutations in Cancer, database. ID4 consists in short deletions of 2-5 bp, in which the deleted sequence is repeated 1-3 times (short-short tandem repeats, SSTR).

Previous experiments in yeast have shown that topoisomerase 1 (Top1) activity is a major source of transcription associated mutagenesis, TAM, resulting in a transcription-dependent signature of 2-5 bp deletions at tandem repeat sequences. Moreover, in vitro assays of Top1 cleavage at ribonucleotides embedded in DNA combined with mutagenesis assays with RNase H2-null (rnh201-null) and top1-null mutant yeast cells (Kim et al Science 2011) have linked ribonucleotides to Top1-TAM deletions at tandem repeat sequences in yeast.

Here, the authors analyzed published mutation data of yeast and found correspondence between the ID4 SSTR deletions and the yeast Top1-deletion signature.

A series of reporters to detect indels at tandem repeats was developed so that the reporters are similar between those used for assays in yeast and those used for assays in human cells. Via assays in yeast WT, rnh201-null, top1-null, top1-null rnh201-null cells, the ID4 signature shows up in rnh201-null as well as wild-type but not in the top1 mutant, as expected, although the mutation rates per bp per

generation are low ($10E-9$ for *rnh201*-null and $10E-10$ for wild-type).

RNASEH2A knockout were generated in HeLa cells containing a reporter integrated. RNASEH2A was also KO in RPE1 cells and mutation accumulation was studied. Up to 7.5 fold increase in 2-5 bp deletions with $10E-10$ rate was seen in the RNASEH2A KO RPE1 cells compared to wild-type cells. These mutations had the ID4 signature also including small deletions at sequences with microhomology.

The authors also analyze DNA from small intestinal tumors (*Trp53*^{-/-} *Rnaseh2b*^{-/-}) and liver tissue (*Trp53*^{+/+} *Rnaseh2b*^{+/+}) from Villin-Cre conditional KO of RNASEH2B and *Tp53* mice. After DNA sequencing tumor specific 2-5 bp deletions with ID4 signatures were found in the tumor samples. The mouse mutation analysis was extended to characterize the sequence of the 2-5 bp deletions. It was found that mutations were frequent at TNT sites. Similar sequence preference was also found in RPE1 RNASEH2A KO cells. The ID4 are more abundant at transcribed genes. A TOP1-TAM model is proposed as cause of ID4, referred in this study as ID-TOP1.

Next, the authors examined available datasets of human cancer, with focus of a chronic lymphocytic leukemia cancer that is also defective in RNASEH2B. 2-bp deletions were found with ID4 signature. The ID4 was more frequent in transcribed regions. Also other cancer genomes in WGS analysis showed ID4 elevated in highly expressed genes with 2 bp deletions at TNT sites. These results support TAM in human cells with ID-TOP1 signature.

Analysis of germline de novo variants from the Gene4Denovo database revealed that 2-5 bp deletions are the most abundant indels found with enrichment in TNT sites. Also correlation with gene expression was found.

The work is very interesting, novel, and topical. No significant technical problems were found.

I have two major critiques.

- It would be valuable to show that such ID4 signature observed in RNASEH2A KO human cells is dependent on TOP1, as done for the yeast experiments. This would strengthen the results. For example to perform a knock down of TOP1 in the human cells and see whether the spectrum of mutations is affected accordingly.

- I do not think that it can be concluded that genome-embedded ribonucleotides cause similar mutational signature in yeast and mammalian cells, but rather that RNase H2 KO causes a similar mutational signature in yeast and mammalian cells.

While ribonucleotides are a major substrate of RNase H2, the actual presence of ribonucleotides at the sites in which the ID4 signature is observed has never been demonstrated at the genomic level in mammalian cells as well as in yeast DNA.

While it is highly expected that at the genome level ID4 sites derive from TOP1 dependent activity at embedded ribonucleotides in RNase H2 null cells, showing that ribonucleotide sites correlate with sites id ID4 signatures would provide final proof of the mechanism.

The first sentence of the Discussion << Here we establish a biological basis for the ID4 cancer signature, establishing that TOP1-mediated cleavage at genome-embedded ribonucleotides causes short deletions >> and the statements indicating mutations at << sites of genome-embedded ribonucleotides >> do not exactly reflect the work done. This study does not present any experiments with genome-embedded ribonucleotides or with ribonucleotides in general. While ribonucleotides embedded in DNA are major targets of RNase H2, RNase H2 has also functions beyond targeting ribonucleotides in DNA. Moreover, also beyond RNase H2 and TOP1, there might be other ribonucleotide targeting enzymes, e.g. the DEAD-box RNA helicase DDX3X (Riva, et al. 2020), recently found in human cells. Therefore, while the assumption that the TOP1-ID4 are linked to sites of genomic ribonucleotides in DNA is undoubtedly valid, it is still an assumption.

It would be important to perform analysis on some of the published (also by the authors) yeast *rnh201*-null libraries of genomic ribonucleotide incorporation to examine whether it is possible to identify and confirm a correlation between genomic ribonucleotide sites, particularly rU sites, and sites of TOP1-ID4 signatures at least for yeast.

It may be worth noting that CrU is found strongly above the genome-wide frequency of CT in nuclear DNA of *rnh201*-null yeast strains of *S. cerevisiae*, *S. paradoxus*, and *S. pombe*, and not in wild-type RNase H2 cells in the study by Balachander et al 2020. This biased ribonucleotide distribution may suggest that CrU sites might be a preferred target of Top1.

Minor points:

- Should indicate in the legend what is the green bar in Figure 2c. In Figure 2c legend should explain difference between HeLa and Parental cells.

- Page 6 line 136, Extended data Figure 3g,h, should be f,g.

Referee #3 (Remarks to the Author):

Reijns and colleagues investigate the mechanistic origins of ID4, a previously described mutational signature of small insertions and deletions derived from cancer genomes. The authors present a plethora of evidence from yeast, human cell lines, mouse cancers, human cancers, and the human germline to construct their argument that ID4 is due to a transcription-associated mutagenic process and they

implicate TOP1 as the main mutagenic actor.

Overall, I find this paper quite exciting. Indeed, identifying the origins of a mutational signature is interesting but the authors have also shown the existence of a widespread transcription-associated indel mutagenic process and attributed it to topoisomerase 1. In my opinion, this manuscript will be of interest to a wide range of biomedical researchers. Nevertheless, before recommending it for acceptance, I do have a number of technical comments that the authors need to address to make sure that their results are robust.

Technical Comments

1) The cosine similarity of ID4 to a yeast Top1-dependent mutational signature is 0.78. Please note that two nonnegative random vectors will have a cosine similarity of 0.75 purely by chance. As such, based on the patterns, the authors cannot claim that these are similar. To check whether these patterns are, indeed, similar, they can compare the TNT sequence motif in cancer samples with high levels of ID4 and the TNT sequence motif observed in yeast.

2) It is really not possible to visually or numerically compare the mutation spectra of WT, *rnh201Δ*, *top1Δ*, and *rnh201Δ top1Δ*. I do not find the circles in figure 1e informative and I strongly suggest replotting these using the format in figure 1a/b (at the very least add such plots to extended data). The cosine similarities are also not informative as we do not know whether these are statistically significant. Similar comment about figure 2e, 3c, 4c, 5e

3) Regarding the results for RNase H2 deficient RPE1 cells, it is hard to understand whether there is a clean pattern of ID4. The authors should include the indel plots for WT, RNASEH2A knockout, and RNASEH2B knockout. Moreover, currently, it is unclear whether the plot in fig 3d is a de novo signature detected from examining all samples or whether it is a plot of all samples with knockouts. If this is a de novo signature, is this signature present in the WT? If this is a combined plot, why were knockouts combined rather than examined separately? What is the pattern of the WT and how different is it from each of the knockouts?

4) From figure 3d, it is visually clear that ID4 and ID5 are not the only mutational signatures required to recapitulate this pattern of mutations. For example, one can see the pattern of ID8 including the 5+ indels at 1 repeats (dark red) and the indels at microhomology (purple). In a recent preprint, ID8 has been attributed to mutations in TOP2A (<https://www.biorxiv.org/content/10.1101/2020.05.22.111666v3>). Does the experimental assay allow distinguishing between mutations due to TOP1 and potential mutations due to TOP2A?

5) The analysis of murine tumors includes samples with very few mutations (i.e., 19 and 32 mutations). Signature assignment is unreliable for such low number of

mutations. Moreover, across all samples, ID4 accounts for only 32% of all indels. How do the author explain the low contribution of ID4?

6) The authors wrote “Consistent with a transcription-associated process, the ID4 signature was most evident in transcribed genomic regions (Fig. 4b).” This statement needs to be supported by statistical analysis. Based on the pattern of ID4, the authors should provide a p-value to show that there is enrichment in transcribed genomic regions than purely expected by chance. Indeed, if significant, this result provides strong evidence for a transcription-associated process. As such, the analysis should also be done for the RNase H2 deficient RPE1 cells.

7) The analysis of CLL samples is very interesting. However, why are the authors using relative percentages/contributions to deletions (fig. 5a) and indels (extended data fig. 6a)? These will be dependent on the other indel mutational processes active in CLL. Unless I am missing something, the authors should be comparing the actual numbers of deletions and the numbers of indels instead of the percentages. Also, the presented p-values should be FDR corrected and reported as q-values.

8) The increase of 2-5bp deletions in highly expressed housekeeping genes across PCAWG as well as their increase in the germline provides strong evidence for a transcription associated process. Is it possible to do this analysis using signature ID4 rather than using only the 2-5bp deletions?

9) How specific is the TNT sequence motif? Essentially, is it possible other topoisomerase or other enzymes to play a role in this mutagenic process and have a similar motif?

Author Rebuttals to Initial Comments:

We thank the reviewers for their enthusiasm and interest in this manuscript. We are grateful for their insightful and helpful comments, which we feel have significantly strengthened the revised manuscript, and hope that they share this view.

The editor asked us to provide additional evidence linking mammalian TOP1 and ID4. In our revision, we provide further analyses and we:

- demonstrate that deletion frequency is dependent on TOP1 cleavage activity level (revised Fig. 5f and Extended Data Fig. 8).
- establish that deletions at the TNT motif are Topoisomerase 1-dependent and conserved between yeast and humans (Extended Data Fig. 6 and 7).

- document a biased sequence context for embedded ribonucleotides that corresponds to increased CrU (and GrU) dinucleotide deletion frequency (Extended data Fig. 7)
- describe published work, that supports direct links between TOP1 activity at DNA-embedded ribonucleotides and ID4 deletions. This includes the biochemical reconstitution of 2 bp deletions at ribonucleotides with human TOP1.

Referees' comments:

Referee #1 (Remarks to the Author):

The work by Reijns et al describes a compelling set of data that links a cancer associated mutation signature ID4 to the action of Topoisomerase I at embedded ribonucleotides. Analysis of mutation signatures in yeast models of Top1 induced mutagenesis made the connection to ID4. These mutations are enhanced by loss of RNaseH2, which removes embedded ribonucleotides. Mutation accumulation studies in vitro and in mouse models confirm the association. The same mutation signature is enhanced in RNaseH2 deficient cancers at the same TnT motifs. The authors present a logical and supported mechanistic model to explain how Top1 can drive these specific mutations at TnT motifs. Finally, the authors show that the human germline can also accumulate deletion mutations at TnT motifs, suggesting Top1 as a source of de novo mutations. Overall, I thought this was a really strong manuscript, supporting interesting and broadly relevant conclusions with solid datasets. It is well written, clear and exciting. Below are several questions for clarifications or suggestions for improvements.

We thank the reviewer for appreciating the strength, clarity and interest of our manuscript.

1. I didn't really understand the statement on Line 144 that this model underestimates the true impact of RER deficiency? Are the authors saying mutated but expressed RNaseH2 retains activity? Or that another pathway exists not found in yeast? Or that there are cells within the population that are expressing RNaseH2 normally?

We apologise, we meant the former: the RNASEH2A+ cells used in these experiments retain RNase H2 activity despite being mutated. We have revised the text on line 144 accordingly, to state:

“However, the increased mutation rate in RNase H2 null HeLa cells likely underestimates the true impact of RER deficiency in human cells, as despite the control RNASEH2A+ HeLa reporter cells retaining protein expression (Fig. 2b), the clone had also acquired mutations at the CRISPR editing site that reduced enzymatic activity (Fig 2c), causing a moderate increase in genomic ribonucleotide content (Extended Data Fig. 3f,g).”

2. In the in vivo experiments only 28% of indels were at 2-5bp deletions. How does this correspond to the in vitro RPE mutation accumulation study in terms of the proportion of indels? I am wondering to what extent the in vivo context, in particular the p53 deficiency, contributes to alternative mutational processes compared to the p53 proficient RPE1 cells.

In fact, 16% of indels were 2-5bp deletions in RNase H2 null RPE1 cells (Extended Data Fig. 4f), so less rather than more deletions were seen in this experiment. Differing proportions in the two experiments likely reflects cell types/tissue context. Deletion frequency may be influenced by processes that either impact on the likelihood of deletions occurring (e.g. ribonucleotide incorporation driven by cellular dNTP and rNTP concentrations) or on the contribution of other background mutational processes (e.g. ID1, ID2 replication slippage mutations) that may dilute the percentage contribution of ID4 to the final mutational pattern. While p53 deficiency could be promoting additional mutational processes, it will be doing so in both experiments, as RNase H2 null and control RPE1 cells were also p53^{-/-}.

We had stated in the legend of Extended Data Fig. 4 and Methods that the cells were p53 deficient, “As RNase H2 deficiency activates the p53 pathway^{16,57}, experiments were performed in a *TP53* knockout background”. However, given the reviewer’s question, we realise we should be more explicit and we now refer to this in the main text:

“To confirm these findings we used a complementary approach to establish the relevance of such mutational events genome-wide, performing mutation accumulation experiments using hTERT-RPE1 (*TP53*^{-/-}) diploid cell lines.”

In addition, we also state in Fig. 3 legend:

“Long-term culture of hTERT-RPE1 *TP53*^{-/-} RNase H2 wildtype (WT) and null cell lines (AKO, BKO: RNASEH2A, RNASEH2B knockout respectively) bottlenecked every 25 doublings by single cell sorting.”

3. The TnT model is very elegant and logical. Did the authors extract the same data from yeast TAM studies to determine if this model fits across species?

We thank the reviewer for this helpful suggestion. We find that the TNT motif is also significantly enriched in yeast, both in the genome-wide dataset for RNase H2 null cells and in our yeast reporter experiments (new Extended Data Fig. 7). The model is therefore applicable across species, and as deletions at the TNT motif are Top1-dependent in the yeast reporter experiments (Extended Data Fig. 7g,h), this further strengthens the link between Topoisomerase 1 and mammalian mutagenesis.

4. One prediction from this study is that Top1 inhibitors would alter the mutational landscape in accordance with the authors model. Are there datasets from Topotecan treated patient genomes that can be analyzed to determine effects on the ID4 signature?

We identified irinotecan-treated colorectal cancers in the Genomics England 100,000 Genome project. Irinotecan is another clinically approved Topoisomerase 1 inhibitor, of the same class as Topotecan and Camptothecin. However, the ID4 signature was not detected in the treated tumours (Extended Data Fig. 8g).

Could this be used to assess the importance of trapped cleavage complexes for this particular signature?

As ID4 is not detected in irinotecan-treated tumours, the above findings favour ribonucleotides as the more important source of this signature. This is consistent with a previous study which found that camptothecin-treated yeast only had a 2-fold increase in 2bp deletions, compared with a ~300-fold increase with RNase H2

deficiency (Sloan et al., 2017). Mechanistically this could be accounted for by drug-induced trapping of Top1 resulting in Top1cc formation, which is not compatible with a second local cleavage event that occurs at ribonucleotides and drives mutagenesis.

We provide this data in Extended Data Fig. 8g, and have added the following statement in the discussion:

“Given that genome-embedded ribonucleotides are the most common endogenous lesion in replicating mammalian cells¹⁶, they are the most likely sites of TOP1-TAM mutagenesis, where TOP1 could cleave before their removal by RNase H2-dependent RER. Processing of TOP1cc may be an alternative, less frequent source of 2-5 bp deletions⁴⁷, but we did not detect ID4 in Topoisomerase 1 inhibitor treated cancers (Extended Data Fig. 8g).”

5. If contribution of ID4 to observed signature is 61%, what is causing the remaining 39%? Or is RER-deficiency causing more than a single mutational signature? I see ID5 in Fig 3 which likely accumulates in normal cells so is this just the result of many generations?

More explanation needed I think.

The reviewer is referring to Fig. 3d,e of the original manuscript, where one might expect ID4 to account for 100% of events, as background mutational processes have been subtracted. However, RER-deficient cells grow more slowly than wildtype cells (Reijns et al., 2012), and given its clock-like behaviour we agree with the reviewer that the ID5 contribution likely reflects RER deficient cells being grown longer to reach the 100 generations in the experiment.

To address the above point, we state in the main text:

“ID5, a clock-like signature⁶, was also enriched in KO cells, likely due to slower growth and longer culture time needed to achieve the same number of doublings for RNase H2 null cells¹⁶.”

Is it possible that R-loop accumulation due to RNaseH2 loss accounts for other mutations? Could an R-loop mutation signature be extracted, now that the RER signature is clear?

R-loop induced genome instability is mutagenic, inducing hyperrecombination (Huertas and Aguilera, 2003) and promoting chromosomal DNA rearrangements (Li and Manley, 2005). This will therefore be most likely detected by assessing large scale genome rearrangement events, rather than the short indels studied here. Although beyond the scope of this work, in future it would be interesting to address this by performing mutation accumulation experiments followed by long-read sequencing in RNase H2 null cell lines and controls complemented with the RNASEH2A separation of function mutations that abrogate RER activity but not activity against RNA/DNA heteroduplexes (Chon et al., 2013; Zimmermann et al., 2018).

To incorporate the reviewer's point, namely that additional mutational signatures associated with TOP1 or RNase H2 could be discovered, we now state in the discussion that "Additional signatures associated with topoisomerases or indeed RNase H2 may be identified in future".

Technical queries

- I still see a faint band for RNaseH2A in the KO1 lane. It would be nice if they included some sequencing validation of the knockouts just to be safe. I appreciate given the phenotypic experiments, and instability of 2B and 2C, that this is likely a formality.

The faint band for RNASEH2A-KO1 represents expression from a mutant allele with an in-frame deletion that removes essential catalytic site residues. The RNase H2 activity assay (Fig. 2C) and alkaline gel (Extended Data Fig. 3f,g) show that both this line and KO2 are similarly RER deficient. We document the mutations generated by CRISPR/Cas9 genome editing, as determined by Sanger sequencing, in Supplementary Table 4.

- There must be a better way to represent the schematic of the experiment (Fig 3a). The dotted lines and asterisks make it a little cumbersome.

We have removed the dashed lines from the schematic, and replaced the asterisk, with the words “single cell sort”.

- 5 of the mice have ~25-40% ID4 composition but number 6 shows 66%, any explanation for this variation?

Mutational events occur stochastically, and so their representation would be expected to vary between individual tumours. Also, mouse 6 had a relatively low number of indels and the estimated percentage contribution of individual mutational signatures will therefore be less accurate. For clarity, we now provide COSMIC-style indel plots for each individual tumour in Extended Data Fig. 6a.

Referee #2 (Remarks to the Author):

The study by Reijns et al analyzes an indel mutation signature of unknown cause, ID4, found in the COSMIC, Catalogue of Somatic Mutations in Cancer, database. ID4 consists in short deletions of 2-5 bp, in which the deleted sequence is repeated 1-3 times (short-short tandem repeats, SSTR).

Previous experiments in yeast have shown that topoisomerase 1 (Top1) activity is a major source of transcription associated mutagenesis, TAM, resulting in a transcription-dependent signature of 2-5 bp deletions at tandem repeat sequences. Moreover, in vitro assays of Top1 cleavage at ribonucleotides embedded in DNA combined with mutagenesis assays with RNase H2-null (rnh201-null) and top1-null mutant yeast cells (Kim et al Science 2011) have linked ribonucleotides to Top1-TAM deletions at tandem repeat sequences in yeast.

Here, the authors analyzed published mutation data of yeast and found correspondence between the ID4 SSTR deletions and the yeast Top1-deletion signature.

A series of reporters to detect indels at tandem repeats was developed so that the reporters are similar between those used for assays in yeast and those used for assays in human cells. Via assays in yeast WT, rnh201-null, top1-null, top1-null

rnh201-null cells, the ID4 signature shows up in rnh201-null as well as wild-type but not in the top1 mutant, as expected, although the mutation rates per bp per generation are low ($10E-9$ for rnh201-null and $10E-10$ for wild-type).

RNASEH2A knockout were generated in HeLa cells containing a reporter integrated. RNASEH2A was also KO in RPE1 cells and mutation accumulation was studied. Up to 7.5. fold increase in 2-5 bp deletions with $10E-10$ rate was seen in the RNASEH2A KO RPE1 cells compared to wild-type cells. These mutations had the ID4 signature also including small deletions at sequences with microhomology.

The authors also analyze DNA from small intestinal tumors (Trp53^{-/-} Rnaseh2b^{-/-}) and liver tissue (Trp53^{+/+} Rnaseh2b^{+/+}) from Villin-Cre conditional KO of RNASEH2B and Tp53 mice. After DNA sequencing tumor specific 2-5 bp deletions with ID4 signatures were found in the tumor samples. The mouse mutation analysis was extended to characterize the sequence of the 2-5 bp deletions. It was found that mutations were frequent at TNT sites. Similar sequence preference was also found in RPE1 RNASEH2A KO cells. The ID4 are more abundant at transcribed genes. A TOP1-TAM model is proposed as cause of ID4, referred in this study as ID-TOP1.

Next, the authors examined available datasets of human cancer, with focus of a chronic lymphocytic leukemia cancer that is also defective in RNASEH2B. 2-bp deletions were found with ID4 signature. The ID4 was more frequent in transcribed regions. Also other cancer genomes in WGS analysis showed ID4 elevated in highly expressed genes with 2 bp deletions at TNT sites. These results support TAM in human cells with ID-TOP1 signature.

Analysis of germline de novo variants from the Gene4Denovo database revealed that 2-5 bp deletions are the most abundant indels found with enrichment in TNT sites. Also correlation with gene expression was found.

The work is very interesting, novel, and topical. No significant technical problems were found.

We thank the reviewer for their interest in our work and for recognising its novelty.

I have two major critiques.

[2.1]- It would be valuable to show that such ID4 signature observed in RNASEH2A KO human cells is dependent on TOP1, as done for the yeast experiments. This would strengthen the results. For example to perform a knock down of TOP1 in the human cells and see whether the spectrum of mutations is affected accordingly.

We have strengthened evidence for the dependence of the mutation signature on TOP1 in humans. This has been a challenging to do, as mammalian TOP1 is cell essential, unlike its yeast counterpart. Therefore, directly recapitulating experiments previously performed in yeast is not feasible.

The reviewer suggests TOP1 depletion; however given it is cell essential, there are significant limitations to this approach as well. TOP1 depletion would need to be at sufficiently low levels and for an extended duration, to allow mutations to accumulate. At the same time, enough enzyme activity is required to maintain viability and allow cells to continue proliferating. Furthermore low levels of TOP1 activity may still be sufficient to cause 2-5 bp deletions and the same mutation spectrum, hence a meaningful outcome would be doubtful.

Nevertheless, we performed a pilot experiment to examine this possibility further. We used conditions we had previously established (Zimmermann et al., 2018), on the basis that TOP1 depletion sufficient to rescue TOP1-mediated DNA breaks in RNase H2 null cells would be a minimum requirement. Unfortunately, while cell viability was at acceptable levels at early time points used in previous experiments (Zimmermann et al., 2018), substantial levels of apoptosis were evident in siTOP1 cells at 96h post transfection (Fig R1). As mutation reporter experiments require cells to be cultured for 21-28 days, this undermined our confidence in the feasibility of this approach. In addition, such TOP1-depleted cells would need to survive additional cellular stressors such as clonal growth.

Fig. R1 Depletion of human TOP1 causes high levels of apoptosis at extended timepoints. **a,b**, Apoptosis measured by FACS analysis of Cleaved Caspase-3 at time points following transfection of siRNA (40 nM) as indicated. **c**, immunoblotting for human TOP1 at 96 h post-transfection, demonstrating depletion of TOP1 to 22% and 18% of siLUC control levels in wildtype and *RNASEH2A*-KO HeLa cells respectively (TOP1 band intensity determined by ImageQuant, normalised to α -Tubulin loading control). Negative control, siLUC, targets Luciferase, not expressed in these cells.

Therefore, we took alternative approaches to demonstrate that the mutation signature is dependent on human TOP1 to address the reviewer's critique. We have incorporated three lines of such evidence into our revised manuscript:

1. Analysis of TOP1-seq data demonstrates deletion frequency to be dependent on TOP1 activity

We identified a dataset in which TOP1 activity was mapped genome-wide in human cells (Baranello et al., 2016). Using this, we establish that the frequency of 2-5bp deletions and the ID-TOP1 signature are dependent on the level of TOP1 activity. This is the case in both transcribed and non-transcribed genome

regions (Fig. 5f and Extended Data Fig. 8e,h), demonstrating that as well as in highly transcribed genes, TOP1 induced mutations will occur in other genomic regions where TOP1 is active to counter topological stress (for instance during DNA replication, Extended Data Fig. 10).

2. The TNT motif is Topoisomerase 1 dependent and conserved between yeast and humans

We find that deletions at TNT motifs are genetically dependent on Top1 enzyme activity in both wild-type and RNase H2 null yeast strains (Extended Data Fig. 7). Taken together with motif conservation across eukaryotes (Extended Data Fig. 6), this provides a strong additional line of experimental evidence supporting the involvement of human Topoisomerase 1 in mediating deletions. This Topoisomerase 1-associated TNT motif is consistently present in all mammalian experiments, as well as cancer and germline analyses (Fig. 4, 5; Extended Data Fig. 6, 8 and 9).

3. Biochemistry demonstrates mammalian TOP1 to cause 2 bp deletions at DNA-embedded ribonucleotides.

We cite biochemical evidence supporting a direct role for human TOP1 in causing deletions at ribonucleotides. Human TOP1 is established to cleave at genome-embedded ribonucleotides (Sekiguchi and Shuman, 1997). Significantly, sequential cleavage by purified human TOP1 at a DNA-embedded ribonucleotide was shown to generate 2 bp deletions (Huang et al., 2017). This biochemical reconstitution therefore directly demonstrates that human Topoisomerase 1 causes ID4 deletions at ribonucleotides.

We incorporate these new findings in Fig. 5, Extended Data Fig. 7, 8 and 9, and in the Results section of the revised manuscript we state :

“Furthermore, using a dataset of TOP1 cleavage events captured by TOP1-seq⁴⁴, we found 2-5 bp deletions increase in frequency with TOP1 enzymatic activity, with such deletions more prevalent in regions of high TOP1 activity (Fig. 5f). Likewise, TOP1-ID deletion rates also corresponded to TOP1 activity and transcription level, in contrast to all other deletions (Extended Data Fig. 8e,f).”

In the discussion we now state:

“TOP1 is cell-essential in mammals, and it is therefore not possible to similarly confirm a genetic dependency on TOP1 in human cells, as has been done in yeast¹⁴. However, conservation of this mechanism across eukaryotes is supported by us finding a Topoisomerase 1 dependent TNT deletion motif present in both yeast and humans, and demonstrating that deletion frequency is dependent on human TOP1 activity levels. Previously published work also provides evidence for TOP1-mutagenesis at ribonucleotide sites in humans. The reversible transesterification reaction of Type 1 Topoisomerases is conserved from yeast to humans⁷, and human TOP1 has site-specific activity for ribonucleotides⁴⁶, causing DNA breaks in mammalian RNase H2 deficient cells³⁹. Furthermore, generation of 2 bp deletions through sequential TOP1 cleavage at embedded ribonucleotides has been biochemically reconstituted with both human and yeast enzymes^{19,37}.”

[2.2]- I do not think that it can be concluded that genome-embedded ribonucleotides cause similar mutational signature in yeast and mammalian cells, but rather that RNase H2 KO causes a similar mutational signature in yeast and mammalian cells.

While ribonucleotides are a major substrate of RNase H2, the actual presence of ribonucleotides at the sites in which the ID4 signature is observed has never been demonstrated at the genomic level in mammalian cells as well as in yeast DNA.

While it is highly expected that at the genome level ID4 sites derive from TOP1 dependent activity at embedded ribonucleotides in RNase H2 null cells, showing that ribonucleotide sites correlate with sites id ID4 signatures would provide final proof of the mechanism.

We agree with the reviewer that ribonucleotides have not been directly demonstrated at deletion sites, either previously in yeast or here in mammalian cells. However, we believe this is understandable, as ribonucleotide incorporation is essentially stochastic and mutational events very rare on a per site basis in cells. Furthermore, once a mutation has occurred, a ribonucleotide is expected to be no longer present at that site. To our knowledge, mutational events at lesion sites *in situ* have not been demonstrated for other randomly occurring endogenous nucleotide lesions (e.g. those resulting from oxidative damage), but have been well accepted on inferential grounds.

The biochemical reconstitution evidence outlined above for yeast and humans provides strong direct evidence that deletions occur at ribonucleotide sites. Our findings of conserved mutational patterns and sequence motifs between yeast and humans also strongly argue for conservation of the mutational process, a mechanism that in yeast has had longstanding acceptance (Kim et al., 2011). We hope that this, in conjunction with the analysis we provide below, correlating ID4 CT/GT deletion sites with CrU/GrU sites, reassures the reviewer on this matter.

The first sentence of the Discussion << Here we establish a biological basis for the ID4 cancer signature, establishing that TOP1-mediated cleavage at genome-embedded ribonucleotides causes short deletions >> and the statements indicating mutations at << sites of genome-embedded ribonucleotides >> do not exactly reflect the work done. This study does not present any experiments with genome-embedded ribonucleotides or with ribonucleotides in general.

We would like to point out that the experiments described in Fig. 1-4 were all performed in cells with elevated levels of genome-embedded ribonucleotides, as confirmed in Extended Data Fig. 3 and 4 for RNase H2 null HeLa and RPE1 cells, and as previously shown for RNase H2 null yeast (Reijns et al., 2015) and mouse intestinal tumours (Aden et al., 2019).

However, we understand the reviewer's point that experimental findings should be distinguished from mechanistic inferences. We have therefore revised our manuscript to be more cautious in the statements we make, for instance in the first paragraph of the discussion, with the first two sentences reading as follows:

“Here we establish a biological basis for the ID4 cancer signature⁶, experimentally demonstrating it to occur in RNase H2 deficient cells both *in vitro* and *in vivo*. This implicates TOP1-mediated cleavage at genome-embedded ribonucleotides as its cause.”

While ribonucleotides embedded in DNA are major targets of RNase H2, RNase H2 has also functions beyond targeting ribonucleotides in DNA.

It is true that RNase H2 also degrades RNA:DNA hybrids, and so is likely to be important in suppressing R-loops (also raised by Reviewer 1, point 1.5), which are

also mutagenic in mammalian cells. However, they do so through promoting chromosomal rearrangements (Li and Manley, 2005) and therefore would not be expected to account for the types of SSTR/SNMH mutations examined here. Mechanistically it is difficult to envisage how R-loops could result in such small 2-5bp deletions specifically in the TNT sequence context at SNMH/SSTRs. Finally, loss of Topoisomerase 1 increases R-loop formation both in yeast (El Hage et al., 2010) and mammalian cells (Tuduri et al., 2009). Therefore, an increase in mutation rate would be expected in *top1Δ* yeast (rather than the observed decrease, Fig. 1d), if R-loops were the mechanism of action by which RNase H2 deficiency causes 2-5 bp deletions.

Moreover, also beyond RNase H2 and TOP1, there might be other ribonucleotide targeting enzymes, e.g. the DEAD-box RNA helicase DDX3X (Riva, et al. 2020), recently found in human cells.

In their recent paper Riva and colleagues demonstrated that DDX3X has endoribonuclease activity, cleaving 5' of genome-embedded ribonucleotides, albeit at micromolar rather than the nanomolar concentrations at which Ribonuclease H2 is active (Riva et al., 2020). The authors suggest DDX3X to act as an accessory RER pathway reducing genome-embedded ribonucleotides. However, this non-mutagenic removal of ribonucleotides by DDX3X, does not appear to be a confounding factor in our experiments, as the presence of substantial ribonucleotides in genomic DNA of cells and tissues was clearly demonstrated in Extended Data Fig. 3f,g and 4c,d, as well as previously published work (Aden et al., 2019; Reijns et al., 2015). Riva et al did not show an additive effect on genome-embedded ribonucleotides levels with combined depletion of RNase H2 and DDX3X. However it remains possible that such alternative RER pathways might moderate mutagenesis levels in RNase H2 deficient cells. We acknowledge this possibility in the revised discussion:

“In addition, alternative RER pathways may exist⁵³ that could reduce TOP1-mutagenesis.”

Therefore, while the assumption that the TOP1-ID4 are linked to sites of genomic ribonucleotides in DNA is undoubtedly valid, it is still an assumption.

We thank the reviewer for stating there is an undoubted validity for linking embedded ribonucleotides and TOP1-mediated deletions. We address the reviewer's point as outlined in our response above, through textual revision to distinguish between experimental observation and inferences drawn from our experiments and the existing literature.

It would be important to perform analysis on some of the published (also by the authors) yeast *rnh201*-null libraries of genomic ribonucleotide incorporation to examine whether it is possible to identify and confirm a correlation between genomic ribonucleotide sites, particularly rU sites, and sites of TOP1-ID4 signatures at least for yeast.

It may be worth noting that CrU is found strongly above the genome-wide frequency of CT in nuclear DNA of *rnh201*-null yeast strains of *S. cerevisiae*, *S. paradoxus*, and *S. pombe*, and not in wild-type RNase H2 cells in the study by Balachander et al 2020. This biased ribonucleotide distribution may suggest that CrU sites might be a preferred target of Top1.

We thank the reviewer for highlighting the relative enrichment of CrU sites in *rnh201Δ* yeast, which we had not previously noted. As suggested, we have examined genome-wide ribonucleotide incorporation patterns in our emRiboSeq datasets (Reijns et al., 2015). In agreement with the reviewer's observations, we do observe enrichment of CrU relative to other rU sequence contexts in *rnh201Δ* yeast. We also see a small increase in GrU dinucleotides over neutral expectations. We now include analysis of *rnh201Δ* ribonucleotide incorporation preference in the revised manuscript, alongside dinucleotide deletion frequencies (Extended Data Fig. 7e,f). A rank order correlation between rU sequence context and dinucleotide deletion sequence is apparent, and in line with the reviewer's expectation. Taken together with our other findings, we believe this to be consistent with Top1 cleavage site preference and ribonucleotide sequence context as two forces acting to determine deletion sites. We now state in the main text:

"Within the TNT motif, deletions were most common at CT and GT dinucleotides in both mammals and yeast (Fig. 4c; Extended Data Fig. 6 and 7b,e), which may be

explained, at least in part, by preferential incorporation of ribouridine at CT and GT dinucleotides (Extended Data Fig. 7f and ³⁸).”

Minor points:

- Should indicate in the legend what is the green bar in Figure 2c. In Figure 2c legend should explain difference between HeLa and Parental cells.

We have revised the legend as suggested:

“HeLa, no modification; Parental, HeLa with reporter (grey); RNASEH2A+, CRISPR-edited reporter clone retaining RNase H2 activity (green); KO1, KO2, CRISPR-mediated RNASEH2A knockout clones (red)”.

- Page 6 line 136, Extended data Figure 3g,h, should be f,g.

Thank you, we have corrected this.

Referee #3 (Remarks to the Author):

Reijns and colleagues investigate the mechanistic origins of ID4, a previously described mutational signature of small insertions and deletions derived from cancer genomes. The authors present a plethora of evidence from yeast, human cell lines, mouse cancers, human cancers, and the human germline to construct their argument that ID4 is due to a transcription-associated mutagenic process and they implicate TOP1 as the main mutagenic actor.

Overall, I find this paper quite exciting. Indeed, identifying the origins of a mutational signature is interesting but the authors have also shown the existence of a widespread transcription-associated indel mutagenic process and attributed it to topoisomerase 1. In my opinion, this manuscript will be of interest to a wide range of biomedical researchers. Nevertheless, before recommending it for acceptance, I do have a number of technical comments that the authors need to address to make sure that their results are robust.

We thank the reviewer for their interest in this study and highlighting its significance for a wide biomedical audience. We appreciate their critique, which has been particularly helpful to us in strengthening the computational analysis aspects of the manuscript

Technical Comments

1) The cosine similarity of ID4 to a yeast Top1-dependent mutational signature is 0.78. Please note that two nonnegative random vectors will have a cosine similarity of 0.75 purely by chance. As such, based on the patterns, the authors cannot claim that these are similar.

We thank the reviewer for highlighting this, and have revised the main text accordingly:

“Similarities to the ID4 signature were apparent with a comparable pattern of small deletions at SSTRs, although mutational events at sites of SNMH were not evident in the yeast data”.

To check whether these patterns are, indeed, similar, they can compare the TNT sequence motif in cancer samples with high levels of ID4 and the TNT sequence motif observed in yeast.

We thank the reviewer for this excellent suggestion (also made by reviewer 1). This sequence motif is indeed also present in the yeast data.

We now include this analysis in Extended Data Fig. 7, and state in the main text:

“this TNT motif was present in 100% of SNMH (n=77) and STR sites (n=124), providing a common unifying sequence context for both deletion types (Fig. 4d), a finding replicated in both our RPE1 (Extended Data Fig. 6f) and yeast datasets (Extended Data Fig. 7)”.

Importantly, demonstrating motif conservation across eukaryotes also helps us strengthen the link between these deletions and Topoisomerase 1 activity, as in yeast we are able to demonstrate that this deletion motif is dependent on Topoisomerase 1 (Extended Data Fig. 7g,h).

2) It is really not possible to visually or numerically compare the mutation spectra of WT, *rnh201Δ*, *top1Δ*, and *rnh201Δ top1Δ*. I do not find the circles in figure 1e informative and I strongly suggest replotting these using the format in figure 1a/b (at the very least add such plots to extended data). The cosine similarities are also not informative as we do not know whether these are statistically significant. Similar comment about figure 2e, 3c, 4c, 5e

As suggested, we have replotted Fig. 1e and Fig. 2e and agree that this representation improves interpretability. For Fig. 3c and Fig. 4c, mutation spectra were already plotted in the ID-83 format (original Fig. 3d, Extended Data Fig. 5b). We now include this COSMIC-style plot for the mouse tumour indels in Fig. 4a, and include similar plots for individual tumours in Extended Data Fig. 6a. Pie charts have been moved to the relevant Extended Data Figures.

For Fig. 1f, we have recalculated the cosine similarities the reviewer is referring to, using the spectra plotted in Fig. 1e. To support statistical significance, we provide bootstrap confidence estimates for strain clustering within these comparisons (Extended Data Fig. 2d). In addition, using a similar approach to Bergstrom et al. (2020) we have also calculated the null expectation for cosine similarities for pairs of random vectors, for the number of samples and mutation categories in this experiment. On this basis, both WT vs *rnh201Δ* and *top1Δ* vs *rnh201Δtop1Δ* comparisons are statistically significant ($p < 0.0001$). We now include p-values in Fig. 1f and show representative schematics for null models in Extended Data Fig. 2e.

3) Regarding the results for RNase H2 deficient RPE1 cells, it is hard to understand whether there is a clean pattern of ID4. The authors should include the indel plots for WT, RNASEH2A knockout, and RNASEH2B knockout. Moreover, currently, it is unclear whether the plot in fig 3d is a de novo signature detected from examining all samples or whether it is a plot of all samples with knockouts. If this is a de novo signature, is this signature present in the WT?

We now provide individual COSMIC ID-83 plots for WT, RNASEH2A-KO and RNASEH2B-KO lines in Extended Data Fig. 5, along with mutational signature composition analyses. This shows that ID4 contributes to the indel signature in both KO lines, whereas no such contribution is detected in any of the three WT lines.

In Fig. 3d, background mutational processes were removed by subtracting mutational events in WT cells, as done in previously published experimental studies (Zou et al., 2021). Fig. 3d therefore represents the combined AKO/BKO mutation spectra after background subtraction, to delineate events specific to RNase H2 deficiency. We apologise that this was not clearer in the original manuscript. We therefore now present mutational signature plots (Extended Data Fig. 5) and composition analyses (Fig. 3c) before and after subtraction. In addition, we state in the revised figure legend:

“**c,d**, ID4 occurs in RNase H2 null cells (**c**), and is the major signature once background mutations observed in wildtype cells are subtracted (**d**).”

Methods have also been revised to describe how de novo signature detection and decomposition was performed for these plots.

If this is a combined plot, why were knockouts combined rather than examined separately? What is the pattern of the WT and how different is it from each of the knockouts?

We had pooled data from the two lines on the basis that mutation profiles were similar between them (Extended Data Fig. 6) and functionally AKO and BKO lines are both null for RNase H2 activity. Combined analysis increased the total number of indels observed in RNase H2 null cells, which we had expected to enhance accuracy of signature decomposition.

As noted above, we have modified the legend to Fig. 3d clarifying how this plot was generated, and in the new Extended Data Fig. 5 provide ID-83 mutation profiles for each of the RPE KO and WT cell lines.

4) From figure 3d, it is visually clear that ID4 and ID5 are not the only mutational signatures required to recapitulate this pattern of mutations. For example, one can see the pattern of ID8 including the 5+ indels at 1 repeats (dark red) and the indels at microhomology (purple).

We agree that visual inspection could suggest other signatures contribute to the overall indel pattern. However, the increase in 5+ deletions (both dark red and dark

purple in the COSMIC-style ID-83 plots) is not significant when comparing wildtype and mutant cell lines (Extended Data Fig. 4e; Fisher's exact, $p = 0.17$). Also, such 5+ indels are not consistently seen across experiments; for instance, they are present in RNase H2 proficient but not KO HeLa cells in the reporter experiment (new Fig. 2e) and do not substantially contribute to the indel pattern in RNase H2 null mouse tumours (Extended Data Fig. 6a). Therefore, we believe there is insufficient evidence to assign additional signatures to RNase H2 deficiency based on this experiment.

In a recent preprint, ID8 has been attributed to mutations in TOP2A (<https://www.biorxiv.org/content/10.1101/2020.05.22.111666v3>). Does the experimental assay allow distinguishing between mutations due to TOP1 and potential mutations due to TOP2A?

This experiment does not directly distinguish between TOP1 and TOP2A. However, a key characteristic of the ID-TOP2 signature are 2-4 bp tandem insertions, a feature also seen with yeast Top2 (Stantial et al., 2020). This is not present in the RNase H2 mutational spectra in our experiments, arguing against a TOP2A contribution. Mechanistically, TOP2A associated indels also appear to be different, with insertions the result of NHEJ-mediated duplication events (Stantial et al., 2020). Further evidence that our findings are specific to TOP1 are outlined in response to the reviewer's point 9 below.

As the reviewer highlights this pre-print work, we now reference it and address the underlying point of this reviewer and reviewer 1's question, that other mutational signatures may be identified in future.

“Additional signatures associated with topoisomerases or indeed RNase H2 may be identified in future, particularly given that ID17 has been recently been linked to TOP2A^{K743N} cancers⁴⁸.”

5) The analysis of murine tumors includes samples with very few mutations (i.e., 19 and 32 mutations). Signature assignment is unreliable for such low number of mutations.

We agree that given the low number of mutations, signature assignment is less accurate on a per mouse basis for tumours 3 and 6. We therefore focus on

aggregate signature analysis across all tumours in the main figure (Fig. 4a,b). Also, instead of a per tumour signature analysis we provide COSMIC ID-83 plots in Extended Data Fig. 6a to demonstrate reproducibility in mutation profiles across the tumours.

Moreover, across all samples, ID4 accounts for only 32% of all indels. How do the author explain the low contribution of ID4?

Multiple processes contribute to somatic mutation spectra in most tumours, and intestinal cancers are well established to be multistep in aetiology. ID4 accounting for a proportion of total indels is therefore in line with our expectations, with its contribution being reduced by other commonly occurring mutational processes. Tumours are by nature highly proliferative, and therefore substantial replication slippage events (ID1/ID2) would be expected to be observed when compared to normal, non-dividing reference tissue. Such replication slippage events (ID1 and 2) are seen at high levels across many cancer types (Alexandrov et al., 2020). ID5 has similarly been shown to contribute to many cancers, including gastro-intestinal tumours (Alexandrov et al., 2020). Indel signatures such as ID4, specific to RER deficiency, will therefore inevitably be diluted by such background patterns.

We state in the text:

“Commonly occurring cancer signatures⁶ ID1, ID2 and ID5 were also seen, in line with expectations of multiple mutational processes active in neoplasia.”

6) The authors wrote “Consistent with a transcription-associated process, the ID4 signature was most evident in transcribed genomic regions (Fig. 4b).” This statement needs to be supported by statistical analysis. Based on the pattern of ID4, the authors should provide a p-value to show that there is enrichment in transcribed genomic regions than purely expected by chance. Indeed, if significant, this result provides strong evidence for a transcription-associated process. As such, the analysis should also be done for the RNase H2 deficient RPE1 cells.

We thank the reviewer for highlighting this, we now provide statistical analysis confirming that this enrichment is significant in RNase H2 deficient RPE1 cells

(Extended Data Fig. 5e), mouse tumours (Fig. 4b) and CLL (Extended data Fig. 8b), and agree that this strengthens evidence for a transcription-associated process.

7) The analysis of CLL samples is very interesting. However, why are the authors using relative percentages/contributions to deletions (fig. 5a) and indels (extended data fig. 6a)?

These will be dependent on the other indel mutational processes active in CLL. Unless I am missing something, the authors should be comparing the actual numbers of deletions and the numbers of indels instead of the percentages. Also, the presented p-values should be FDR corrected and reported as q-values.

Two cohorts were analysed in this study (Genomics England and ICGC-CLL). Variant calling rates across all genotypes differed significantly between the two cohorts, for all indels and for 2-5bp deletions (Fig. R2).

Fig. R2: All indel counts differ significantly between GEL and ICGC CLL cohorts. Indel counts per tumour (left, all indels; right, 2-5 bp deletions), for all CLL cases irrespective of *RNASEH2B* copy number status, by study. Box, 25-75%; line, median; whiskers 5-95% with data points for values outside this range. P-values, 2-sided Mann-Whitney test.

We believe this to be due to different sequence alignment tools (Isaac2 and BWA respectively). To pool these datasets, we had therefore normalised variants counts as percentages to adjust for this analytic difference. Importantly, when we analyse both cohorts separately on the basis of mutation counts, we also observe a significant increase in 2-5 bp deletions for *RNASEH2B* null CLL, demonstrating that the findings replicates between independent studies. We now provide the actual count data on a per cohort basis in Fig. 5a.

As requested, we also provide q-values (p-values corrected for a false discovery rate of 0.05), to account for the multiple tests performed in Fig. 5a and Extended Data Fig. 8a.

8) The increase of 2-5bp deletions in highly expressed housekeeping genes across PCAWG as well as their increase in the germline provides strong evidence for a transcription associated process. Is it possible to do this analysis using signature ID4 rather than using only the 2-5bp deletions?

To address the reviewer's question, we performed mutation signature analysis using SigProfiler on PCAWG cancers annotated to have a ID4 contribution. This established a correlation between signature ID4 and housekeeping gene expression levels. (Fig. R3).

Figure R3: ID4 contribution correlates with housekeeping gene expression.

n=137 PCAWG tumours with the ID4 signature, mutational events stratified by housekeeping gene expression decile, as defined in Extended Data Fig. 8c.

However, applying this to the entire PCAWG dataset was less successful. This may be due to ID4 signature contribution becoming swamped by other mutational processes in the wider dataset.

Therefore, we took an alternative approach to assess the frequency of ID4 compliant mutations per expression decile, by identifying 2-5 bp deletions at SSTR and SNMH sequences, as hallmark features of this signature. Doing so, we were able to confirm a correlation between ID4 mutation contribution and housekeeping gene transcription across the PCAWG pan-cancer dataset. We reasoned that this approach could be extended further to include the TNT sequence motif, which combined with ID4 features, defines ID-TOP1. This demonstrated a similar correlation with expression as for ID4 compliant mutations (Fig. R4)

Figure R4. ID4 and ID-TOP1 mutations correlate with housekeeping gene expression in PCAWG cancers. n = 1,830 cancers. ID4 mutations defined as 2-5 bp deletions at SSTR/SNMH sequences. ID-TOP1 defined as for ID4, but additional requirement of TNT sequence motif at deletion site. Dotted line, genome-wide rate set to 1.

We incorporate plots for ID-TOP1 versus expression in Extended Data Fig. 8 and 9, for pan-cancer and germline analyses, respectively.

9) How specific is the TNT sequence motif? Essentially, is it possible other topoisomerase or other enzymes to play a role in this mutagenic process and have a similar motif?

The TNT motif appears to be highly specific, as using our yeast reporter we find that TNT deletions are nearly entirely dependent on Top1 (Extended Data Fig. 7), indicating this to be the predominant source of such deletions, rather than other topoisomerases or other enzymes.

It remains possible that an additional enzymatic activity could have acquired similar properties to mammalian TOP1, to contribute to this mutational pattern. However, it would need to have very similar properties to account for sequence site cleavage specificity, correlation with transcription, and to also create short gaps that can undergo sequence realignment at SSTR/SNMH elements. Biochemically, this is difficult to envisage for other topoisomerases, as both TOP2 (type 2) and TOP3 (type 1A) enzymes differ from TOP1 enzymes, in forming 5' rather than 3' TOPcc intermediates. Therefore the first step of the "TNT mechanism" in which nucleophilic attack by the 2'OH present on the ribose ring from the ribonucleotide occurs against the adjacent 3'-Top1cc complex, releasing the enzyme, could not occur in the same manner. In addition, TOP2 does not have the same stringent sequence preference for thymidine at the -1 position (Spitzner and Muller, 1988).

Where other enzymes are most likely to play a role in this mutagenic process is alongside Top1 rather than in its place. For instance, enzymes such as Tdp1/Apn2 are thought to rapidly resolve most Top1-induced lesions (Top1cc and 2'3-cPs, respectively) avoiding formation of small deletions in most cases. Therefore, loss of these enzymes would promote Top1-dependent short deletions. We alluded to this possibility in the original discussion, but now state more explicitly:

"As such, this mutational process is likely to be significant not only in cancers with RER deficiency, but also those with high TOP1 activity and tumours with defects in

relevant repair mechanisms, such as enzymes that process TOP1cc⁷ or non-ligatable TOP1-induced nicks⁵⁰⁻⁵².”

Rebuttal references

- Aden, K., Bartsch, K., Dahl, J., Reijns, M.A.M., Esser, D., Sheibani-Tezerji, R., Sinha, A., Wottawa, F., Ito, G., Mishra, N., *et al.* (2019). Epithelial RNase H2 Maintains Genome Integrity and Prevents Intestinal Tumorigenesis in Mice. *Gastroenterology* *156*, 145-159.e119.
- Alexandrov, L.B., Kim, J., Haradhvala, N.J., Huang, M.N., Tian Ng, A.W., Wu, Y., Boot, A., Covington, K.R., Gordenin, D.A., Bergstrom, E.N., *et al.* (2020). The repertoire of mutational signatures in human cancer. *Nature* *578*, 94-101.
- Baranello, L., Wojtowicz, D., Cui, K., Devaiah, B.N., Chung, H.J., Chan-Salis, K.Y., Guha, R., Wilson, K., Zhang, X., Zhang, H., *et al.* (2016). RNA Polymerase II Regulates Topoisomerase 1 Activity to Favor Efficient Transcription. *Cell* *165*, 357-371.
- Bergstrom, E.N., Barnes, M., Martincorena, I., and Alexandrov, L.B. (2020). Generating realistic null hypothesis of cancer mutational landscapes using SigProfilerSimulator. *BMC Bioinformatics* *21*, 438.
- Chon, H., Sparks, J.L., Rychlik, M., Nowotny, M., Burgers, P.M., Crouch, R.J., and Cerritelli, S.M. (2013). RNase H2 roles in genome integrity revealed by unlinking its activities. *Nucleic Acids Res* *41*, 3130-3143.
- El Hage, A., French, S.L., Beyer, A.L., and Tollervey, D. (2010). Loss of Topoisomerase I leads to R-loop-mediated transcriptional blocks during ribosomal RNA synthesis. *Genes Dev* *24*, 1546-1558.
- Huang, S.N., Williams, J.S., Arana, M.E., Kunkel, T.A., and Pommier, Y. (2017). Topoisomerase I-mediated cleavage at unrepaired ribonucleotides generates DNA double-strand breaks. *EMBO J* *36*, 361-373.
- Huertas, P., and Aguilera, A. (2003). Cotranscriptionally formed DNA:RNA hybrids mediate transcription elongation impairment and transcription-associated recombination. *Mol Cell* *12*, 711-721.
- Kim, N., Huang, S.-y.N., Williams, J.S., Li, Y.C., Clark, A.B., Cho, J.-E., Kunkel, T.A., Pommier, Y., and Jinks-Robertson, S. (2011). Mutagenic processing of ribonucleotides in DNA by yeast topoisomerase I. *Science (New York, NY)* *332*, 1561-1564.
- Li, X., and Manley, J.L. (2005). Inactivation of the SR protein splicing factor ASF/SF2 results in genomic instability. *Cell* *122*, 365-378.
- Reijns, M.A.M., Kemp, H., Ding, J., de Procé, S.M., Jackson, A.P., and Taylor, M.S. (2015). Lagging-strand replication shapes the mutational landscape of the genome. *Nature* *518*, 502-506.
- Reijns, M.A.M., Rabe, B., Rigby, R.E., Mill, P., Astell, K.R., Lettice, L.A., Boyle, S., Leitch, A., Keighren, M., Kilanowski, F., *et al.* (2012). Enzymatic removal of ribonucleotides from DNA is essential for mammalian genome integrity and development. *Cell* *149*, 1008-1022.
- Riva, V., Garbelli, A., Casiraghi, F., Arena, F., Trivisani, C.I., Gagliardi, A., Bini, L., Schroeder, M., Maffia, A., Sabbioneda, S., *et al.* (2020). Novel alternative ribonucleotide excision repair pathways in human cells by DDX3X and specialized DNA polymerases. *Nucleic Acids Res* *48*, 11551-11565.
- Sekiguchi, J., and Shuman, S. (1997). Site-specific ribonuclease activity of eukaryotic DNA topoisomerase I. *Mol Cell* *1*, 89-97.

Sloan, R., Huang, S.-Y.N., Pommier, Y., and Jinks-Robertson, S. (2017). Effects of camptothecin or TOP1 overexpression on genetic stability in *Saccharomyces cerevisiae*. *DNA repair* 59, 69-75.

Spitzner, J.R., and Muller, M.T. (1988). A consensus sequence for cleavage by vertebrate DNA topoisomerase II. *Nucleic Acids Res* 16, 5533-5556.

Stantial, N., Rogojina, A., Gilbertson, M., Sun, Y., Miles, H., Shaltz, S., Berger, J., Nitiss, K.C., Jinks-Robertson, S., and Nitiss, J.L. (2020). Trapped topoisomerase II initiates formation of de novo duplications via the nonhomologous end-joining pathway in yeast. *Proc Natl Acad Sci U S A* 117, 26876-26884.

Tuduri, S., Crabbe, L., Conti, C., Tourriere, H., Holtgreve-Grez, H., Jauch, A., Pantesco, V., De Vos, J., Thomas, A., Theillet, C., *et al.* (2009). Topoisomerase I suppresses genomic instability by preventing interference between replication and transcription. *Nat Cell Biol* 11, 1315-1324.

Zimmermann, M., Murina, O., Reijns, M.A.M., Agathangelou, A., Challis, R., Tarnauskaite, Ž., Muir, M., Fluteau, A., Aregger, M., McEwan, A., *et al.* (2018). CRISPR screens identify genomic ribonucleotides as a source of PARP-trapping lesions. *Nature* 559, 285-289.

Zou, X., Koh, G.C.C., Nanda, A.S., Degasperi, A., Urgo, K., Roumeliotis, T.I., Agu, C.A., Badja, C., Momen, S., Young, J., *et al.* (2021). A systematic CRISPR screen defines mutational mechanisms underpinning sig s caused by replication errors and endogenous DNA damage. *Nat Cancer* 2, 643-657.

Reviewer Reports on the First Revision:

Referee #1 (Remarks to the Author):

The authors have suitably addressed all of my concerns.

Referee #2 (Remarks to the Author):

The authors have significantly worked to strengthen the study and their findings. They have satisfactorily addressed the critical points that were raised by this reviewer.

Referee #3 (Remarks to the Author):

The authors have addressed my comments.

Author Rebuttals to First Revision:

N/A